# Functional Linear Regression of Cumulative Distribution Functions

**Qian Zhang**                                              *zhan3761@purdue.edu*
*Department of Statistics*
*Purdue University*

**Anuran Makur**                                            *amakur@purdue.edu*
*Department of CS and School of ECE*
*Purdue University*

**Kamyar Azizzadenesheli**                                  *kamyara@nvidia.com*
*Nvidia Corporation*

**Reviewed on OpenReview:** *https://openreview.net/forum?id=ZOqJCP4eMk*

## Abstract

The estimation of cumulative distribution functions (CDF) is an important learning task with a great variety of downstream applications, such as risk assessments in predictions and decision making. In this paper, we study functional regression of contextual CDFs where each data point is sampled from a linear combination of context dependent CDF basis functions. We propose functional ridge-regression-based estimation methods that estimate CDFs accurately everywhere. In particular, given $n$ samples with $d$ basis functions, we show estimation error upper bounds of $\widetilde{O}(\sqrt{d/n})$ for fixed design, random design, and adversarial context cases. We also derive matching information theoretic lower bounds, establishing minimax optimality for CDF functional regression. Furthermore, we remove the burn-in time in the random design setting using an alternative penalized estimator. Then, we consider agnostic settings where there is a mismatch in the data generation process. We characterize the error of the proposed estimators in terms of the mismatched error, and show that the estimators are well-behaved under model mismatch. Moreover, to complete our study, we formalize infinite dimensional models where the parameter space is an infinite dimensional Hilbert space, and establish a self-normalized estimation error upper bound for this setting. Notably, the upper bound reduces to the $\widetilde{O}(\sqrt{d/n})$ bound when the parameter space is constrained to be $d$-dimensional. Our comprehensive numerical experiments validate the efficacy of our estimation methods in both synthetic and practical settings.

## 1 Introduction

Estimating cumulative distribution functions (CDF) of random variables is a salient theoretical problem that underlies the study of many real-world phenomena. For example, Huang et al. (2021) and Liu et al. (2022) recently showed that estimating CDFs is sufficient for risk assessment, thereby making CDF estimation a key building block for such decision-making problems. In a similar vein, it is known that CDFs can also be used to directly compute distorted risk functions (Wirch & Hardy, 2001), coherent risks (Artzner et al., 1999), conditional value-at-risk and mean-variance (Cassel et al., 2023), and cumulative prospect theory risks (Prashanth et al., 2016). Furthermore, CDFs are also useful in calculating various risk functionals appearing in insurance premium design, portfolio design, behavioral economics, behavioral finance, and healthcare applications (Rockafellar et al., 2000; Shapiro et al., 2014; Prashanth et al., 2016; Wong et al., 2022). Given the broad utility of estimating CDFs, there is a vast (and fairly classical) literature that tries to understand this problem.

In particular, the renowned *Glivenko-Cantelli theorem* (Cantelli, 1933; Glivenko, 1933) states that given independent samples of a random variable, one can construct a consistent estimator for its CDF. Tight non-asymptotic sample complexity rates for such estimation using the Kolmogorov-Smirnov (KS) distance as the loss have also been established in the literature (Cantelli, 1933; Glivenko, 1933; Dvoretzky et al., 1956; Massart, 1990). However, these results are all limited to the setting of a single random variable. In contrast, many modern learning problems, such as doubly-robust estimators in contextual bandits, treatment effects, and Markov decision processes (Huang et al., 2021; Kallus et al., 2019; Huang et al., 2022), require us to simultaneously learn the CDFs of potentially infinitely many random variables from limited data. Hence, the classical results on CDF estimation do not address the needs of such emerging learning applications.

**Contributions.** In this work, as a first step towards developing general CDF estimation methods that fulfill the needs of the aforementioned learning problems, we study *functional linear regression of CDFs*, where samples are generated from CDFs that are convex combinations of context-dependent CDF bases. Our model resembles the well-studied linear regression and stochastic linear bandits problem. In linear regression, researchers analyzed finite-dimensional parametric models with pre-selected feature functions. These pre-designed features result from extensive feature engineering processes carried out for the underlying task. Similarly, within the domain of contextual bandits, researchers studied the stochastic linear bandit problem using a linear model (Lattimore & Szepesvári, 2020, Equation (19.1)) with finite dimension and known feature map. Thus, it is natural to commence the analysis assuming the access to known "feature" CDFs, which ultimately bestows the advantages intrinsic to linear regression. As our main contribution, we define both least-squares regression and ridge regression estimators for the unknown linear weight parameter, and establish corresponding estimation error bounds for the fixed design, random design, adversarial, and self-normalized settings. In particular, given $n$ samples with $d$ CDF bases, we prove estimation error upper bounds that scale like $\widetilde{O}(\sqrt{d/n})$ (neglecting sub-dominant factors). Our derivations are inspired by the classical finite dimensional fixed design, random design, and adversarial self-normalized theories (Peña et al., 2008; Abbasi-Yadkori et al., 2011b). Our results achieve the same problem-dependent scaling as in canonical finite dimensional linear regression (Abbasi-Yadkori et al., 2011b;a; Hsu et al., 2012b), and importantly, in contrast to the mentioned works, do not depend on the label/reward/response magnitude. Moreover, we derive $\Omega(\sqrt{d/n})$ information theoretic lower bounds for functional linear regression of CDFs. This establishes minimax estimation rates of $\widetilde{\Theta}(\sqrt{d/n})$ for the CDF functional regression problem. We later show that this result directly implies the concentration of CDFs in KS distance. We also propose a new penalized estimator that theoretically eliminates the requirement on the burn-in time of sample size in the random design setting. Then, we consider agnostic settings where there is a mismatch between our linear model and the actual data generation process. We characterize the estimation error of the proposed estimator in terms of the mismatch error, and demonstrate that the estimator is well-behaved under model mismatch. To complete our study, we generalize the parameter space in the linear model from finite-dimensional Euclidean spaces to general infinite-dimensional Hilbert spaces, extend the ridge regression estimator to the infinite-dimensional model with proper regularization, and establish a corresponding self-normalized estimation error upper bound which immediately recovers our previous $\widetilde{O}(\sqrt{d/n})$ upper bound when the parameter space is restricted to be $d$-dimensional. Finally, we present numerical results for synthetic and real data experiments to illustrate the performance of our estimation methods.

**Related works.** A complementary approach to the proposed CDF regression framework is quantile regression (Koenker & Bassett Jr, 1978). Although quantile regression may appear to be closely related to CDF regression at first glance, the two problems have very different flavors. Indeed, unlike CDFs, quantiles are not sufficient for law invariant risk assessment. Besides, due to their infinite range, quantile estimation is quite challenging, resulting in analyses that only consider pointwise estimation (Takeuchi et al., 2006). However, as it is necessary to estimate multiple quantiles for CDF estimation, a simultaneous analysis of multiple quantile estimates is needed theoretically, which typically requires a union bound on the failure probability that increases linearly with the number of estimates. Furthermore, the estimated multiple quantiles may not be monotonically increasing with respect to the probability values, requiring extra effort to construct a valid CDF from a finite series of quantile estimates. Moreover, any such construction will incur a non-convergent KS distance between the estimated CDF and the true CDF for some distribution, as a general CDF can exhibit jumps or flat regions at any position. Additionally, the quality of the estimated CDF from multiple estimated

quantiles relies heavily on the selection of grid points of probability values, which is instance-dependent and may require knowledge of the distribution the learner seeks to estimate. Thus, establishing a universal rule for choosing grid points that yield reasonable CDF estimates via quantile regression across diverse distributions proves challenging. In practice, the introduction of grid points introduces numerous hyperparameters to tune, adding artificial complexity to the methodology. Perhaps more importantly, quantile regression can be ill-posed in many machine learning settings. For example, quantiles are not estimatable in decision-making problems and games with mixed random variables (which take both discrete and continuous values). For these reasons, our focus in this paper will be on CDF regression.

Several works have delved into the realm of conditional CDF estimation. Hall et al. (1999) estimated conditional CDFs for fixed cutoff $y$ and context $x$ using local logistic methods and adjusted Nadaraya-Watson estimators. However, their analysis necessitates the assumption of strong regularity conditions on the conditional CDF (including at least continuous second-order derivatives), the marginal CDF of the context, and the data generating process. They established asymptotic convergence only for fixed cutoff and context. Ferraty et al. (2006) introduced a kernel-type nonparametric estimator for conditional CDFs at a fixed context $x$. Their analysis mandates that the samples are independent and identically distributed (iid), in addition to some regularity assumptions concerning the marginal distribution of $x$ and the smoothness of the conditional CDF. Their theoretical findings, too, revolve around asymptotic scenarios and apply solely to fixed contexts. Chung & Dunson (2009) proposed a special class of conditional CDFs based on probit stick-breaking process mixture models. They developed an MCMC algorithm for posterior sampling of parameters but did not furnish theoretical assurances regarding consistency. Distinguishing itself from existing endeavors, this paper introduces a novel linear model (1) or (18) where we presume knowledge of an arbitrary family of contextual CDFs and aim to estimate the weight parameter $\theta_*$. Consequently, our model possesses the capability to encompass any conditional CDF, enabling the estimation of the conditional CDF across all values of the context $x$ and cutoff $y$ by estimating one parameter. Furthermore, we embrace an adversarial data generation process (see Scheme I in Section 2), which surpasses the limitations of the iid setting in terms of generality. We provide tight non-asymptotic analysis of the estimation error by showing matching upper bounds and lower bounds of the error. Additionally, our model (1) or (18) readily accommodates the integration of estimated CDFs from previous works on conditional CDF estimation into the family of feature contextual CDFs, thereby enhancing the overall quality of the conditional CDF estimates. Furthermore, the probability approximately correct and Vapnik–Chervonenkis theory (Devroye et al., 2013) has been extended to CDF with new measures of complexities (Liu et al., 2022).

Chernozhukov et al. (2013) and Koenker et al. (2013) study "distribution regression" where for a fixed cutoff $y$, they estimate parameters in conditional CDF models by maximizing log likelihood of $\mathbb{1}\{y \geq Y_i\}$ for outcome samples $Y_1, \ldots, Y_n$. Thus, both works require specific models for conditional CDFs. Chernozhukov et al. (2013) introduced a "distribution regression" model where the conditional CDF takes the form of a link function evaluated at the inner product of vector transformations of the context $X$ and outcome $Y$. However, due to the dependence of the log likelihood on the cutoff $y$ within this model, their estimator is inherently pointwise. They established asymptotic convergence of the estimated conditional CDF. Nonetheless, this hinges on certain assumptions concerning the true parameter functions, which is challenging to validate. Koenker et al. (2013) considered the "linear local-scale model" where the outcome is the summation of a linear local function of the context and the product of a linear scale function of the context and an independent random error boasting a smooth density. Their convergence results are of an asymptotic nature, assuming iid samples, alongside other conditions on the expected log likelihood and the asymptotic covariance function which also pose substantial verification challenges. Furthermore, the maximum likelihood estimation (MLE) used in both papers only accesses the indicators denoting whether the samples $Y_1, \ldots, Y_n$ surpass a fixed cutoff $y$, which underutilizes the wealth of information inherent in the samples. In stark contrast, our estimator (2) or (21) uses the one-sample empirical CDFs ($\mathbb{1}\{Y_i \leq \cdot\}$) which fully exploit the sample information. Moreover, as previously mentioned, the estimated CDFs derived in the above distribution regression problems can be seamlessly integrated into our proposed model.

In some literature, "distribution regression" takes on a distinctive meaning, referring to the model where the context is a sequence of samples from some distribution which, together with the outcome, is sampled from some meta joint distribution (Póczos et al., 2013; Szabó et al., 2016). The task is to learn a mapping from

the distribution of the context to the outcome. Contrastingly, our model (1) or (18) operates in a different realm: the outcome is a sample from a mixture of contextual CDFs and the task is to learn the weight parameter $\theta_*$. Thus, our model diverges from the above notion of distribution regression. Our focus is not on estimating a mapping from distributions to outcomes but on estimating a parameter that governs the condition distribution. Moreover, there is no meta distribution that the samples follow in our adversarial data generating process.

Our results on CDF regression have the potential for downstream applications in stochastic bandits (Thompson, 1933; Robbins, 1952; Lattimore & Szepesvári, 2020) where learning algorithms necessitate the estimation of reward distributions under selected actions and the adaptive exploration of the action space. Then, under the linear assumption of the reward distributions, our CDF regression method can serve to estimate the CDFs of the rewards in stochastic bandit algorithms, with readily available theoretical results for integration into the analysis. For instance, in the infinite-armed bandit problem (Berry et al., 1997; Wang et al., 2022), assuming that the underlying distribution of arms satisfies our linear model, our method, in conjunction with an exploration algorithm for arm selection, can be employed to estimate the CDF of the underlying distribution, which actually enables the estimation of any distribution functional, thereby broadening the class of indicator-based functionals considered in Wang et al. (2022). Furthermore, since estimating the CDF of the reward under a target policy in stochastic bandits is adequate for assessing various risk functionals associated with the target policy (Huang et al., 2021), with our linear assumption on the reward distribution, our method can be applied to the risk assessment of policies in stochastic bandits. Then, combined with an exploration algorithm to select policies, our method becomes a valuable tool for minimizing diverse risks in stochastic bandits, which also extends the conventional scope of minimizing expected regret in stochastic bandits.

**Outline.** We briefly outline the rest of the paper. Notation and formal setup for our problem are given in Section 2. We propose our estimation paradigm and analyze its theoretical performance in Section 3. We derive corresponding lower bounds on the estimation error in Section 4. We establish upper bounds on the estimation error under the existence of a mismatch in our proposed model in Section 5. We generalize the problem from estimating finite dimensional parameters to estimating infinite dimensional parameters, extend our estimation paradigm to this infinite dimensional setting, and prove an upper bound on estimation error in Section 6. Numerical results are displayed in Section 7. Conclusions are drawn and future research directions are suggested in Section 8. All the proofs and additional results are presented in the appendices.

## 2 Preliminaries

In this section, we introduce the notation used in the paper and set up the learning problem of contextual CDF regression.

**Notation.** Let $\mathbb{N}$ denote the set of positive integers. For any $n \in \mathbb{N}$, let $[n]$ denote the set $\{1, \ldots, n\}$. For any measure space $(\Omega, \mathcal{F}, \mathfrak{m})$, define the Hilbert space $\mathcal{L}^2(\Omega, \mathfrak{m}) := \{f : \Omega \to \mathbb{R} \mid \int_\Omega |f|^2 d\mathfrak{m} < \infty\}$ with $\mathcal{L}^2$-norm $\|f\|_{\mathcal{L}^2(\Omega,\mathfrak{m})} := \sqrt{\int_\Omega |f|^2 d\mathfrak{m}}$ for $f \in \mathcal{L}^2(\Omega, \mathfrak{m})$. For any positive definite matrix $A \in \mathbb{R}^{d \times d}$, define $\|\cdot\|_A$ to be the weighted $\ell^2$-norm in $\mathbb{R}^d$ induced by $A$, i.e., $\|x\|_A = \sqrt{x^\top A x}$ for $x \in \mathbb{R}^d$. For the standard Euclidean (or $\ell^2$-) norm $\|\cdot\|_{I_d}$, where $I_d$ denotes the $d \times d$ identity matrix, we omit the subscript $I_d$ and simply write $\|\cdot\|$. For any square matrix $A$, let $\mu_{\min}(A)$ denote the smallest eigenvalue of $A$, $\mu_{\max}(A)$ denote the largest eigenvalue of $A$ and $\|A\|_2$ denote the spectral norm of the matrix $A$, i.e., $\|A\|_2 := \sqrt{\mu_{\max}(A^\top A)}$. Let $\mathrm{KS}(F_1, F_2) := \sup_{x \in \mathbb{R}} |F_1(x) - F_2(x)|$ denote the KS distance between two CDFs $F_1$ and $F_2$. Finally, let $\mathbb{1}\{\cdot\}$ denote the indicator function. More technical notation dealing with measurability issues is provided at the beginning of Appendix B.

**Problem setup.** In this paper, we consider the problem of functional linear regression of CDFs. To define this problem, let $\mathcal{X}$ denote the context space, and let $F(x, \cdot) : \mathbb{R} \to [0, 1]$ be the CDF of some $\mathbb{R}$-valued random variable for any $x \in \mathcal{X}$. We assume that $\mathcal{X}$ is a Polish space throughout the paper. For a context

$x \in \mathcal{X}$, we observe a sample $y$ from its corresponding CDF $F(x, \cdot)$. We next summarize two schemes to generate $(x, y)$ samples:

- **Scheme I (Adversarial).** For each $j \in \mathbb{N}$, an adversary picks $x^{(j)} \in \mathcal{X}$ (either deterministically or randomly) in an adaptive way given knowledge of the previous $y^{(i)}$'s for $i < j$, and then $y^{(j)} \in \mathbb{R}$ is sampled from $F(x^{(j)}, \cdot)$. This includes the canonical **fixed design** setting as a special case, where all $x^{(j)}$'s are fixed a priori without knowledge of $y^{(j)}$'s.

- **Scheme II (Random).** For each $j \in \mathbb{N}$, $x^{(j)} \in \mathcal{X}$ is sampled from some probability distribution $P_X^{(j)}$ on $\mathcal{X}$ independently, and then $y^{(j)} \in \mathbb{R}$ is sampled from $F(x^{(j)}, \cdot)$ independently. This is known as the **random design** setting in the regression context.

Scheme I and Scheme II generalize the assumptions of the data generation process in canonical ridge regression in Abbasi-Yadkori et al. (2011a) and Hsu et al. (2012b) to the problem of CDF estimation, respectively. Note that although the random design setting in Scheme II is a special case of Scheme I, we emphasize it because it has specific properties that deserve a separate treatment. The adversarial setting in Scheme I is more general than what is typically considered for regression, and our corresponding self-normalized analysis has several potential future applications in risk assessment for reinforcement learning, e.g., in contextual bandits (Abbasi-Yadkori et al., 2011a).

The task of contextual CDF regression is to recover $F$ from a sample $\{(x^{(j)}, y^{(j)})\}_{j \in [n]}$ of size $n$. As an initial step towards this problem, inspired by the well-studied linear regression and linear contextual bandits problems (Lattimore & Szepesvári, 2020, Equation (19.1)), where finite-dimensional parametric models with pre-selected feature functions are assumed, we consider a *linear model* for $F$. Let $d$ be a fixed positive integer. For each $i \in [d]$ and $x \in \mathcal{X}$, let $\phi_i(x, \cdot) : \mathbb{R} \to [0, 1]$ be a feature function that is a CDF of a $\mathbb{R}$-valued random variable with range contained in some Borel set $S \subseteq \mathbb{R}$, and assume that $\phi_i$ is measurable. Then, we define the vector-valued function $\Phi : \mathcal{X} \times \mathbb{R} \to [0, 1]^d$, $\Phi(x, t) = [\phi_1(x, t), \ldots, \phi_d(x, t)]^{\top}$. We assume that there exists some *unknown* $\theta_* \in \Delta^{d-1}$, where $\Delta^{d-1} := \{(\theta_1, \ldots, \theta_d) \in \mathbb{R}^d : \sum_{i=1}^{d} \theta_i = 1, \theta_i \geqslant 0 \text{ for } 1 \leqslant i \leqslant d\}$ denotes the probability simplex in $\mathbb{R}^d$, such that,

$$F(x, t) = \theta_*^{\top} \Phi(x, t), \quad \forall \ x \in \mathcal{X}, \ t \in \mathbb{R}. \tag{1}$$

Thus, we can view $\Phi$ as a "basis" for contextual CDF learning.

We visualize the sample generation process in Figure 1 where the contextual CDFs are shown in the left column and the one-sample empirical CDFs ($\mathbb{1}\{y \leqslant \cdot\}$ for sample $y$) are shown in the right column. It is worth mentioning the differences between our model and the mixture model with known basis distributions in the statistics literature. First, the basis distributions in our model depend on the context of the sample and are not fixed. Second, in mixture models, the samples are assumed to be independent while in our Scheme I, the samples can be dependent since $x^{(j)}$ is picked adversarially given knowledge of the previous $y^{(i)}$'s. Thus, the mixture model with known basis distributions only corresponds to the fixed design setting with the same context $x^{(j)} = x$ for all samples.

As explained in the sampling schemes above, given $x^{(j)}$ at the $j$th sample, the observation $y^{(j)}$ is generated according to the CDF $F(x^{(j)}, \cdot) = \theta_*^{\top} \Phi(x^{(j)}, \cdot)$. For notational convenience, we will often refer to the vector-valued function $\Phi(x^{(j)}, \cdot)$ as $\Phi_j(\cdot)$ for all $j \in [n]$, so that $F(x^{(j)}, \cdot) = \theta_*^{\top} \Phi_j(\cdot)$. Under the linear model in (1), our *goal is to estimate the unknown parameter $\theta_*$* from the sample $\{(x^{(j)}, y^{(j)})\}_{j \in [n]}$ in a (regularized) least-squares error sense. This in turn recovers the contextual CDF function $F$.

## 3 Upper bounds on estimation error

In this section, we propose an estimation paradigm for the unknown parameter $\theta_*$ in Section 3.1, derive the upper bounds on the associated estimation error in Section 3.2, and propose a new penalized estimator that theoretically eliminates the burn-in time of the sample size in the random setting in Section 3.3.

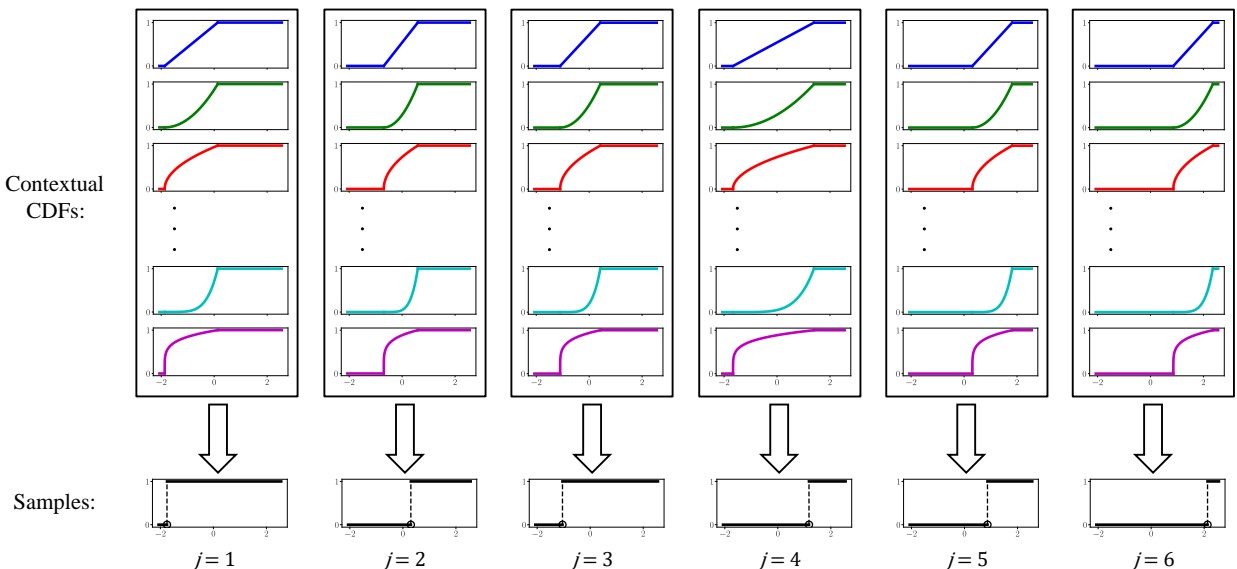

Figure 1: A visualization of the data generating process. For each $j \in [6]$ with context $x^{(j)} \in \mathcal{X}$, the upper row shows the $d$ contextual CDFs ($\phi_i(x^{(j)}, \cdot)$, $i \in [d]$) under the context $x^{(j)}$. For $y^{(j)}$ drawn from the CDF $F(x^{(j)}, \cdot) = \theta_*^\top \Phi(x^{(j)}, \cdot)$ where $\Phi(x^{(j)}, \cdot) := [\phi_i(x^{(j)}, \cdot), \dots, \phi_d(x^{(j)}, \cdot)]^\top$, the bottom row shows the sample empirical CDF $\mathrm{I}_{y^{(j)}}(\cdot) := \mathbb{1}\{y^{(j)} \leqslant \cdot\}$.

### 3.1 Ridge regression estimator

We begin by formally stating our least-squares functional regression optimization problem to learn $\theta_*$. Given a probability measure $\mathfrak{m}$ on $S$, the sample $\{(x^{(j)}, y^{(j)})\}_{j \in [n]}$, and the set of basis functions $\{\Phi_j\}_{j \in [n]}$, we propose to estimate $\theta_*$ by minimizing the (ridge or) $\ell^2$-regularized squared $\mathcal{L}^2(S, \mathfrak{m})$-distance between the estimated and empirical CDFs:

$$\widehat{\theta}_\lambda := \arg\min_{\theta \in \mathbb{R}^d} \sum_{j=1}^n \|\mathrm{I}_{y^{(j)}} - \theta^\top \Phi_j\|_{\mathcal{L}^2(S, \mathfrak{m})}^2 + \lambda\|\theta\|^2, \tag{2}$$

where $\lambda \geqslant 0$ is the hyper-parameter that determines the level of regularization, and the function observation $\mathrm{I}_{y^{(j)}}(t) := \mathbb{1}\{y^{(j)} \leqslant t\}$ is an empirical CDF of $y^{(j)}$ that forms an unbiased estimator for $F(x^{(j)}, \cdot)$ conditioned on past contexts and observations. Hence, in Scheme I, we only require that $\mathrm{I}_{y^{(j)}} - \theta^\top \Phi_j$ is a zero-mean function given past contexts and observations, making our analysis suitable for online learning problems where the later contexts can depend on the past contexts and observations. We remark that the adoption of $\mathcal{L}^2$-distance in (2) is natural. Indeed, researchers have considered the $\mathcal{L}^2$-distance between a one-sample empirical CDF and a CDF estimate in the definition of Continuous Ranked Probability Score (CRPS) (Hersbach, 2000) to assess the performance of the CDF estimate in approximating data distributions. In fact, viewing the one-sample empirical CDF as the response and the basis contextual CDFs as the feature in linear regression, it is natural to consider the least squares method, precisely corresponding to minimizing the $\mathcal{L}^2$-distance in our functional setting. Notice further that $\widehat{\theta}_\lambda$ in (2) is an *improper estimator* since it may not lie in $\Delta^{d-1}$. However, since $\Delta^{d-1}$ is compact in $\mathbb{R}^d$, $\widetilde{\theta}_\lambda := \arg\min_{\vartheta \in \Delta^{d-1}} \|\vartheta - \widehat{\theta}_\lambda\|_A$ exists for any positive definite $A \in \mathbb{R}^{d \times d}$. Moreover, since $\Delta^{d-1}$ is also convex, we have $\|\widetilde{\theta}_\lambda - \theta\|_A \leqslant \|\widehat{\theta}_\lambda - \theta\|_A$ (Beck, 2014, Theorem 9.9) for any $\theta \in \Delta^{d-1}$ including $\theta_*$. This means that an upper bound on $\|\widehat{\theta}_\lambda - \theta_*\|_A$ is also an upper bound on $\|\widetilde{\theta}_\lambda - \theta_*\|_A$. Additionally, as we will see later, the $\widehat{\theta}_\lambda$ has a closed-form analytic solution which benefits the analysis of the estimation error. Therefore, we focus our analysis on the improper estimator $\widehat{\theta}_\lambda$, noting that its projection onto $\Delta^{d-1}$ yields an estimator $\widetilde{\theta}_\lambda$ for which the same upper bounds hold.

When $\lambda > 0$, the objective function in (2) is a $(2\lambda)$-strongly convex function of $\theta \in \mathbb{R}^d$ (see, e.g., Bertsekas et al., 2003, for the definition), and is uniquely minimized at

$$\widehat{\theta}_\lambda = \left( \sum_{j=1}^n \int_S \Phi_j \Phi_j^\top d\mathfrak{m} + \lambda I_d \right)^{-1} \left( \sum_{j=1}^n \int_S \mathrm{I}_{y^{(j)}} \Phi_j d\mathfrak{m} \right). \tag{3}$$

For the unregularized case where $\lambda = 0$, we omit the subscript $\lambda$ and write $\widehat{\theta}$ to denote a corresponding estimator in (2). Note that when $\lambda = 0$, if $\mu_{\min}(\sum_{j=1}^n \int_S \Phi_j \Phi_j^\top d\mathfrak{m}) > 0$, the objective function in (2) is still strongly convex, and is uniquely minimized at $\widehat{\theta}$ given in (3) with $\lambda = 0$. In practice, one can deploy standard numerical methods to compute the integral in (3), and the computational complexity of the matrix inversion is cubic in the dimension $d$. However, iterative methods can be used to obtain better dimension dependence in the running time. As a remark, since the probability density functions (PDFs) of the basis distributions may not exist, the samples in Scheme I can be dependent, and the distributions of the contexts in Scheme II are unknown, the likelihood function of the samples generally does not exist in our problem setting, which rules out the usage of MLE. But our estimator (2) always exists. Moreover, we focus on non-asymptotic analysis of our estimator and prove self-normalized upper bounds for the estimation error, which is rarely analyzed for MLEs.

Lastly, it is worth remarking upon the choice of measure $\mathfrak{m}$ used above. In order for the estimator in (2) to be well-defined, since $\mathrm{I}_y(t), \theta^\top \Phi(x,t) \in [0,1]$ for any $t, y \in \mathbb{R}$ and $x \in \mathcal{X}$, it suffices to ensure that $\mathfrak{m}(S) < \infty$ (i.e., $\mathfrak{m}$ is a finite measure). This is the reason why we restrict $\mathfrak{m}$ to be a probability measure on $S$. Furthermore, the probability measure $\mathfrak{m}$ can in general be chosen to adapt to specific problem settings. For example, the uniform measure $\mathfrak{m}_\mathsf{U}$ on $S$ is often easy to compute for some choices of $S$. Specifically, if $0 < \mathsf{Leb}(S) < \infty$, where $\mathsf{Leb}$ denotes the Lebesgue measure, $\mathfrak{m}_\mathsf{U}$ is defined by $\frac{d\mathfrak{m}_\mathsf{U}}{d\mathsf{Leb}} = \frac{1}{\mathsf{Leb}(S)}$, where $\frac{d\mathfrak{m}_\mathsf{U}}{d\mathsf{Leb}}$ is the Radon-Nikodym derivative. If $S$ is a finite set with cardinality $\#S$, $\mathfrak{m}_\mathsf{U} = \frac{1}{\#S} \sum_{s \in S} \delta_s$, where $\delta_s$ denotes the Dirac measure at $s$. On the other hand, when $S = \mathbb{R}$, $\mathfrak{m}$ can be set to the Gaussian measure $\gamma_{c,\sigma^2}$ defined by $\gamma_{c,\sigma^2}(dx) = \frac{1}{\sqrt{2\pi\sigma^2}} e^{-(x-c)^2/(2\sigma^2)} dx$ with $c \in \mathbb{R}$ and $\sigma^2 > 0$.

### 3.2 Self-normalized bounds in various settings

For samples generated according to Scheme I, we prove self-normalized upper bounds on the error $\widehat{\theta}_\lambda - \theta_*$. For any probability measure $\mathfrak{m}$ on $S$, define $U_n := \sum_{j=1}^n \int_S \Phi_j \Phi_j^\top d\mathfrak{m}$ and $U_n(\lambda) = U_n + \lambda I_d$ for $n \in \mathbb{N}$ and $\lambda \geqslant 0$. For $n, d \in \mathbb{N}$, $\lambda, \tau \in (0, \infty)$, and $\delta \in (0, 1)$, define

$$\varepsilon_\lambda(n, d, \delta) := \sqrt{d \log(1 + n/\lambda) + 2\log(1/\delta)} + \sqrt{\lambda} \|\theta_*\| \quad \text{and} \tag{4}$$

$$\varepsilon(n, d, \delta, \tau) := \left( \sqrt{d} + \sqrt{8d \log(1/\delta)} + \frac{4}{3} \sqrt{d/n} \log(1/\delta) \right) / \sqrt{\tau}. \tag{5}$$

The next theorem states our self-normalized upper bound on the estimation error.

**Theorem 1** (Self-normalized bound in adversarial setting). *Assume $\mathfrak{m}$ is a probability measure on $S$ and $\{(x^{(j)}, y^{(j)})\}_{j \in \mathbb{N}}$ is sampled according to Scheme I with $F$ defined in (1). For any $\lambda > 0$ and $\delta \in (0, 1)$, with probability at least $1 - \delta$, for all $n \in \mathbb{N}$, the estimator defined in (2) satisfies*

$$\|\widehat{\theta}_\lambda - \theta_*\|_{U_n(\lambda)} \leqslant \varepsilon_\lambda(n, d, \delta). \tag{6}$$

Moreover, for the unregularized case, we have the following result.

**Proposition 2** (Self-normalized bound in adversarial setting for unregularized estimator). *Under the same assumptions as Theorem 1, if $U_N$ is positive definite for a fixed $N \in \mathbb{N}$, then for any $\delta \in (0, 1)$ and $n \geqslant N$, with probability at least $1 - \delta$, the estimator defined in (2) with $\lambda = 0$ satisfies*

$$\|\widehat{\theta} - \theta_*\|_{U_n} \leqslant \varepsilon(n, d, \delta, \mu_{\min}(U_n)/n). \tag{7}$$

The proofs of Theorem 1 and Proposition 2 are provided in Appendix B.1. Informally, Theorem 1 and Proposition 2 convey that with high probability, the self-normalized errors $\|\widehat{\theta}_\lambda - \theta_*\|_{U_n(\lambda)}$ and $\|\widehat{\theta} - \theta_*\|_{U_n}$

scale as $\widetilde{O}(\sqrt{d})$ in the $\ell^2$-regularized and unregularized cases, where $\widetilde{O}(\cdot)$ ignores logarithmic and other subdominant factors. We note that Theorem 1 and Proposition 2 also imply upper bounds on the (un-normalized) error $\|\widehat{\theta}_\lambda - \theta_*\|$. Indeed, for any positive definite matrix $A \in \mathbb{R}^{d \times d}$ and vector $a \in \mathbb{R}^d$, we have $\|a\| \leqslant \mu_{\min}(A)^{-1/2}\|a\|_A$. Thus, for example, (6) in Theorem 1 implies that $\|\widehat{\theta}_\lambda - \theta_*\| \leqslant \mu_{\min}(U_n(\lambda))^{-1/2}\varepsilon_\lambda(n, d, \delta) = \widetilde{O}\big(\sqrt{d/(1 + \mu_{\min}(U_n))}\big)$ with high probability. Then, for the projected estimator $\widetilde{\theta}_\lambda \in \Delta^{d-1}$, we have $\|\widetilde{\theta}_\lambda - \theta_*\| \leqslant \widetilde{O}\big(\min\{1, \sqrt{d/(1 + \mu_{\min}(U_n))}\}\big)$ by the property of $\Delta^{d-1}$. When $\mu_{\min}(U_n) = \Theta(n)$, we have $\|\widehat{\theta}_\lambda - \theta_*\| = \widetilde{O}\big(\sqrt{d/n}\big)$.

The key idea in the proof of Theorem 1 is to first notice that $\widehat{\theta}_\lambda - \theta_* = U_n(\lambda)^{-1}W_n - U_n(\lambda)^{-1}(\lambda\theta_*)$, where $W_n := \sum_{j=1}^n \int_S (\mathrm{I}_{y^{(j)}}\Phi_j - \theta_*^\top \Phi_j \Phi_j)d\mathfrak{m}$. We next show that $\{\overline{M}_n\}_{n \geqslant 0}$ where $\overline{M}_n := \frac{\lambda^{d/2}}{\det(U_n(\lambda))^{1/2}} \exp\left(\frac{1}{2}\|W_n\|_{U_n(\lambda)^{-1}}^2\right)$ is a super-martingale. *Doob's maximal inequality* for super-martingales is then used in conjunction with some careful algebra to establish (6). To prove Proposition 2, we use a vector *Bernstein inequality* for bounded martingale difference sequences (Hsu et al., 2012a, Proposition 1.2) to show a high probability upper bound for $\|W_n\|$. Note that $U_N$ being positive definite implies that $U_n$ is positive definite for $n \geqslant N$. Since $\|\widehat{\theta} - \theta_*\|_{U_n} = \|W_n\|_{U_n^{-1}} \leqslant \|W_n\|/\sqrt{\mu_{\min}(U_n)}$, we establish (7).

Since the fixed design is a special case of the adversarial setting, Theorem 1 and Proposition 2 imply the same $\widetilde{O}(\sqrt{d})$-style upper bounds as a corollary in the fixed design setting.

**Corollary 3** (Self-normalized bound in fixed design setting). *For an arbitrary probability measure $\mathfrak{m}$ on $S$ and an arbitrary sequence $\{x^{(j)}\}_{j \in \mathbb{N}} \in \mathcal{X}^{\mathbb{N}}$, assume that $y^{(j)}$ is sampled from $F(x^{(j)}, \cdot)$ independently for each $j \in \mathbb{N}$ with $F$ defined in (1). For any $\lambda > 0$ and $\delta \in (0, 1)$, with probability at least $1 - \delta$, the estimator defined in (2) satisfies (6) for all $n \in \mathbb{N}$.*

*If $U_N$ is positive definite for some fixed $N \in \mathbb{N}$, then for any $\delta \in (0, 1)$ and $n \geqslant N$, with probability at least $1 - \delta$, the estimator defined in (2) with $\lambda = 0$ satisfies (7).*

The proof of Corollary 3 is inline with those of Theorem 1 and Proposition 2.

Furthermore, based on Theorem 1 and Proposition 2, we prove self-normalized upper bounds on the estimation error under Scheme II, which corresponds to the random design setting in linear regression. For any probability measure $\mathfrak{m}$ on $S \subseteq \mathbb{R}$, define $\Sigma^{(j)} := \mathbb{E}_{x^{(j)} \sim P_X^{(j)}}\left[\int_S \Phi_j \Phi_j^\top d\mathfrak{m}\right]$ and $\Sigma_n := \sum_{j=1}^n \Sigma^{(j)}$ for $j, n \in \mathbb{N}$.

**Theorem 4** (Self-normalized bound in random design setting). *Assume $\mathfrak{m}$ is a probability measure on $S$, $\{(x^{(j)}, y^{(j)})\}_{j \in \mathbb{N}}$ is sampled according to Scheme II with $F$ defined in (1), and $\mu_{\min}\big(\Sigma^{(j)}\big) \geqslant \sigma_{\min}$ for some constant $\sigma_{\min} > 0$ and all $j \in \mathbb{N}$. For any $\delta \in (0, 1/2)$ and $n \geqslant \frac{32d^2}{\sigma_{\min}^2}\log(\frac{d}{\delta})$, with probability at least $1 - 2\delta$, the estimator in (2) with $\lambda = 0$ satisfies*

$$\|\widehat{\theta} - \theta_*\|_{\Sigma_n} \leqslant 2\varepsilon(n, d, \delta, \sigma_{\min}). \tag{8}$$

Moreover, for regularized estimators, we have the following result.

**Proposition 5** (Self-normalized bound in random design setting for regularized estimator). *Under the same assumptions as Theorem 4, for any $\lambda > 0$, $\delta \in (0, 1/2)$, and $n \geqslant \frac{32d^2}{\sigma_{\min}^2}\log\left(\frac{d}{\delta}\right)$, with probability at least $1 - 2\delta$, the estimator defined in (2) satisfies*

$$\|\widehat{\theta}_\lambda - \theta_*\|_{\Sigma_n} \leqslant \sqrt{2}\varepsilon_\lambda(n, d, \delta). \tag{9}$$

The proofs of Theorem 4 and Proposition 5 are given in Appendix B.2. As before, they convey that in the random design setting, the self-normalized errors $\|\widehat{\theta}_\lambda - \theta_*\|_{\Sigma_n}$ and $\|\widehat{\theta} - \theta_*\|_{\Sigma_n}$ scale as $\widetilde{O}(\sqrt{d})$ with high probability in the $\ell^2$-regularized and unregularized cases. Moreover, we once again note that Theorem 4 and Proposition 5 imply upper bounds on the (un-normalized) error $\|\widehat{\theta}_\lambda - \theta_*\|$. For example, since $\sigma_{\min}$ is a positive constant, (8) implies that $\|\widehat{\theta} - \theta_*\| \leqslant 2\mu_{\min}(\Sigma_n)^{-1/2}\varepsilon(n, d, \delta, \sigma_{\min}) = \widetilde{O}\big(\sqrt{d/n}\big)$ with high probability since $\mu_{\min}(\Sigma_n) \geqslant n\sigma_{\min}$ by Weyl's inequality (Weyl, 1912). Moreover, it is not hard to show that for general $\Sigma_n$ and $\lambda > 0$, (9) can be generalized to $\|\widehat{\theta}_\lambda - \theta_*\|_{\Sigma_n(\lambda)} = \widetilde{O}(\sqrt{d})$ which again implies that $\|\widehat{\theta}_\lambda - \theta_*\| = \widetilde{O}(\min\{1, \sqrt{d/(\mu_{\min}(\Sigma_n) + 1)}\})$.

The main idea in the proofs of Theorem 4 and Proposition 5 is to establish a high probability lower bound on $\mu_{\min}(\Delta_n)$, where $\Delta_n := \Sigma_n^{-\frac{1}{2}}(U_n - \Sigma_n)\Sigma_n^{-\frac{1}{2}}$. This can be achieved using the *matrix Hoeffding's inequality* (Tropp, 2012, Theorem 1.3). Then, we show that for any $\lambda \geqslant 0$, $\|\widehat{\theta}_\lambda - \theta_*\|_{\Sigma_n} \leqslant (1 + \mu_{\min}(\Delta_n))^{-1/2}\|\widehat{\theta}_\lambda - \theta_*\|_{U_n(\lambda)}$. For Theorem 4, we prove that $\mu_{\min}(U_n) \geqslant \mu_{\min}(\Sigma_n)(\mu_{\min}(\Delta_n) + 1)$. Then, we can lower bound $\mu_{\min}(U_n)$ in (7) by a multiple of $\mu_{\min}(\Sigma_n)$ with high probability. Thus, (8) follows from (7) and the high probability lower bound on $\mu_{\min}(\Delta_n)$. For Proposition 5, (9) follows from (6) and the high probability lower bound on $\mu_{\min}(\Delta_n)$.

We briefly compare our results in this section with related results in the literature. In the (canonical, finite dimensional) adversarial linear regression setting, Abbasi-Yadkori et al. (2011a) and Zhou et al. (2021) show an $\widetilde{O}(\sqrt{d})$ upper bound for the self-normalized error of the ridge least-squares estimator. Specifically, the upper bound in Hsu et al. (2012b) is $\widetilde{O}(R\sqrt{d})$ for the case where the noise term is $R$-sub-Gaussian and the upper bound in Zhou et al. (2021) is $\widetilde{O}(\sigma\sqrt{d} + R)$ for the case where the noise term is bounded by $R$ with variance bounded by $\sigma^2$. The functional regression upper bound in (6) aligns precisely with this scaling (neglecting sub-dominant factors) with respect to $d$ and $n$. Moreover, the upper bounds of Abbasi-Yadkori et al. (2011a) and Zhou et al. (2021) are susceptible to the magnitudes of the responses, as evidenced by their multiplicative constants of $d$. In contrast, the multiplicative constant in our upper bound is 1, ensuring that our upper bound remains independent of response scales. This independence constitutes a notable advantage, distinguishing our linear model from those explored in previous works. In the (canonical, finite dimensional) random design linear regression setting, Hsu et al. (2012b) show $\widetilde{O}(\sqrt{d})$ upper bounds for the self-normalized error of the unregularized least-squares estimator under some conditions on the distribution of covariates. The upper bound in (8) for the unregularized case also matches this scaling (neglecting sub-dominant factors). Nevertheless, it's crucial to acknowledge that our linear model (1) is characterized by a unique complexity. Unlike the canonical linear regression framework, where the features are finite-dimensional vectors, and the response is a scalar, both features and response are functions in our model. Consequently, the theoretical results of Abbasi-Yadkori et al. (2011a), Zhou et al. (2021), and Hsu et al. (2012b) are not applicable to our estimators. This intricacy introduces numerous analytical challenges, setting it apart from the conventional linear regression paradigm. Furthermore, in our infinite dimensional model (18) studied in latter chapters, we elevate the parameter from a finite-dimensional vector to a function (infinite dimensional vector), ushering in even more formidable complexities and challenges during the analysis.

Finally, we note that an upper bound on $\|\widehat{\theta}_\lambda - \theta_*\|$ immediately implies an upper bound on the KS distance between our estimated CDF and the true one. Let $\widehat{F}_\lambda(x, \cdot) := \widehat{\theta}_\lambda^\top \Phi(x, \cdot)$ denote the estimated CDF for any $x \in \mathcal{X}$. Then, under the linear model (1), we have

$$\sup_{x \in \mathcal{X}} \mathrm{KS}(\widehat{F}_\lambda(x, \cdot), F(x, \cdot)) = \sup_{x \in \mathcal{X}, t \in S} |(\widetilde{\theta}_\lambda - \theta_*)^\top \Phi(x, t)| \leqslant \|\widetilde{\theta}_\lambda - \theta_*\| \sup_{x \in \mathcal{X}, t \in S} \|\Phi(x, t)\|$$
$$\leqslant \sqrt{d}\|\widehat{\theta}_\lambda - \theta_*\|,$$

where we use the Cauchy-Schwarz inequality and the fact that $\sup_{x \in \mathcal{X}, t \in S} \|\Phi(x, t)\| \leqslant \sqrt{d}$. Since $\|\widehat{\theta}_\lambda - \theta_*\| = \widetilde{O}\left(\min\{1, \sqrt{d/(1 + \mu_{\min}(U_n))}\}\right)$ (see discussion below Proposition 2 and 5) and $\widehat{F}_\lambda, F \in [0, 1]$, we have $\sup_{x \in \mathcal{X}} \mathrm{KS}(\widehat{F}_\lambda(x, \cdot), F(x, \cdot)) = \widetilde{O}\left(\min\{1, d/\sqrt{(1 + \mu_{\min}(U_n))}\}\right)$. It is worth mentioning that the above upper bound on the estimation error in KS distance may not be sharp because we focus on a tight analysis of the estimation of $\theta_*$ instead of $F(x, \cdot)$ for some $x \in \mathcal{X}$. Nevertheless, in Appendix A, we show that when $\mu_{\min}(U_n) = 0$ ($\mu_{\min}(\Sigma_n) = 0$), the minimax risk in terms of the uniform KS distance for the estimation of $F$ is lower bounded by $\Omega(1)$ for the adversarial (random) setting.

### 3.3 Burn-in-time-free upper bound

Note that the theoretical guarantees in Theorem 4 and Proposition 5 require a burn-in time of the sample size $n$: $n \geqslant \frac{32d^2}{\sigma_{\min}^2}\log(\frac{d}{\delta})$. Motivated by Pires & Szepesvári (2012), we propose a new estimator $\widecheck{\theta}_\lambda$ in (10) to eliminate the burn-in time of $n$:

$$\widecheck{\theta}_\lambda \in \arg\min_{\theta \in \mathbb{R}^d} \left(\|U_n(\lambda)\theta - u_n\| + \Delta_n^U(\delta)\|\theta\|\right), \tag{10}$$

where $\lambda \geqslant 0$, $\delta \in (0,1)$, $u_n := \sum_{j=1}^n \int_S \mathrm{I}_{y^{(j)}} \Phi_j d\mathfrak{m}$, and $\Delta_n^U(\delta)$ is a positive number such that $\Delta_n^U(\delta) \geqslant \|U_n - \Sigma_n\|$ with probability at least $1 - \delta$. For notatoinal convenience, we use $\breve{\theta}$ to denote $\breve{\theta}_0$. To calculate $\breve{\theta}_\lambda$ in (10), it is necessary to first choose $\Delta_n^U(\delta)$ for which we prove a lower bound in the following lemma.

**Lemma 6.** *Assume $\mathfrak{m}$ is a probability measure on $S$ and $\{(x^{(j)}, y^{(j)})\}_{j \in \mathbb{N}}$ is sampled according to Scheme II with $F$ defined in (1). For any $\delta \in (0,1)$ and $n \in \mathbb{N}$, any $\Delta_n^U(\delta) \geqslant d\sqrt{8n \log(d/\delta)}$ satisfies $\Delta_n^U(\delta) \geqslant \Delta_n^U$ with probability at least $1 - \delta$.*

The proof of Lemma 6 follows from the *matrix Hoeffding's inequality* (Tropp, 2012, Theorem 1.3) and the boundedness of CDFs, and is provided in Appendix E. Then, we show the following upper bound on the estimation error of $\breve{\theta}_\lambda$.

**Theorem 7** (Self-normalized bound in random setting without burn-in time)**.** *Under the same assumptions as Lemma 6, for any $\delta \in (0, 1/2)$ and $n \in \mathbb{N}$, if $\mu_{\min}(\Sigma_n) > 0$, then, with probability at least $1 - 2\delta$, the estimator defined in (10) with $\lambda = 0$ satisfies*

$$\|\breve{\theta} - \theta_*\| \leqslant \frac{1}{\mu_{\min}(\Sigma_n)}\left[2d\sqrt{8n\log(d/\delta)}\|\theta_*\| + 2\left(\sqrt{nd} + \sqrt{8nd\log(1/\delta)} + \frac{4}{3}\sqrt{d}\log(1/\delta)\right)\right]. \tag{11}$$

The proof of Theorem 7 is provided in Appendix B.3. It conveys that for any $n \in \mathbb{N}$, as long as $\mu(\Sigma_n) > 0$, $\|\breve{\theta} - \theta_*\| \leqslant \widetilde{O}\left(\frac{d\sqrt{n}}{\mu_{\min}(\Sigma_n)}\right)$ holds with high probability. Under the assumption that $\mu_{\min}(\Sigma^{(j)}) \geqslant \sigma_{\min}$ for any $j \in \mathbb{N}$ as in Theorem 4 and Proposition 5, we have that $\|\breve{\theta} - \theta_*\| \leqslant \widetilde{O}(d/\sqrt{n})$ with high probability for any $n \in \mathbb{N}$. Compared with the $\widetilde{O}(\sqrt{d/n})$ upper bound of the estimation error of $\widehat{\theta}$ in Theorem 4, $\breve{\theta}$ suffers a larger error rate wrt the dimension $d$ in order to eliminate the burn-in time of the sample size $n$. Thus, $\breve{\theta}$ is more applicable to the estimation of $\theta_*$ for small sample size and small dimension. However, it is worth mentioning that since the estimation errors of proper estimators which are contained in the probability simplex are always bounded by 2, our upper bound in (11) is only non-trivial for the projection of $\breve{\theta}$ to the probability simplex when $n = \Omega(d^2 \log(d/\delta)/\sigma_{\min}^2)$ which aligns with the scale of the burn-in time of $\widehat{\theta}$. Thus, the estimator (10) only eliminates the burn-in time among improper estimators.

The proof of Theorem 7 builds on the upper bound shown in Pires & Szepesvári (2012) for the estimator that minimizes the unsquared penalized loss as in (10). By Pires & Szepesvári (2012, Theorem 3.4), we have that with probability at least $1 - \delta$,

$$\|\Sigma_n(\lambda)\breve{\theta}_\lambda - \Sigma_n\theta_*\| \leqslant (\lambda + 2\Delta_n^U(\delta))\|\theta_*\| + 2\|u_n - \mathbb{E}[u_n]\|.$$

Then, we can bound $\|u_n - \mathbb{E}[u_n]\|$ with high probability by the vector Bernstein inequality (Hsu et al., 2012a, Proposition 1.2). By setting $\lambda = 0$ and $\Delta_n^U(\delta) = d\sqrt{8n\log(d/\delta)}$ as is guaranteed by Lemma 6, we obtain (11) after some derivation.

## 4 Minimax lower bounds

To show that our estimator (2) is minimax optimal, we prove information theoretic lower bounds on the $\ell^2$-norm of the estimation error for any estimator. Recall that for a distribution family $\mathcal{Q}$ and (parameter) function $\xi : \mathcal{Q} \to \mathbb{R}^d$, the minimax $\ell^2$-risk is defined as,

$$\mathfrak{R}(\xi(\mathcal{Q})) := \inf_{\hat{\xi}} \sup_{Q \in \mathcal{Q}} \mathbb{E}_{z \sim Q}[\|\hat{\xi}(z) - \xi(Q)\|], \tag{12}$$

where the infimum is over all (possibly randomized) estimators $\hat{\xi}$ of $\xi$ based on a sample $z$, and the supremum is over all distributions in the family $\mathcal{Q}$. To specialize this definition for our problem, for any $x \in \mathcal{X}$ and $\theta \in \mathbb{R}^d$, let $P_{Y|x;\theta}^\Phi$ denote the probability measure defined by the CDF $\theta^\top \Phi(x, \cdot)$. Moreover, for any sequence $x^{1:n} := (x^{(1)}, \ldots, x^{(n)}) \in \mathcal{X}^n$, define the collection of product measures, $\mathcal{P}_{x^{1:n}}^d := \left\{\otimes_{j=1}^n P_{Y|x^{(j)};\theta}^\Phi : \theta \in \Delta^{d-1}, \Phi \in \mathfrak{B}_d\right\}$, where

$$\mathcal{B}_d := \{[\phi_1, \ldots, \phi_d]^\top : \phi_i : \mathcal{X} \times \mathbb{R} \to [0,1] \text{ is measurable and } \phi_i(x, \cdot) \text{ is a CDF on } \mathbb{R}, \forall i \in [d]\}.$$

For any distribution $P \in \mathcal{P}_{x^{1:n}}^d$, let $\theta(P)$ be a parameter in $\Delta^{d-1}$ such that $P = \otimes_{j=1}^n P_{Y|x^{(j)};\theta}^\Phi$. Then, we have the following theorem in the adversarial setting.

**Theorem 8** (Information theoretic lower bound in adversarial setting). *For any $d \geqslant 2$ and any sequence $x^{1:n} = (x^{(1)}, \ldots, x^{(n)}) \in \mathcal{X}^n$, we have*

$$\mathfrak{R}(\theta(\mathcal{P}_{x^{1:n}}^d)) = \Omega\left(\min\{1, \sqrt{d/(1 + \mu_{\min}(U_n))}\}\right). \tag{13}$$

The proof uses *Fano's method* (Fano, 1961) and is given in Appendix C.1. Note that strictly speaking, the above theorem is written for the fixed design setting. However, a lower bound in the fixed design setting also implies the same lower bound in adversarial setting. Furthermore, by our discussion below Theorem 1, (6) implies that in the adversarial setting,

$$\mathbb{P}\left[\|\widehat{\theta}_\lambda - \theta_*\|^2 \geqslant \frac{C_1 d \log(n) + C_2 + C_3 r}{1 + \mu_{\min}(U_n)}\right] \leqslant e^{-r}$$

for $r > 0$ and some constants $C_1$, $C_2$, and $C_3$, which immediately implies that $\mathbb{E}[\|\widehat{\theta}_\lambda - \theta_*\|] = \widetilde{O}(\sqrt{d/(1 + \mu_{\min}(U_n))})$ and $\mathbb{E}[\|\widetilde{\theta}_\lambda - \theta_*\|] = \widetilde{O}(\min\{1, \sqrt{d/(1 + \mu_{\min}(U_n))}\})$. Thus, our estimator $\widetilde{\theta}_\lambda$ is minimax optimal. When $\mu_{\min}(U_n) = \Theta(n)$, the optimal rate is $\widetilde{\Theta}(\sqrt{d/n})$ in the adversarial setting.

In the proof of Theorem 8, we construct a family of $\Omega(a/\sqrt{d})$-packing subsets of $\Delta^{d-1}$ for $a \in (0,1)$ under $\ell^2$-distance. We then show that when $\phi_1, \ldots, \phi_d$ are the CDFs of $d$ Bernoulli distributions, for any $\theta^{(1)} \neq \theta^{(2)}$ in such a packing subset, the Kullback-Leibler (KL) divergence (see definition in Appendix C.1) satisfies

$$D(P_{Y|x^{(j)};\theta^{(1)}} \| P_{Y|x^{(j)};\theta^{(2)}}) = O(a^2(1 + \mu_{\min}(U_n))/d)$$

for any $j \in [n]$. Since the above family of Bernoulli distributions is a subset of $\mathcal{P}_{x^{1:n}}^d$, we are able to show that $\mathfrak{R}(\theta(\mathcal{P}_{x^{1:n}}^d)) = \Omega(\sqrt{d/(1 + \mu_{\min}(U_n))})$ using Fano's method and the aforementioned bound on KL divergence.

Next, to analyze minimax $\ell^2$-risk under the random setting, let $\mathcal{D}_\mathcal{X}$ denote the set of all probability distributions on $\mathcal{X}$. For any $P_X \in \mathcal{D}_\mathcal{X}$, let $P_X P_{Y|X;\theta}^\Phi$ denote the joint distribution of $(X, Y)$ such that the marginal distribution of $X$ is $P_X$ and the conditional distribution of $Y$ given $X = x$ is $P_{Y|x;\theta}^\Phi$. Define the distribution family

$$\mathcal{P}_n^d := \left\{ \otimes_{j=1}^n P_X^{(j)} P_{Y|X;\theta}^\Phi : \theta \in \mathbb{R}^d, \ \Phi \in \mathcal{B}_d, \ P_X^{(j)} \in \mathcal{D}_\mathcal{X} \right\},$$

and for any $P \in \mathcal{P}_n^d$, let $\theta(P)$ denote the parameter in $\Delta^{d-1}$ such that $P = \otimes_{j=1}^n P_X^{(j)} P_{Y|X;\theta}^\Phi$. Clearly, for any $x^{1:n} \in \mathcal{X}^n$, we have $\{\otimes_{j=1}^n \delta_{x^{(j)}} P_{Y|X;\theta} : \theta \in \Delta^{d-1}\} \subseteq \mathcal{P}_n^d$. Thus, each $\mathcal{P}_{x^{1:n}}^d$ is a collection of marginal distributions of elements belonging to such subsets of $\mathcal{P}_n^d$. Then, by the definition of minimax $\ell^2$-risk, Theorem 8 immediately implies the following corollary.

**Corollary 9** (Information theoretic lower bound in random setting). *For any $d \geqslant 2$,*

$$\mathfrak{R}(\theta(\mathcal{P}_n^d)) = \Omega\left(\min\{1, \sqrt{d/(1 + \mu_{\min}(\Sigma_n))}\}\right) \tag{14}$$

The proof is given in Appendix C.2. By the discussion below Proposition 5, our estimator $\widetilde{\theta}_\lambda$ ($\lambda > 0$) is minimax optimal. When $\mu_{\min}(\Sigma_n) = \Theta(n)$ as in Theorem 4 and Corollary 5, the lower bound on the Euclidean norm of the estimation error is also $\Omega(\sqrt{d/n})$ in random setting. Following the discussion below Theorem 4, (8) implies that in random setting, $\mathbb{P}[\|\widehat{\theta} - \theta_*\| \geqslant C_1\sqrt{d/n} + C_2\sqrt{rd/n} + C_3 r\sqrt{d}/n] \leqslant e^{-r}$ for $r > 0$ and constants $C_1$, $C_2$, and $C_3$, which immediately implies that $\mathbb{E}[\|\widehat{\theta} - \theta_*\|] = \widetilde{O}(\sqrt{d/n})$. Thus, the estimator (2) is minimax optimal with rate $\widetilde{\Theta}(\sqrt{d/n})$ in random setting when $\mu_{\min}(\Sigma_n) = \Theta(n)$.

## 5 Mismatched model

In general, a mismatch may exist between the true target function and our linear model (1) with basis $\Phi$. So, in analogy with canonical linear regression where additive Gaussian random variables are used to model the

error term (Montgomery et al., 2021), we consider the following *mismatched model*:

$$F(x,t) = \theta_*^\top \Phi(x,t) + e(x,t), \quad \forall\ x \in \mathcal{X}, t \in \mathbb{R}, \tag{15}$$

where an additive error function depending on the context is included to model the mismatch in (1). Note that in (15), each $F(x,\cdot)$ is a CDF and $e : \mathcal{X} \times S \to [-1, 1]$ is a measurable function. One equivalent interpretation of (15) is as follows. Suppose that their exists another contextual CDF function $\phi_e$ such that $F(x,\cdot)$ is a mixture of the linear model $\theta_*^\top \Phi(x,\cdot)$ and the new feature function $\phi_e(x,\cdot)$, i.e., for some $q \in [0, 1]$,

$$F(x,t) = (1-q)\theta_*^\top \Phi(x,t) + q\phi_e(x,t) = \theta_*^\top \Phi(x,t) + q\left(\phi_e(x,t) - \theta_*^\top \Phi(x,t)\right), \ \forall\ x \in \mathcal{X}, t \in \mathbb{R}.$$

Then, we naturally obtain an additive error function $e(x,t) = q\left(\phi_e(x,t) - \theta_*^\top \Phi(x,t)\right)$.

Given a sample $\{(x^{(j)}, y^{(j)})\}_{j \in [n]}$ generated using the mismatched model (15), let $e_j(t)$ denote $e(x^{(j)}, t)$ for $j \in [n]$. Moreover, define $E_n := \sum_{j=1}^n \int_S e_j \Phi_j d\mathfrak{m}$ and $B_n := \mathbb{E}[E_n] = \sum_{j=1}^n \mathbb{E}\left[\int_S e_j \Phi_j d\mathfrak{m}\right]$. Then, we have the following theoretical guarantees for the task of estimating $\theta_*$ using the estimator in (2) in the adversarial and random settings.

**Theorem 10** (Self-normalized bound in mismatched adversarial setting). *Assume $\mathfrak{m}$ is a probability measure on $S$ and $\{(x^{(j)}, y^{(j)})\}_{j \in \mathbb{N}}$ is sampled according to Scheme I with $F$ defined in (15). For any $\lambda > 0$ and $\delta \in (0, 1)$, with probability at least $1 - \delta$, the estimator defined in (2) satisfies that for all $n \in \mathbb{N}$,*

$$\|\widehat{\theta}_\lambda - \theta_*\|_{U_n(\lambda)} \leqslant \varepsilon_\lambda(n, d, \delta) + \|E_n\|/\sqrt{\lambda}. \tag{16}$$

The proof of Theorem 10 follows the same approach as the proof of Theorem 1, and it is provided in Appendix I.1. Furthermore, Theorem 10 implies Corollary 11 for the mismatched random setting.

**Corollary 11** (Self-normalized bound in mismatched random setting). *Assume $\mathfrak{m}$ is a probability measure on $S$, $\{(x^{(j)}, y^{(j)})\}_{j \in \mathbb{N}}$ is sampled according to Scheme II with $F$ defined in (15), and $\mu_{\min}(\Sigma^{(j)}) \geqslant \sigma_{\min}$ for some $\sigma_{\min} > 0$ and all $j \in \mathbb{N}$. For any $\lambda > 0$, $\delta \in (0, 1/2)$, and $n \geqslant \frac{32 d^2}{\sigma_{\min}^2} \log\left(\frac{d}{\delta}\right)$, with probability at least $1 - 2\delta$, the estimator defined in (2) satisfies*

$$\|\widehat{\theta}_\lambda - \theta_*\|_{\Sigma_n} \leqslant \sqrt{2}\varepsilon_\lambda(n, d, \delta) + \sqrt{2/\lambda}\|B_n\|. \tag{17}$$

The proof of Corollary 11 is given in Appendix I.2. It follows from the proofs of Theorem 10 and Proposition 5.

In the adversarial setting, comparing (16) in Theorem 10 with (6) in Theorem 1, we see that the effect of the additive error in the mismatched model is captured by the additional $\|E_n\|/\sqrt{\lambda}$ term in our self-normalized error upper bound. Similarly, in the random setting, comparing (17) in Corollary 11 with (9), we again see that the effect of the additive error is captured by the additional $\sqrt{2/\lambda}\|B_n\|$ term in the self-normalized upper bound.

# 6 Infinite dimensional model

So far, we have been assuming finite-dimensional models where the number of base contextual CDFs $\phi_i$'s per sample is finite. It is natural to consider generalizing the linear model to be infinite-dimensional and estimating an infinite dimensional parameter $\theta_*$ which shall be considered as a function on the "index" space of the base functions. In Section 6.1, we formally introduce the infinite-dimensional linear model. We present necessary definitions and technical facts for the statement of the estimator and theorem in Section 6.2. We extend the estimator $\widehat{\theta}_\lambda$ in (2) with properly chosen regularization and provide a high probability upper bound on the estimation error of the generalized estimator in Section 6.3.

## 6.1 Formal model

First, we introduce the infinite dimensional index space $\Omega$ and the generalized basis function $\Phi$. Assume that $(\Omega, \mathcal{F}_\Omega, \mathfrak{n})$ is a measure space with $\mathfrak{n}(\Omega) < \infty$ and $\Phi : \mathcal{X} \times \Omega \times \mathbb{R} \to [0, 1]$, $(x, \omega, t) \mapsto \Phi(x, \omega, t)$ is a

$(\mathcal{B}(\mathcal{X}) \otimes \mathcal{F}_\Omega \otimes \mathcal{B}(\mathbb{R}))/\mathcal{B}([0,1])$-measurable function (see Appendix B for the explanations of notation) such that for any $x \in \mathcal{X}$ and $\mathfrak{n}$-a.e. $\omega \in \Omega$, $\Phi(x, \omega, \cdot)$ is the CDF of some $\mathbb{R}$-valued random variable with its range contained in some Borel set $S \subseteq \mathbb{R}$. Define the following mapping,

$$\langle \cdot, \cdot \rangle : \ \mathcal{L}^2(\Omega, \mathfrak{n}) \times \mathcal{L}^2(\Omega, \mathfrak{n}) \to \mathbb{R}, \ (f, g) \mapsto \int_\Omega f g d\mathfrak{n}.$$

Then, $\langle \cdot, \cdot \rangle$ is an inner product on $\mathcal{L}^2(\Omega, \mathfrak{n})$ and $(\mathcal{L}^2(\Omega, \mathfrak{n}), \langle \cdot, \cdot \rangle)$ is a Hilbert space. Let $\| \cdot \|$ denote the norm by induced $\langle \cdot, \cdot \rangle$ on $\mathcal{L}^2(\Omega, \mathfrak{n})$. Assume that $(\mathcal{L}^2(\Omega, \mathfrak{n}), \langle \cdot, \cdot \rangle)$ is separable. Then, there exists a countable orthonormal basis on $(\mathcal{L}^2(\Omega, \mathfrak{n}), \langle \cdot, \cdot \rangle)$. For notational convenience, we write $\mathcal{L}^2(\Omega, \mathfrak{n})$ to represent the Hilbert space $(\mathcal{L}^2(\Omega, \mathfrak{n}), \langle \cdot, \cdot \rangle)$. Let $\boldsymbol{e} = \{e_i\}_{i=1}^\infty$ be an arbitrary countable orthonormal basis of $\mathcal{L}^2(\Omega, \mathfrak{n})$ and $\boldsymbol{\sigma} = \{\sigma_i\}_{i \in \mathbb{N}}$ be an arbitrary real sequence such that $\sum_{i=1}^\infty |\sigma_i| < \infty$. Assume that there exists some *unknown* $\theta_* \in \mathcal{H}_{\boldsymbol{\sigma}, \boldsymbol{e}}$ such that $\theta_* \geqslant 0$ $\mathfrak{n}$-a.e., $\int_\Omega \theta_* \mathfrak{n} = 1$, and the target function $F$ satisfies the following model

$$F(x, t) = \langle \theta_*(\cdot), \Phi(x, \cdot, t) \rangle, \quad \forall \ x \in \mathcal{X}, \ t \in \mathbb{R}. \tag{18}$$

## 6.2 Technical preliminaries

The proofs of the theoretical results in this section are provided in Appendix F. Given the sample $\{(x^{(j)}, y^{(j)})\}_{j \in \mathbb{N}} \subseteq \mathcal{X} \times \mathbb{R}$, define the function $\Phi_j : \ \Omega \times \mathbb{R} \to [0, 1]$, $(\omega, t) \mapsto \Phi(x_j, \omega, t)$ for any $j \in \mathbb{N}$. Since $\mathfrak{n}(\Omega) < \infty$ and $|\Phi(x, \omega, t)| \leqslant 1$ for any $x \in \mathcal{X}$, $\mathfrak{n}$-a.e. $\omega \in \Omega$, and any $t \in \mathbb{R}$, we have $\Phi_j \in \mathcal{L}^2(\Omega, \mathfrak{n})$ for any $j \in \mathbb{N}$. Then, for any $j \in \mathbb{N}$, we define,

$$\Psi_j : \mathcal{L}^2(\Omega, \mathfrak{n}) \times S \to \mathbb{R}, \ (\theta, t) \mapsto \langle \theta(\cdot), \Phi_j(\cdot, t) \rangle.$$

Then, by Holder's inequality, for any $j \in \mathbb{N}$ and $\theta \in \mathcal{L}^2(\Omega, \mathfrak{n})$, we have $\sup_{t \in \mathbb{R}} |\Psi_j(\theta, t)| \leqslant \mathfrak{n}(\Omega) \int_\Omega |\theta|^2 d\mathfrak{n} < \infty$. It follows that $\Psi_j(\theta, \cdot) \in \mathcal{L}^2(S, \mathfrak{m})$. Moreover, we have that for any $n \in \mathbb{N}$, any $\theta \in \mathcal{L}^2(\Omega, \mathfrak{n})$, and $\mathfrak{n}$-a.e. $\omega \in \Omega$,

$$\left| \sum_{j=1}^n \int_S \Psi_j(\theta, t) \Phi_j(\omega, t) \mathfrak{m}(dt) \right| \leqslant \sum_{j=1}^n \int_S |\Psi_j(\theta, t) \Phi_j(\omega, t)| \, \mathfrak{m}(dt) \leqslant n\mathfrak{n}(\Omega) \int_\Omega |\theta|^2 d\mathfrak{n}.$$

Since $\mathfrak{n}(\Omega) < \infty$, it follows that the function $\omega \mapsto \sum_{j=1}^n \int_S \Psi_j(\theta, t) \Phi_j(\omega, t) \mathfrak{m}(dt)$ is in $\mathcal{L}^2(\Omega, \mathfrak{n})$. Thus, for any $n \in \mathbb{N}$, we can define an operator $U_n : \mathcal{L}^2(\Omega, \mathfrak{n}) \to \mathcal{L}^2(\Omega, \mathfrak{n})$ by

$$(U_n \theta)(\omega) := \sum_{j=1}^n \int_S \Psi_j(\theta, t) \Phi_j(\omega, t) \mathfrak{m}(dt) = \sum_{j=1}^n \int_S \langle \theta(\cdot), \Phi_j(\cdot, t) \rangle \Phi_j(\omega, t) \mathfrak{m}(dt) \tag{19}$$

for any $\theta \in \mathcal{L}^2(\Omega, \mathfrak{n})$. We show the following properties of $U_n$.

**Lemma 12.** *For any $n \in \mathbb{N}$, $U_n$ is a self-adjoint positive Hilbert-Schmidt integral operator with $\|U_n\| \leqslant n\mathfrak{n}(\Omega)$. Thus, it is also a compact operator.*

Now, we assume that $U_n$ satisfies Assumption 13 for some $n \in \mathbb{N}$.

**Assumption 13.** *Assume that $e_i$ is an eigenfunction of $U_n$ with the corresponding eigenvalue denoted with $\lambda_i$ for any $i \in \mathbb{N}$.*

Under the Assumption 13 on $U_n$, we can conclude from Lemma 12 that:

**Corollary 14.** *Assume that $U_n$ satisfies Assumption 13 for some $n \in \mathbb{N}$. Then, we have $0 \leqslant \lambda_i \leqslant n\mathfrak{n}(\Omega)$ for any $i \in \mathbb{N}$ and $\lambda_i \to 0$.*

Define the set $\mathcal{L}^2_{\boldsymbol{\sigma}}(\Omega, \mathfrak{n}) := \left\{ \theta \in \mathcal{L}^2(\Omega, \mathfrak{n}) : \sum_{i=1}^\infty \frac{|\langle e_i, \theta \rangle|^2}{\sigma_i^4} < \infty \right\}$. Then, we have that

**Lemma 15.** *For any $\boldsymbol{\sigma} = \{\sigma_i\}_{i \in \mathbb{N}} \subseteq \mathbb{R}$ satisfying $\sum_{i=1}^\infty |\sigma_i| < \infty$, $\mathcal{L}^2_{\boldsymbol{\sigma}}(\Omega, \mathfrak{n})$ is a linear subspace of $\mathcal{L}^2(\Omega, \mathfrak{n})$.*

For any $\theta \in \mathcal{L}^2_{\boldsymbol{\sigma}}(\Omega, \mathfrak{n})$, we have

$$\sum_{i=1}^\infty \left| \lambda_i + \frac{1}{\sigma_i^2} \right|^2 |\langle e_i, \theta \rangle|^2 \leqslant \sum_{i=1}^\infty 2\lambda_i^2 |\langle e_i, \theta \rangle|^2 + \sum_{i=1}^\infty \frac{2}{\sigma_i^4} |\langle e_i, \theta \rangle|^2 \leqslant 2\|U_n\theta\|^2 + 2 \sum_{i=1}^\infty \frac{|\langle e_i, \theta \rangle|^2}{\sigma_i^4} < \infty,$$

which implies that $\{\sum_{i=1}^{m}(\lambda_i + \frac{1}{\sigma_i^2})\langle e_i, \theta\rangle e_i\}_{m\in\mathbb{N}}$ is a Cauchy sequence and thus converges in $\mathcal{L}^2(\Omega, \mathfrak{n})$ to $\sum_{i=1}^{\infty}(\lambda_i + \frac{1}{\sigma_i^2})\langle e_i, \theta\rangle e_i \in \mathcal{L}^2(\Omega, \mathfrak{n})$. Therefore, we can define the operator $U_{n,\boldsymbol{\sigma}} : \mathcal{L}_{\boldsymbol{\sigma}}^2(\Omega, \mathfrak{n}) \to \mathcal{L}^2(\Omega, \mathfrak{n})$, $\theta \mapsto \sum_{i=1}^{\infty}\left(\lambda_i + \frac{1}{\sigma_i^2}\right)\langle e_i, \theta\rangle e_i$ for which we show the following lemma.

**Lemma 16.** $U_{n,\boldsymbol{\sigma}}$ *is bijective linear operator from* $\mathcal{L}_{\boldsymbol{\sigma}}^2(\Omega, \mathfrak{n})$ *onto* $\mathcal{L}^2(\Omega, \mathfrak{n})$. $U_{n,\boldsymbol{\sigma}}^{-1}$ *is a bounded linear operator on* $\mathcal{L}^2(\Omega, \mathfrak{n})$ *with* $\|U_{n,\boldsymbol{\sigma}}^{-1}\| \leqslant \sup_{i\in\mathbb{N}} \sigma_i^2$ *and* $U_{n,\boldsymbol{\sigma}}^{-1}\theta = \sum_{i=1}^{\infty}\frac{\sigma_i^2\langle e_i, \theta\rangle}{1+\lambda_i\sigma_i^2} e_i$ *for any* $\theta \in \mathcal{L}^2(\Omega, \mathfrak{n})$. *Moreover,* $U_{n,\boldsymbol{\sigma}}^{-1}$ *is positive and self-adjoint.*

Consequently, we can define the following mapping

$$\|\cdot\|_{U_{n,\boldsymbol{\sigma}}} : \mathcal{L}_{\boldsymbol{\sigma}}^2(\Omega, \mathfrak{n}) \to [0, \infty), \ \theta \mapsto \sqrt{\langle \theta, U_{n,\boldsymbol{\sigma}}\theta\rangle} = \sqrt{\|\theta\|_{U_n}^2 + \sum_{i=1}^{\infty}\frac{|\langle e_i, \theta\rangle|^2}{\sigma_i^2}}, \tag{20}$$

where $\|\theta\|_{U_n} := \sqrt{\langle \theta, U_n\theta\rangle}$ for any $\theta \in \mathcal{L}^2(\Omega, \mathfrak{n})$. Define the set

$$\mathcal{H}_{\boldsymbol{\sigma},e} := \left\{\theta \in \mathcal{L}^2(\Omega) : \sum_{i=1}^{\infty} |\langle e_i, \theta\rangle|^2/\sigma_i^2 < \infty\right\}$$

and the mapping $\langle\cdot, \cdot\rangle_{\boldsymbol{\sigma},e} : \mathcal{H}_{\boldsymbol{\sigma},e} \times \mathcal{H}_{\boldsymbol{\sigma},e} \to \mathbb{R}$, $(f, g) \mapsto \sum_{i=1}^{\infty}\frac{\langle e_i, f\rangle\langle e_i, g\rangle}{\sigma_i^2}$. Similar to the proofs of Lemma 15, we can show that $\mathcal{H}_{\boldsymbol{\sigma},e}$ is a linear subspace of $\mathcal{L}^2(\Omega, \mathfrak{n})$. Moreover, $(\mathcal{H}_{\boldsymbol{\sigma},e}, \langle\cdot, \cdot\rangle_{\boldsymbol{\sigma},e})$ is also a separable Hilbert space with $\{\sigma_i e_i\}_{i\in\mathbb{N}}$ being an orthonormal basis. For notational convenience, we write $\mathcal{H}_{\boldsymbol{\sigma},e}$ to represent the Hilbert space $(\mathcal{H}_{\boldsymbol{\sigma},e}, \cdot, \langle\cdot\rangle_{\boldsymbol{\sigma},e})$ and use $\|\cdot\|_{\boldsymbol{\sigma},e}$ to denote the induced norm on $\mathcal{H}_{\boldsymbol{\sigma},e}$. Moreover, we show the following lemma.

**Lemma 17.** *For any real sequence* $\boldsymbol{\sigma} = \{\sigma_i\}_{i\in\mathbb{N}}$ *with* $\lim_{i\to\infty} \sigma_i = 0$, $\mathcal{H}_{\boldsymbol{\sigma},e} \subseteq \mathcal{L}_{\boldsymbol{\sigma}}^2(\Omega, \mathfrak{n})$.

## 6.3 Self-normalized upper bound

Since we have proved that $\Psi_j(\theta, \cdot) \in \mathcal{L}^2(S, \mathfrak{m})$ for any $j \in \mathbb{N}$ and $\theta \in \mathcal{L}^2(\Omega, \mathfrak{n})$, the following loss function is well-defined on $\mathcal{H}_{\boldsymbol{\sigma},e}$,

$$L(\theta; \boldsymbol{\sigma}) := \sum_{j=1}^{n} \|\mathrm{I}_{y^{(j)}}(\cdot) - \Psi_j(\theta, t)\|_{\mathcal{L}^2(S,\mathfrak{m})}^2 + \sum_{i=1}^{\infty}\frac{|\langle e_i, \theta\rangle|^2}{\sigma_i^2}.$$

In fact, assuming the convention that $0/0 = 0$ and $1/0 = \infty$, we can extend the domain of $L(\cdot; \boldsymbol{\sigma})$ to $\mathcal{L}^2(\Omega, \mathfrak{n})$ by extending its codomain from $[0, \infty)$ to $[0, \infty]$.

We propose to estimate $\theta_*$ by minimizing the above loss function over $\mathcal{H}_{\boldsymbol{\sigma},e}$:

$$\widehat{\theta}_{\boldsymbol{\sigma}} := \underset{\theta\in\mathcal{H}_{\boldsymbol{\sigma},e}}{\arg\min} L(\theta; \boldsymbol{\sigma}). \tag{21}$$

Since $\sum_{i=1}^{\infty}\frac{|\langle e_i, \theta\rangle|^2}{\sigma_i^2} = \infty$ for any $\theta \in \mathcal{L}^2(\Omega, \mathfrak{n})$, we also have $\widehat{\theta}_{\boldsymbol{\sigma}} = \arg\min_{\theta\in\mathcal{L}^2(\Omega,\mathfrak{n})} L(\theta; \lambda)$. We have the following formula for $\widehat{\theta}_{\boldsymbol{\sigma}}$ in (21).

**Proposition 18.** *The solution to the optimization problem* (21) *is given as the following,*

$$\widehat{\theta}_{\boldsymbol{\sigma}} = U_{n,\boldsymbol{\sigma}}^{-1}\left(\sum_{j=1}^{n} \int_S \mathrm{I}_{y^{(j)}}(t)\Phi_j(\cdot, t)\mathfrak{m}(dt)\right). \tag{22}$$

The proof of Proposition 18 is provided in Appendix G. Under the adversarial setting, we show the following upper bound for the self-normalized estimation error of $\widehat{\theta}_{\boldsymbol{\sigma}}$ in (21).

**Theorem 19** (Self-normalized bound in adversarial setting for infinite dimensional model). *Assume $\mathfrak{m}$ is a probability measure on $(S, \mathcal{B}(S))$, $\mathfrak{n}$ is a finite measure on $(\Omega, \mathcal{F}_\Omega)$, $\boldsymbol{e} = \{e_i\}_{i=1}^\infty$ is an orthonormal basis of $\mathcal{L}^2(\Omega, \mathfrak{n})$, $\boldsymbol{\sigma} = \{\sigma_i\}_{i\in\mathbb{N}}$ is a real sequence satisfying $\sum_{i=1}^\infty |\sigma_i| < \infty$, $\theta_* \in \mathcal{H}_{\boldsymbol{\sigma},\boldsymbol{e}}$ satisfies $\theta_* \geqslant 0$ $\mathfrak{n}$-a.e. and $\int_\Omega \theta_* \mathfrak{n} = 1$, and $\{(x^{(j)}, y^{(j)})\}_{j\in\mathbb{N}}$ is sampled according to Scheme I with $F$ defined in (18).*

*For any given $n \in \mathbb{N}$ and any $\delta \in (0, 1)$, if $U_n$ defined in (19) satisfies Assumption 13 and $\boldsymbol{\sigma}$ satisfies that $|\sigma_i| < \frac{1}{\sqrt{\lambda_i}}$ for any $i \in \mathbb{N}$, then, with probability at least $1 - \delta$, the estimator $\widehat{\theta}_{\boldsymbol{\sigma}}$ defined in (21) satisfies*

$$\|\widehat{\theta}_{\boldsymbol{\sigma}} - \theta_*\|_{U_n,\boldsymbol{\sigma}} \leqslant \sqrt{\left(\sum_{i=1}^\infty \log\left(1 + \lambda_i \sigma_i^2\right)\right) + 2\log\frac{1}{\delta}} + \|\theta_*\|_{\boldsymbol{\sigma},\boldsymbol{e}}. \tag{23}$$

*In particular, for any given $n \in \mathbb{N}$ and any $\delta \in (0, 1)$, if $U_n$ defined in (19) satisfies Assumption 13 and $\boldsymbol{\sigma}$ satisfies that $|\sigma_i| < \frac{1}{\sqrt{n\mathfrak{n}(\Omega)}}$ for any $i \in \mathbb{N}$, then, with probability at least $1 - \delta$, the estimator $\widehat{\theta}_{\boldsymbol{\sigma}}$ defined in (21) satisfies*

$$\|\widehat{\theta}_{\boldsymbol{\sigma}} - \theta_*\|_{U_n,\boldsymbol{\sigma}} \leqslant \sqrt{\left(\sum_{i=1}^\infty \log\left(1 + n\mathfrak{n}(\Omega)\sigma_i^2\right)\right) + 2\log\frac{1}{\delta}} + \|\theta_*\|_{\boldsymbol{\sigma},\boldsymbol{e}}. \tag{24}$$

The detailed proof of Theorem 19 is provided in Appendix D. Since $\sum_{i=1}^\infty |\sigma_i| < \infty$ and $\theta_* \in \mathcal{H}_{\boldsymbol{\sigma},\boldsymbol{e}}$, we have that $\|\theta_*\|_{\boldsymbol{\sigma},\boldsymbol{e}} < \infty$ and $\sum_{i=1}^\infty |\sigma_i|^2 < \infty$ which implies that

$$\sum_{i=1}^\infty \log\left(1 + \lambda_i \sigma_i^2\right) \leqslant \sum_{i=1}^\infty \log\left(1 + n\mathfrak{n}(\Omega)\sigma_i^2\right) < \infty.$$

Thus, the RHS terms in (23) and (24) are finite and $\widehat{\theta}_{\boldsymbol{\sigma}} - \theta_* \in \mathcal{L}^2_{\boldsymbol{\sigma}}(\Omega, \mathfrak{n})$. (24) conveys that with high probability,

$$\|\widehat{\theta}_{\boldsymbol{\sigma}} - \theta_*\|_{U_n,\boldsymbol{\sigma}} \leqslant \widetilde{O}\left(1 + \sqrt{\sum_{i=1}^\infty \log(1 + n\mathfrak{n}(\Omega)\sigma_i^2)}\right).$$

When $\Omega = [d]$ for some $d \in \mathbb{N}$ and $\mathfrak{n}$ is the counting measure on $\Omega$, (21) reduces to (2) after setting $e_i = \mathbb{1}_{\{i\}}$ and $\sigma_i = \frac{1}{\sqrt{\lambda}}$ for any $i \in [d]$ and some $\lambda > 0$. Then, by (20) and (24), we have $\|\widehat{\theta}_{\boldsymbol{\sigma}} - \theta_*\|_{U_n} \leqslant \widetilde{O}(\sqrt{d})$ and

$$\|\widehat{\theta}_{\boldsymbol{\sigma}} - \theta_*\|_{U_n} \leqslant \widetilde{O}\big(\sqrt{d/(1 + \mu_{\min}(U_n))}\big),$$

which also recovers the result in Theorem 1. Thus, Theorem 19 is a generalization of Theorem 1 for the possibly infinite dimensional model (18).

The proof of Theorem 19 generalizes the approach used in the proof of Theorem 1 to the setting of the infinite dimensional model (18). However, there are plenty of technical challenges in dealing with the infinite dimensional $\mathcal{L}^2$ space. First of all, since the vectors in the proof of Theorem 1 are generalized to functions and the matrices are generalized to operators, we need to ensure that these functions are well-defined in some proper spaces and figure out the domain/codomain and properties (e.g., linearity, boundedness, self-adjointness, positivity, compactness, invertibility, etc) of those operators. As in the proof of Theorem 1, we would like to write $\widehat{\theta}_{\boldsymbol{\sigma}} - \theta_* = U_{n,\boldsymbol{\sigma}}^{-1} W_n - U_{n,\boldsymbol{\sigma}}^{-1}(\varsigma\theta_*)$ where,

$$W_n := \sum_{j=1}^n \int_S \mathrm{I}_{y^{(j)}}(t)\Phi_j(\omega, t)\mathfrak{m}(dt) - \int_S \Psi_j(\theta_*, t)\Phi_j(\omega, t)\mathfrak{m}(dt),$$

and $\varsigma\theta_* := \sum_{i=1}^\infty \frac{\langle e_i, \theta_*\rangle}{\sigma_i^2} e_i$. However, this sequence $\{\sum_{i=1}^m \frac{\langle e_i, \theta_*\rangle}{\sigma_i^2} e_i\}_{m\in\mathbb{N}}$ only converges for $\theta_* \in \mathcal{L}^2_{\boldsymbol{\sigma}}(\Omega, \mathfrak{n})$ but not $\mathcal{H}_{\boldsymbol{\sigma},\boldsymbol{e}}$. Thus, for general $\theta_* \in \mathcal{H}_{\boldsymbol{\sigma},\boldsymbol{e}}$, $\varsigma\theta_*$ does not exist and we instead consider the finite-rank operator $\varsigma_m : \theta \mapsto \sum_{i=1}^m \frac{\langle e_i, \theta\rangle}{\sigma_i^2} e_i$ on $\mathcal{L}^2(\Omega, \mathfrak{n})$ and the sequence $\{\theta_{*,m} := U_{n,\boldsymbol{\sigma}}^{-1}(U_n\theta_* + \varsigma_m\theta_*)\}_{m\in\mathbb{N}}$ which we show satisfies $\|\theta_{*,m} - \theta_*\|_{U_n,\boldsymbol{\sigma}} \to 0$ as $m \to \infty$. Then, since it suffices to bound

$$\|\widehat{\theta}_{\boldsymbol{\sigma}} - \theta_{*,m}\|_{U_n,\boldsymbol{\sigma}} \leqslant \|U_{n,\boldsymbol{\sigma}}^{-1} W_n\|_{U_n,\boldsymbol{\sigma}} + \|U_{n,\boldsymbol{\sigma}}^{-1}\varsigma_m\theta_*\|_{U_n,\boldsymbol{\sigma}} \leqslant \|W_n\|_{U_{n,\boldsymbol{\sigma}}^{-1}} + \|\theta_*\|_{\boldsymbol{\sigma},\boldsymbol{e}}.$$

To bound $\|W_n\|_{U_{n,\sigma}^{-1}}$, we use the martingale approach as in the proof of Theorem 1. However, after proving that $\{M_n(\alpha) := \exp\left(\langle\alpha, W_n\rangle - \frac{1}{2}\|\alpha\|_{U_n}^2\right)\}_{n\geqslant 0}$ is a super-martingale for any $\alpha \in \mathcal{L}^2(\Omega, \mathfrak{n})$ wrt the natural filtration $\{\mathcal{F}_n := \sigma(x_1, y_1, \ldots, x_n, y_n, x_{n+1})\}_{n\geqslant 0}$, it is difficult to pick a properly defined "Gaussian" random variable in $\mathcal{L}^2(\Omega, \mathfrak{n})$. Inspired by Lifshits (2012, Example 2.2), we define $\beta = \sum_{i=1}^{\infty} \sigma_i \zeta_i e_i$ with $\{\zeta_i\}_{i\in\mathbb{N}}$ being a sequence of independent $N(0,1)$-random variables. Note that $\beta \in \mathcal{L}^2(\Omega, \mathfrak{n})$ a.s. if $\sum_{i=1}^{\infty}\sigma_i^2 < \infty$. Thus, we can define $\overline{M}_n := \mathbb{E}[M_n(\beta)|\mathcal{F}_\infty]$ with $\mathcal{F}_\infty := \sigma(\cup_{n=1}^{\infty}\mathcal{F}_n)$. Then, we prove that $\{M_n\}_{n\geqslant 0}$ is also a super-martingale wrt $\{\mathcal{F}_n\}_{n\geqslant 0}$ and the question remained is to calculate $M_n$. However, directly generalizing (41), we would get

$$\text{``}\|W_n\|_{U_{n,\sigma}^{-1}}^2 - \|\beta - U_{n,\sigma}^{-1}W_n\|_{U_{n,\sigma}}^2 = 2\langle\beta, W_n\rangle - \|\beta\|_{U_{n,\sigma}}^2\text{''}$$

which does not make sense because $\|\beta\|_{U_{n,\sigma}}$ could be $\infty$ with positive probability. Since it is hard to deal with this in the integration over the the law of $\beta$, we instead adopt the similar approach as we do for $\theta_*$. Define $\beta_m := \sum_{i=1}^{m}\sigma_i\zeta_i e_i$ and $W_{n,m} := \sum_{i=1}^{m}\langle e_i, W_n\rangle e_i$. Then, after some calculation, we get

$$\|W_{n,m}\|_{U_{n,\sigma}^{-1}}^2 - \|\beta_m - U_{n,\sigma}^{-1}W_{n,m}\|_{U_{n,\sigma}}^2 = 2\langle\beta_m, W_{n,m}\rangle - \|\beta_m\|_{U_{n,\sigma}}^2 \text{ and}$$

$$\mathbb{E}[\exp(H_m)|\mathcal{F}_\infty] = \frac{1}{\sqrt{\prod_{i=1}^{m}(1+\lambda_i\sigma_i^2)}}\exp\left(\frac{1}{2}\|W_{n,m}\|_{U_{n,\sigma}^{-1}}^2\right),$$

where $\exp(H_m) := \exp\left\{\langle\beta_m, W_{n,m}\rangle - \frac{1}{2}\|\beta_m\|_{U_n}^2\right\}$. Afterwards, we use dominated convergence theorem to conclude that,

$$\lim_{m\to\infty}\mathbb{E}[\exp(H_m)|\mathcal{F}_\infty] = \mathbb{E}[M_n|\mathcal{F}_\infty] = \overline{M}_n, \ a.s..$$

The verification the integrability of the dominating function $\exp\left(n\sum_{i=1}^{\infty}|\sigma_i\zeta_i| + \frac{1}{2}\sum_{i=1}^{\infty}\lambda_i\sigma_i^2\zeta_i^2\right)$ is also quite technical, during which the condition that $\sum_{i=1}^{\infty}|\sigma_i| < \infty$ is used. Finally, we obtain that $\overline{M}_n = \frac{1}{\sqrt{\prod_{i=1}^{\infty}(1+\lambda_i\sigma_i^2)}}\exp\left(\frac{1}{2}\|W_n\|_{U_{n,\sigma}^{-1}}^2\right)$. Then, by applying *Doob's maximal inequality* for super-martingales, we can bound $\|W_n\|_{U_{n,\sigma}^{-1}}^2$ which yields the final bound on $\|\widehat{\theta}_\sigma - \theta_*\|_{U_{n,\sigma}}$ in (23). (24) immediately follows from (23) and Corollary 14.

## 7 Numerical studies

In this section, we demonstrate the scaling of estimation errors of the proposed estimator empirically in our synthetic data experiments in Section 7.1 and illustrate the practical utility of the proposed estimator in our real data experiments in Section 7.2.

### 7.1 Synthetic data experiments

This section contains the experimental results on discrete and continuous synthetic data.

**Bernoulli data experiments.** To illustrate that our estimator (2) achieves the $\ell^2$-error rate of $\widetilde{\Theta}(\sqrt{d/(1+\mu_{\min}(U_n))})$ in the estimation of $\theta$ under model (1), we consider the Bernoulli data generated according to the hard instance used to show the lower bound in the proof of Theorem 8 in Appendix C.1. Specifically, after choosing a true parameter $\theta_* \in \Delta^{d-1}$ of dimension $d \in \mathbb{N}$, for any $j \in \mathbb{N}$, we set $\phi_i(x_j, \cdot)$ as the CDF of Bernoulli$(p_{ji})$ for $i \in [d]$, where $p_j := [p_{j1}, \ldots, p_{jd}]^\top \in [0,1]^d$ is defined as follows. When $j \in [d]$, we set $p_{ji} = 1 - \frac{c_j}{2d^3} - \frac{c_j\mathbb{1}\{i=j\}}{2d^3}$; when $j > d$, we set $p_{ji} = 1 - \frac{c_j\mu_{\min}(R_{j-1})}{2d^2} - \frac{c_j\mu_{\min}(R_{j-1})\mathbb{1}\{i=(j\mod d)\}}{2d^2}$, where mod denotes the modulo operation, $c_j$'s are constants independent of $d$, and $R_j := q_jq_j^\top + \frac{1}{n}\sum_{k=1}^{j-1}q_kq_k^\top$ for any $j \geqslant d$ with $q_j := [1-p_{j1}, \ldots, 1-p_{jd}]^\top$. Then, we sample $y_j$ independently from Bernoulli$(\theta_*^\top p_j)$ whose CDF is $\theta_*^\top\Phi_j$. Given $n$ samples, we calculate $\widehat{\theta}_\lambda$ using different values of $\lambda$ according to (21) with $S = [0,1]$ and $\mathfrak{m} = \mathsf{Leb}([0,1])$. We evaluate the performance using the un-normalized $\ell^2$-error $\|\widehat{\theta}_\lambda - \theta_*\|$, the self-normalized error $\|\widehat{\theta}_\lambda - \theta_*\|_{U_n(\lambda)}$, and the KS distance $\mathrm{KS}(\widehat{F}_\lambda(x,\cdot), F(x,\cdot))$ (for KS distance, we consider the family of Bernoulli distributions with parameters in $[1-\frac{1}{d^2}, 1-\frac{1}{2d^2}]$ to align with the setting of $p_j$ in

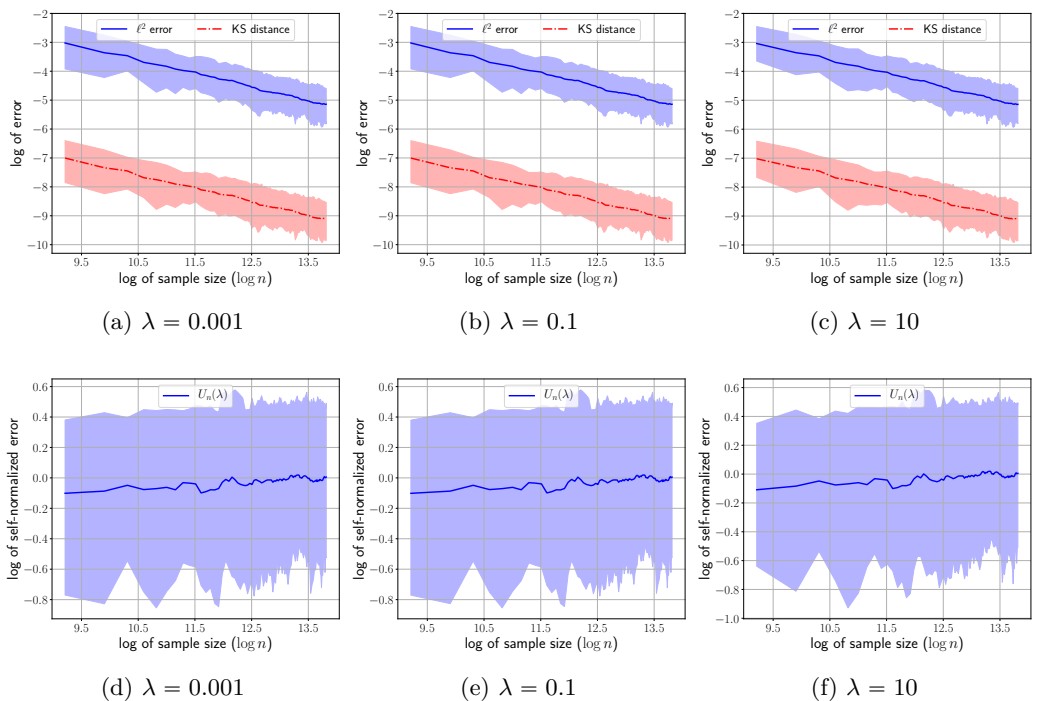

Figure 2: Means and 90% confidence intervals of un-normalized $\ell^2$-errors $\|\widehat{\theta}_\lambda - \theta_*\|$, KS distances $\mathrm{KS}(\widehat{F}_\lambda(x, \cdot), F(x, \cdot))$, and self-normalized errors $\|\widehat{\theta}_\lambda - \theta_*\|_{U_n(\lambda)}$ against sample size $n$ in logarithmic scale in Bernoulli synthetic data experiments.

data generation). We repeat the experiments 100 times to calculate means and 90% confidence intervals of the errors.

We first study the dependence of estimation errors of our estimator (2) on sample size $n$ with the dimension $d = 5$. Specifically, for $\lambda = 0.001, 0.1$, and 10, we run the experiments with $n$ ranging from $10^4$ to $10^6$ and plot the means and 90% confidence intervals of the errors against $n$ (both in logarithmic scale) in Figure 2. According to Figure 2, for different values of $\lambda$, the slopes of the curves of $\log\|\widehat{\theta}_\lambda - \theta_*\|$, $\log \mathrm{KS}(\widehat{F}_\lambda(x, \cdot), F(x, \cdot))$, and $\log\|\widehat{\theta}_\lambda - \theta_*\|_{U_n(\lambda)}$ against $\log n$ are around $-0.5, -0.5$, and $0.025$, which obeys the $\widetilde{\Theta}(\sqrt{d/n})$, $\widetilde{O}(d/\sqrt{n})$ (assuming $\mu_{\min}(U_n)$ grows linearly with $n$), and $O(\sqrt{d\log(1 + n/\lambda)})$ upper bounds on the errors $\|\widehat{\theta}_\lambda - \theta_*\|$, $\mathrm{KS}(\widehat{F}_\lambda(x, \cdot), F(x, \cdot))$, and $\|\widehat{\theta}_\lambda - \theta_*\|_{U_n(\lambda)}$ respectively according to Theorem 1 and 8.

Then, we study the dependence of estimation errors of estimator (2) on dimension $d$ with the sample size $n = 10^6$. For $\lambda = 0.001, 0.1$, and 10, we run the experiments with $d$ ranging from 10 to 100. Then, we plot the means and 90% confidence intervals of $\log\|\widehat{\theta}_\lambda - \theta_*\|$ and $\log \mathrm{KS}(\widehat{F}_\lambda(x, \cdot), F(x, \cdot))$ against $\log d - \log \mu_{\min}(U_n(\lambda))$ as well as $\log\|\widehat{\theta}_\lambda - \theta_*\|_{U_n(\lambda)}$ against $\log d$ in Figure 3. According to Figure 3, for different values of $\lambda$, the slopes of the curves of $\log\|\widehat{\theta}_\lambda - \theta_*\|$ and $\log \mathrm{KS}(\widehat{F}_\lambda(x, \cdot), F(x, \cdot))$ against $\log d - \log \mu_{\min}(U_n(\lambda))$ are around $0.5$ and $-0.5$ respectively, and the slopes of the curves of $\log\|\widehat{\theta}_\lambda - \theta_*\|_{U_n(\lambda)}$ against $\log d$ are around $0.5$. These results also obey the $\widetilde{\Theta}(\sqrt{d/(1 + \mu_{\min}(U_n))})$, $\widetilde{O}(d/\sqrt{(1 + \mu_{\min}(U_n))})$, and $O(\sqrt{d\log(1 + n/\lambda)})$ upper bounds on the errors $\|\widehat{\theta}_\lambda - \theta_*\|$, $\mathrm{KS}(\widehat{F}_\lambda(x, \cdot), F(x, \cdot))$, and $\|\widehat{\theta}_\lambda - \theta_*\|_{U_n(\lambda)}$ respectively according to Theorem 1 and 8.

**Polynomial CDF data experiments.** For $d \in \mathbb{N}$, $r(i) := i$ if $1 \leqslant i \leqslant \frac{d+1}{2}$, and $r(i) := \frac{2}{2i-d+1}$ if $\frac{d+1}{2} < i \leqslant d$, we consider the following basis CDFs:

$$\phi_i(x, t) = \mathbb{1}\{t \in [0, 1/x]\}(xt)^{r(i)} + \mathbb{1}\{t > 1/x\}, \; i \in [d]. \tag{25}$$

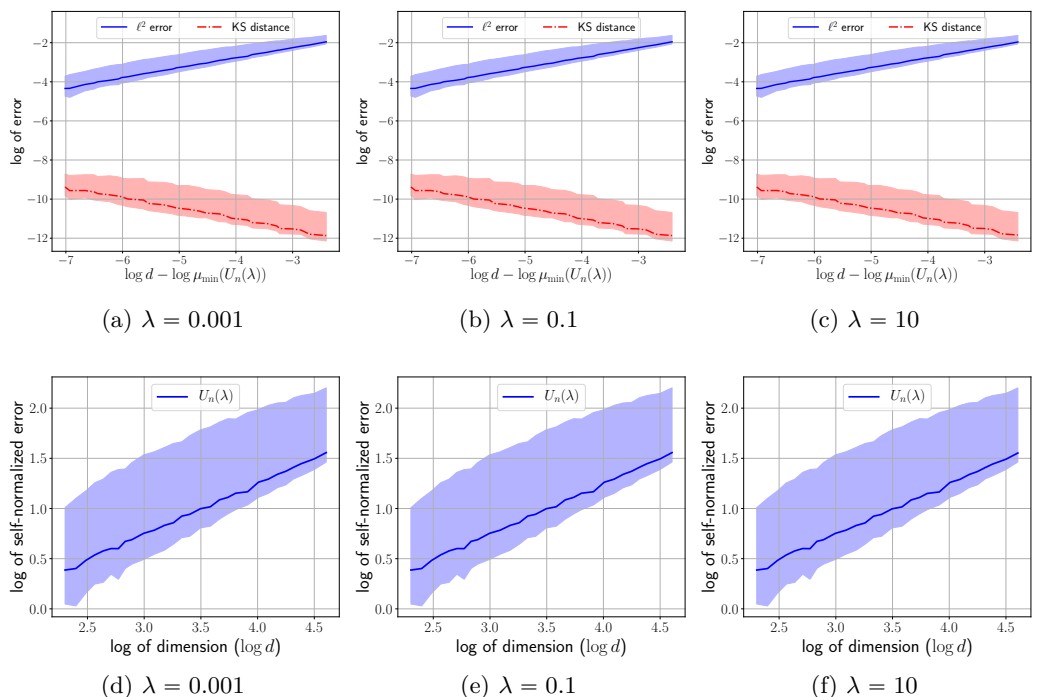

Figure 3: Means and 90% confidence intervals of un-normalized $\ell^2$-errors $\|\widehat{\theta}_\lambda - \theta_*\|$, KS distances $\text{KS}(\widehat{F}_\lambda(x,\cdot), F(x,\cdot))$, and self-normalized errors $\|\widehat{\theta}_\lambda - \theta_*\|_{U_n(\lambda)}$ against $d/\mu_{\min}(U_n(\lambda))$ and dimension $d$ in logarithmic scale in Bernoulli synthetic data experiments.

To simulate $n$ samples, we first choose a true parameter $\theta_*$. For each $j \in [n]$, $x_j$ is sampled independently from the uniform distribution on $[0.5, 2]$. Then, we sample $y_j$ independently from the CDF $\theta_*^\top \Phi(x_j, \cdot)$ using the inverse CDF method for $j \in [n]$. Given the simulated sample, we calculate $\widehat{\theta}_\lambda$ using (3) with $S = [0, 2]$, $\mathfrak{m}$ chosen as the uniformly distribution $\mathfrak{m}_\mathsf{U}$ on $S$, and different values of $\lambda$. We evaluate the performance by calculating $\ell^2$-error $\|\widehat{\theta}_\lambda - \theta_*\|$, the self-normalized errors $\|\widehat{\theta}_\lambda - \theta_*\|_{U_n(\lambda)}$ and $\|\widehat{\theta}_\lambda - \theta_*\|_{\Sigma_n}$, and the KS distance $\text{KS}(\widehat{F}_\lambda(x,\cdot), F(x,\cdot))$. To obtain stable results, we repeat the simulation independently 100 times in each setting to calculate 90% confidence intervals and means of the errors.

Fixing $d = 5$, we study the dependence of estimation errors of our estimator (2) on sample size $n$ using $\lambda = 0.001, 0.1$, and 10. We run the experiments with $n$ ranging from $10^4$ to $10^6$ and plot the means and 90% confidence intervals of the errors against $n$ (both in logarithmic scale) in Figure 4. According to Figure 4, for different values of $\lambda$, the slopes of the curves of $\log \|\widehat{\theta}_\lambda - \theta_*\|_{U_n(\lambda)}$ and $\|\widehat{\theta}_\lambda - \theta_*\|_{\Sigma_n}$ against $\log n$ are around 0, which obeys the $O(\sqrt{d \log(1 + n/\lambda)})$ upper bounds proved in Theorem 1 and Proposition 5. When $\lambda$ is negligible compared to $\mu_{\min}(U_n))$, the $\widetilde{O}(\sqrt{d/(\lambda + \mu_{\min}(U_n))})$ bound on $\ell^2$-error followed from Theorem 1 implies the $\widetilde{O}(\sqrt{d/n})$ $\ell^2$-error bound if $\mu_{\min}(U_n))$ grows linearly with $n$. Indeed, for small $\lambda = 0.001$, the slope of the curve of $\log \|\widehat{\theta}_\lambda - \theta_*\|$ against $\log n$ in Figure 4a is around $-0.5$. When $\lambda$ is comparable with $\mu_{\min}(U_n))$, as is observed in Figure 4b and 4c, the slopes of the curves of $\ell^2$-errors are larger than $-0.5$, which is expected from the $\widetilde{O}(\sqrt{d/(\lambda + \mu_{\min}(U_n))})$ bound. The slopes of the curves of the KS distances against $\log n$ are smaller than $0.5$, also obeying the $\widetilde{O}(d/\sqrt{(\lambda + \mu_{\min}(U_n))})$ bound implied by Theorem 1.

Next, fixing $n = 10^5$, we run the experiments with $d$ ranging from 10 to 100 using $\lambda = 0.001, 0.1$, and 10. We plot the means and 90% confidence intervals of the errors against $d$ (both in logarithmic scale) in Figure 5. According to Figure 5, for different values of $\lambda$, the slopes of the curves of $\log \|\widehat{\theta}_\lambda - \theta_*\|$, $\log \|\widehat{\theta}_\lambda - \theta_*\|_{U_n(\lambda)}$, and $\log \|\widehat{\theta}_\lambda - \theta_*\|_{\Sigma_n}$ against $\log d$ are around 0, obeying the respective $\widetilde{O}(\sqrt{d/(\lambda + \mu_{\min}(U_n))})$, $O(\sqrt{d \log(1 + n/\lambda)})$, and $O(\sqrt{d \log(1 + n/\lambda)})$ bounds proved in Theorem 1 and Proposition 5. The slopes of the curves of the

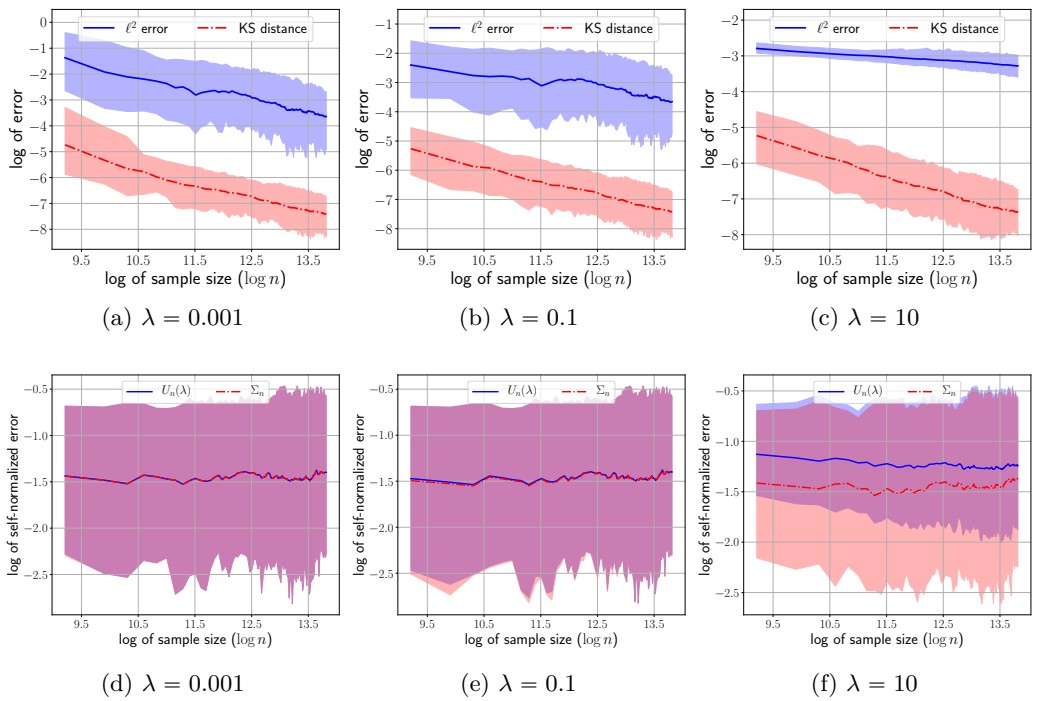

Figure 4: Means and 90% confidence intervals of un-normalized $\ell^2$-errors $\|\widehat{\theta}_\lambda - \theta_*\|$, KS distances $\mathrm{KS}(\widehat{F}_\lambda(x,\cdot), F(x,\cdot))$, and self-normalized errors $\|\widehat{\theta}_\lambda - \theta_*\|_{U_n(\lambda)}$ and $\|\widehat{\theta}_\lambda - \theta_*\|_{\Sigma_n}$ against sample size $n$ in logarithmic scale in polynomial CDF synthetic data experiments.

KS distances are smaller than 1, which also obeys the $\widetilde{O}(d/\sqrt{(\lambda + \mu_{\min}(U_n))})$ bound implied by Theorem 1. Since the lower bounds are proved for the worst case of any estimator, the results above do not violate our theoretical results on lower bound.

## 7.2 Real data experiments

We compare the empirical performance of our estimator (2) and other methods on two real-world datasets: the California house price dataset and the adult income dataset.

**California house price dataset.** We evaluate the performance of estimator (2) on the California house price dataset (Mohapatra, 2022) of size $n = 20,640$ from Kaggle. There are 10 attributes among which we use median house value as the samples $y$ from target CDFs and all other attributes as the contexts $x$ ($d = 9$). We standardize all the ordinal variables.

We apply the proposed estimator (2) and three other methods, MLE, empirical CDF (ECDF), and kernel density estimation (KDE) to estimate contextual CDFs for this dataset. Specifically, for ECDF, given samples $y^{(1)}, \ldots, y^{(n)}$, the empirical CDF is as follows,

$$\widehat{F}_E(t) := \frac{1}{n} \sum_{j=1}^{n} \mathrm{I}_{y^{(j)}}(t) = \frac{1}{n} \sum_{j=1}^{n} \mathbb{1}\{y^{(j)} \leqslant t\}. \tag{26}$$

For KDE, we apply the function "density" in the R package "stats" with Gaussian, rectangular, and triangular kernels. Note that only the samples $y$ are used to estimate one CDF without considering the contexts $x$ in ECDF and KDE. For the proposed estimator and MLE, we assume the linear model (1). We consider the following family of basis CDFs:

$$\phi_i(x, t) = (1 - w) F_N(t; \beta_{N,i}^{(1)} x_i + \beta_{N,i}^{(0)}, \sigma_i^2) + w F_L(t; \beta_{L,i}^{(1)} x_i + \beta_{L,i}^{(0)}, b_i), \ t \in \mathbb{R}, \ i \in [d], \tag{27}$$

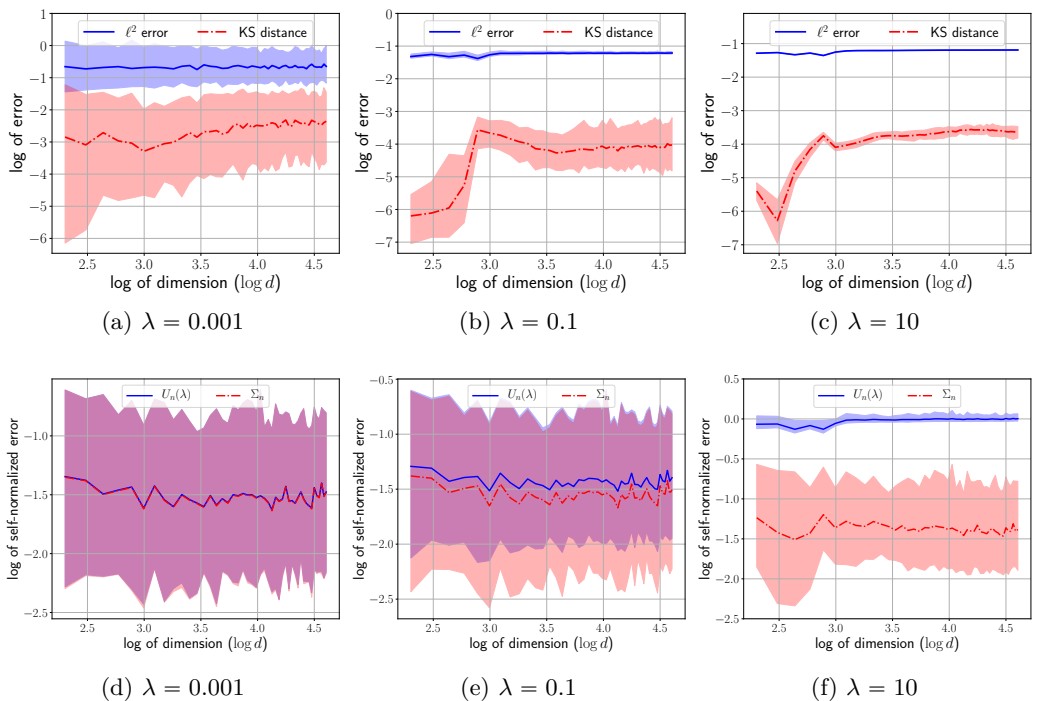

Figure 5: Means and 90% confidence intervals of un-normalized $\ell^2$-errors $\|\widehat{\theta}_\lambda - \theta_*\|$, KS distances $\mathrm{KS}(\widehat{F}_\lambda(x,\cdot), F(x,\cdot))$, and self-normalized errors $\|\widehat{\theta}_\lambda - \theta_*\|_{U_n(\lambda)}$ and $\|\widehat{\theta}_\lambda - \theta_*\|_{\Sigma_n}$ against dimension $d$ in logarithmic scale in polynomial CDF synthetic data experiments.

where $x = (x_1, \ldots, x_d)$ is the context, $F_N(\cdot; \mu, \sigma^2)$ is the CDF of the Gaussian distribution $N(\mu, \sigma^2)$, $F_L(\cdot; \mu, b)$ is the CDF of the Laplace distribution $\mathrm{Laplace}(\mu, b)$, $w$ is the weight of Laplace distributions, $\beta_{N,i}^{(1)}$ ($\beta_{N,i}^{(0)}$) is the coefficient (intercept) in the Gaussian linear model of $x_i$, and $\beta_{L,i}^{(1)}$ ($\beta_{L,i}^{(0)}$) is the coefficient (intercept) in the Laplace linear model of $x_i$.

We split the whole dataset into subsets of fractions 1/3, 1/2, and 1/6. 1/3 data points are used to estimate the coefficients and intercepts under Gaussian or Laplace linear models separately by maximizing log likelihood. For Laplace linear model, it corresponds to the least absolute residual regression which we solve using the function "lad" in the R package "L1pack" (Osorio & Wolodzko, 2023). Afterwards, we estimate $\sigma_i^2$'s and $b_i$'s using the sample variance and the mean absolute deviation of the their corresponding residuals respectively. Then, we apply different methods on the second subset (training dataset) of 1/2 data points. For the proposed estimator, we calculate $\widehat{\theta}_\lambda$ using (3) with $S = \mathbb{R}$, $\mathfrak{m} = \gamma_{0,100}$, and $\lambda = 0.1, 1, 5$. For MLE, we can formulate the likelihood function of the parameter $\theta$ in (1) with $\Phi$ specified in (27). Let $\widehat{\theta}_{MLE}$ denote a maximizer of likelihood function. Since under (1), MLE corresponds to solving a convex minimization problem in a convex set (the probability simplex $\Delta^{d-1}$), we use the solver "SCS" in the R package "CVXR" (Fu et al., 2020) to calculate $\widehat{\theta}_{MLE}$. Let $\widehat{F}_E$ denote the ECDF calculated by (26) using the training dataset. Let $\widehat{F}_{KG}$, $\widehat{F}_{KR}$, and $\widehat{F}_{KT}$ denote the CDF calculated by KDE with Gaussian, rectangular, and triangular kernels respectively using the training dataset. Given samples $y^{(1)}, \ldots, y^{(n)}$, we define the $\mathcal{L}^2$-error of an estimated CDF $\widehat{F}$ as

$$\frac{1}{n} \sum_{j=1}^n \|\mathrm{I}_{y^{(j)}} - \widehat{F}\|_{\mathcal{L}^2(S,\mathfrak{m})}^2, \tag{28}$$

where we also set $S = \mathbb{R}$ and $\mathfrak{m} = \gamma_{0,100}$. Note that when $S = \mathbb{R}$ and $\mathfrak{m} = \mathsf{Leb}(\mathbb{R})$, the $\mathcal{L}^2$-error in (28) corresponds to the renowned Continuous Ranked Probability Score (CRPS) (Hersbach, 2000) used to assess the performance of a CDF in approximating data distribution. We calculate $\mathcal{L}^2$-errors on the third subset

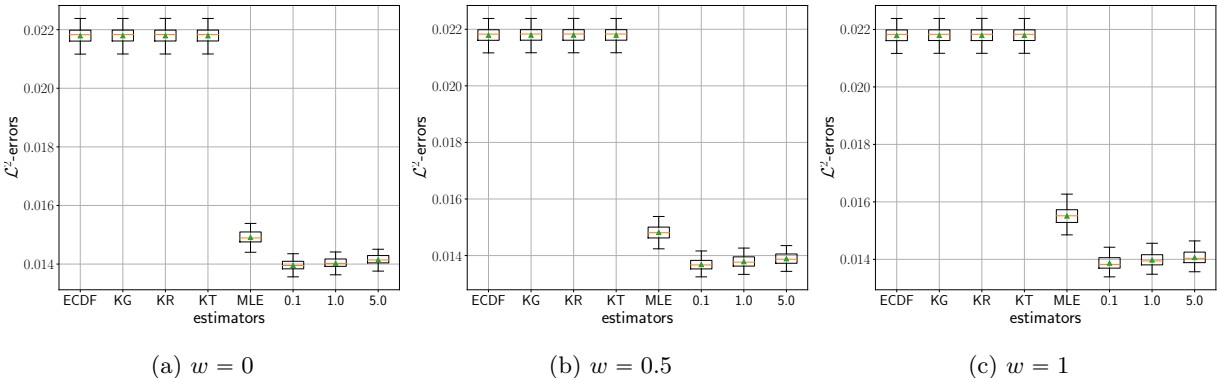

Figure 6: Box plots of $\mathcal{L}^2$-errors in the California house price data experiment. "ECDF" refers to the empirical CDF defined in (26). "KG", "KR", and "KT" refer to the kernel density estimation method using Gaussian, rectangular, and triangular kernels respectively. "0.1", "1.0", and "5.0" refer to our estimator $\widehat{\theta}_\lambda$ in (2) with $\lambda = 0.1$, 1.0, and 5.0 respectively.

(test dataset) of 1/6 data points for the four methods described previously. For ECDF and KDE, we plug $\widehat{F}_E$, $\widehat{F}_{KG}$, $\widehat{F}_{KR}$, and $\widehat{F}_{KT}$ in (28). For MLE and the proposed estimator, we calculate $\mathcal{L}^2$-errors using $\frac{1}{n}\sum_{j=1}^n \|\mathrm{I}_{y^{(j)}} - \widehat{\theta}_{MLE}^\top \Phi_j\|_{\mathcal{L}^2(S,\mathfrak{m})}^2$ and $\frac{1}{n}\sum_{j=1}^n \|\mathrm{I}_{y^{(j)}} - \widehat{\theta}_\lambda^\top \Phi_j\|_{\mathcal{L}^2(S,\mathfrak{m})}^2$ with different values of $\lambda$.

We run the experiments with $w = 0$, 0.5, and 1 in (27). To get stable results, we permute the dataset uniformly at random independently and repeat the experiments 100 times to calculate $\mathcal{L}^2$-errors. We draw the box plots of the $\mathcal{L}^2$-errors of different methods with different values of $w$ in Figure 6. As is shown in the figure, ECDF and KDE have comparable $\mathcal{L}^2$-errors which are much larger than the other two methods. For all choices of $w$ and $\lambda$, our estimator (2) achieves the smallest $\mathcal{L}^2$-error than any other method, indicating that its performance is very robust in the choices of basis CDFs and regularization level. Also, $\mathcal{L}^2$-error of our estimator decreases with the value of $\lambda$ as expected. Thus, with different basis contextual CDFs, our estimator (2) has better performance in approximating target data distributions and the performance is stable wrt the value of $\lambda$ in (2).

**Adult income dataset.** The adult income dataset (Becker & Kohavi, 1996) was extracted from the 1994 census bureau database. The typical learning task is to predict whether income exceeds \$50K/yr based on other attributes in the census data. Thus, we use the attributes age, workclass, education, marital-status, occupation, relationship, race, sex, capital-gain, capital-loss, hours-per-week, and native-country as the contexts $x$ ($d = 12$), and use income (i.e., whether income exceeds \$50K/yr) as the samples $y$ from target CDFs. We standardize all of the ordinal attributes. The total number of samples is $n = 48,842$.

Since the samples follow Bernoulli distributions, KDE is not considered. We apply our estimator (2), MLE, and ECDF on this dataset. For our estimator and MLE, we assume model (1) and use the following mixtures of logistic and probit models as basis CDFs:

$$\phi_i(x,t) = wF_B(t; f_{\mathrm{logi}}(\beta_{L,i}^{(1)}x_i + \beta_{L,i}^{(0)})) + (1-w)F_B(t; F_N(\beta_{P,i}^{(1)}x_i + \beta_{P,i}^{(0)}; 0, 1)), \ i \in [d], \tag{29}$$

where $t \in \mathbb{R}$, $x = (x_1, \ldots, x_d)$ denotes the context, $F_B(\cdot; p)$ denotes the CDF of the Bernoulli distribution with parameter $p$, $f_{\mathrm{logi}}(a) := 1/(1 + e^{-a})$ for any $a \in \mathbb{R}$, $w$ is the weight of the logistic model, $\beta_{L,i}^{(1)}$ ($\beta_{L,i}^{(0)}$) denotes the coefficient (intercept) in the logistic model of $x_i$, and $\beta_{P,i}^{(1)}$ ($\beta_{P,i}^{(0)}$) denotes the coefficient (intercept) in the probit model of $x_i$. We split the whole dataset into subsets of fractions 1/3, 1/2, and 1/6. 1/3 data points are used to estimate the coefficients and intercepts in (29) with the function "glm" in the R package "stats".

We apply all methods on the second subset (training dataset) of 1/2 data points. For our estimator, we calculate $\widehat{\theta}_\lambda$ using (3) with $S = [0,1]$, $\mathfrak{m} = \mathsf{Leb}([0,1])$, and $\lambda = 0.1, 1, 5$. For MLE, we also use the solver

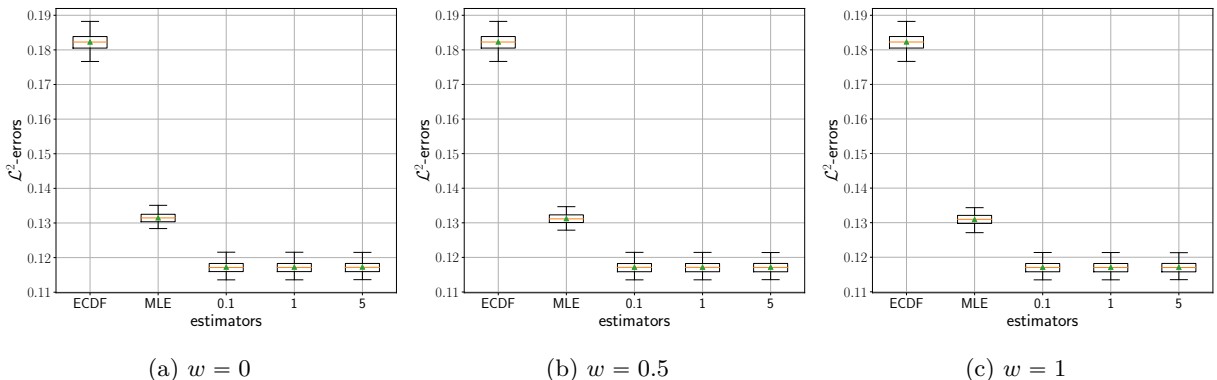

Figure 7: Box plots of $\mathcal{L}^2$-errors in adult income data experiments. "ECDF" refers to the empirical CDF defined in (26). "0.1", "1.0", and "5.0" refer to the estimator $\widehat{\theta}_\lambda$ (2) with $\lambda = 0.1$, 1.0, and 5.0 respectively.

"SCS" in the R package "CVXR" (Fu et al., 2020) to calculate $\widehat{\theta}_{MLE}$ as in the previous example. We use $\widehat{F}_E$ to denote the ECDF calculated by (26) using the training dataset. Then, we calculate $\mathcal{L}^2$-errors (28) with $S = [0, 1]$ and $\mathfrak{m} = \mathsf{Leb}([0, 1])$ for the three methods on the third subset (test dataset) of $1/6$ data points.

We run the experiments described above using $w = 0$, 0.5, and 1 in (29) 100 times with the dataset permuted randomly in each run to get stable results. In Figure 7, we report the calculated $\mathcal{L}^2$-errors in box plots. According to the figure, our estimator (2) achieves the smallest $\mathcal{L}^2$-errors for all choices of $\lambda$ and weight $w$ and ECDF has the largest $\mathcal{L}^2$-error for all choices of $w$. Moreover, the performance of our estimator is very robust wrt $\lambda$ and $w$. Thus, with a wide range of the basis contextual CDFs, our estimator (2) achieves good and robust performance in approximating target data distributions.

## 8 Conclusion

In this paper, we propose a linear model for contextual CDFs and estimators for the coefficient parameter in this model. We prove $\widetilde{O}(\sqrt{d/n})$ upper bounds on the estimation error of our estimator under the adversarial and random settings, and show that the upper bounds are tight up to logarithmic factors by proving $\Omega(\sqrt{d/n})$ information theoretic lower bounds. Additionally, when a mismatch exists in the linear model, we prove that the estimation error of our estimator only increases by an amount commensurate with the mismatch error. Furthermore, we increase the generality of our linear model by expanding the parameter space into an infinite dimensional Hilbert space. Within this framework, we generalize our estimator and subsequently establish self-normalized upper bounds for this general estimator. Moreover, we elucidate the scaling of the estimation error of our estimator empirically and showcase its practical utility on real-world datasets.

Our current work assumes that the bases are known a priori. So, a fruitful future research direction would be to focus on the basis selection problem for CDF regression with possibly infinitely many base functions. More generally, it is a promising future direction to consider the adaptive setting where the learner seeks to learn both the basis $\Phi$ and the weight parameter $\theta_*$ in (1) by querying samples from $\phi_i$'s and $F = \theta_*^\top \Phi$. A closely related challenge is presented in multi-distribution learning where the learner strategically queries samples from different distributions with the objective of minimizing the expected risk uniformly across all distributions (Haghtalab et al., 2022; Awasthi et al., 2023). Leveraging our existing results on estimating $\theta_*$ based on $\Phi$, we can decompose this learning problem into two subproblems: (i) the estimation of the basis CDFs $\phi_i$'s from their respective samples, and (ii) the planning of the queries to samples from different distributions. As discussed in related works, numerous existing results address the estimation of a single contextual CDF based on certain distributional assumptions, placing the primary challenge in solving (ii). A natural idea is to initiate from the offline algorithm, akin to Wang et al. (2022): for each $i \in [d]$, the learner queries a pre-specified number of samples from each distribution to estimate $\phi_i$, and then queries samples from $F$ to estimate $\theta_*$ with the learned $\Phi$, where the allocation of queries to each distribution is determined

by minimizing the upper bound of the estimation error of $\theta_*$ under the constraint of a constant sum. Though this approach is straightforward, we can conjecture whether the offline algorithm is optimal, considering that the learning of $\theta_*$ may not contribute much to the learning of each individual basis CDF. Moving forward, we can explore online algorithms which adaptively determine the next oracle to query. The design of the online algorithms in Wang et al. (2022) and Awasthi et al. (2023) are anticipated to offer valuable guidance in this direction.

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

# A Discussion on the minimax lower bound for the estimation of CDFs

First, for any contextual CDFs $F_1$ and $F_2$, define the uniform KS distance by

$$\mathrm{KS}(F_1, F_2) := \sup_{x \in \mathcal{X}} \mathrm{KS}(F_1(x, \cdot), F_2(x, \cdot)).$$

Similar to the minimax $\ell^2$-risk defined in (12), we can define the minimax risk in terms of the uniform KS distance for the estimation of the contextual CDF $F$. For any distribution family $\mathcal{Q}$ and the contextual CDF function $\Xi : \mathcal{Q} \to [0, 1]^{\mathcal{X} \times \mathbb{R}}$, the minimax risk in terms of the uniform KS distance is defined as

$$\mathfrak{R}(\Xi(\mathcal{Q}); \mathrm{KS}) := \inf_{\hat{\Xi}} \sup_{Q \in \mathcal{Q}} \mathbb{E}_{z \sim Q}[\mathrm{KS}(\hat{\Xi}(z), \Xi(Q))].$$

We follow the notation in Section 4. With a slight abuse of notation, let $F(P) = \theta(P)^\top \Phi$. For the random setting, define the distribution family $\mathcal{P}_0 := \left\{ \otimes_{j=1}^n P_X^{(j)} P_{Y|X;\theta} : \theta \in \mathbb{R}^d, \ P_X^{(j)} \in \mathcal{D}_{\mathcal{X}} \text{ such that } \mu_{\min}(\Sigma_n) = 0 \right\}$. Then, we have the following results.

**Proposition 20.** *For any sequence $x^{1:n} = (x^{(1)}, \ldots, x^{(n)}) \in \mathcal{X}^n$ such that $\mu_{\min}(U_n) = 0$, we have*

$$\mathfrak{R}(F(\mathcal{P}_{x^{1:n}}); \mathrm{KS}) = \Omega(1). \tag{30}$$

*For the random setting (Scheme II), we have*

$$\mathfrak{R}(F(\mathcal{P}_0); \mathrm{KS}) = \Omega(1). \tag{31}$$

*Proof of Proposition 20.* According to the discussion below Theorem 8, the discussion above Corollary 9, and Appendix C.2, it suffices to show (30) under the fixed design setting.

Let us consider the fixed design setting where $\phi_i(x, \cdot)$ are the CDFs of Bernoulli distributions for $i \in [d]$, $d \geqslant 2$. Let $q_{ji}$ denote the zero probability of the Bernoulli distribution with CDF $\phi_i(x^{(j)}, \cdot) = \Phi_{ji}(\cdot)$. We set $S = [0, 1]$ and $\mathfrak{m} = \mathsf{Leb}$. Then, we have $U_n = \sum_{j=1}^n q_j q_j^\top$ where $q_j = [q_{j1}, \ldots, q_{jd}]^\top$.

For any $\theta_* \in \Delta^{d-1}$, we have $F(x^{(j)}, t) = \theta_*^\top q_j$ under model (1) for any $t \in [0, 1)$. Suppose that $q_{ji} = q_{11} \in [0, 1]$ for any $i \in [d]$ and $j \in [n]$. Then for any $\theta_* \in [0, 1]$, the samples $y^{(j)}$'s for $j \in [n]$ are generated from the same distribution which is the Bernoulli distribution with success probability $1 - q_{11}$. We have $\mu_{\min}(U_n) = 0$. Thus, the condition of the proposition is satisfied.

For $n + 1$, suppose that $q_{n+1,1} = 1$ and $q_{n+1,2} = 0$. Then, for any estimate $\check{F}_n$ of $F$, we have $\check{F}_n(x^{(n+1)}, 1/2) \in [0, 1]$. If $\check{F}_n(x^{(n+1)}, 1/2) \in [0, 1/2]$, consider the case where $\theta_* = \theta^{(1)} = [1, 0, \ldots, 0]^\top$. Then, we have $|\check{F}_n(x^{(n+1)}, 1/2) - F(x^{(n+1)}, 1/2)| \geqslant 1/2$. If $\check{F}_n(x^{(n+1)}, 1/2) \in (1/2, 1]$, consider the case where $\theta_* = \theta^{(2)} = [0, 1, \ldots, 0]^\top$. Then, we also have $|\check{F}_n(x^{(n+1)}, 1/2) - F(x^{(n+1)}, 1/2)| \geqslant 1/2$. Thus, we have

$$\mathfrak{R}(F(\mathcal{P}_{x^{1:n}}); \mathrm{KS}) = \inf_{\check{F}_n} \sup_{P \in \mathcal{P}_{x^{1:(n)}}} \mathrm{KS}(\check{F}_n, F) = \Omega(1).$$

Thus, the minimax risk in terms of the uniform KS distance of any estimate of $F$ is $\Omega(1)$. $\qquad \square$

Recall that according to the discussion at the end of Section 3.2, for the plug-in estimate $\hat{F}_\lambda$ of $F$ using our projected estimator $\hat{\theta}_\lambda$, we have the $\widetilde{O}(\min\{1, d/\sqrt{1 + \mu_{\min}(U_n)}\})$ upper bound in terms of the uniform KS distance. Proposition 20 implies that this plug-in estimate $\hat{F}_\lambda$ is minimax optimal when $\mu_{\min}(U_n) = 0$.

It is worth noting that with the assumption that $\mu_{\min}(U_n) = \Theta(n)$, the

$$\widetilde{O}(\min\{1, d/\sqrt{1 + \mu_{\min}(U_n)}\}) = \widetilde{O}(d/\sqrt{n})$$

upper bound of $\hat{F}_\lambda$ implies that the minimax lower bound in estimating $F$ is improved. Thus, we can see that $\mu_{\min}(U_n)$ or $\mu_{\min}(\Sigma_n)$ plays an important role in the estimation of $F$.

# B    Proofs of upper bounds for the finite dimensional model

We first briefly expand on the notation for the proofs of our theoretical results. For any topological space $A$, let $\mathcal{B}(A)$ denote the Borel $\sigma$-algebra of $A$. For any two measurable spaces, $(A_1, \mathcal{A}_1)$ and $(A_2, \mathcal{A}_2)$, a function $f : A_1 \to A_2$ is $\mathcal{A}_1/\mathcal{A}_2$-measurable if for any $E \in \mathcal{A}_2$, we have $f^{-1}(E) \in \mathcal{A}_1$. When $\mathcal{A}_2$ is the Borel $\sigma$-algebra on $A_2$, we sometimes write $f$ is $\mathcal{A}_1$-measurable to mean that $f$ is $\mathcal{A}_1/\mathcal{A}_2$-measurable for brevity. When $\mathcal{A}_1$ is the Borel $\sigma$-algebra on $A_1$ and $\mathcal{A}_2$ is the Borel $\sigma$-algebra on $A_2$, we sometimes simply write $f$ is measurable to mean that $f$ is $\mathcal{A}_1/\mathcal{A}_2$-measurable for brevity. For any two $\sigma$-finite measure spaces $(A_1, \mathcal{A}_1, \nu_1)$ and $(A_2, \mathcal{A}_2, \nu_2)$, let $A_1 \times A_1 := \{(a_1, a_2) : a_1 \in A_1, a_2 \in A_2\}$ denote the product space, let $\mathcal{A}_1 \otimes \mathcal{A}_2 := \sigma(\{E_1 \times E_2 : E_1 \in \mathcal{A}_1, E_2 \in \mathcal{A}_2\})$ denote the product $\sigma$-algebra of $\mathcal{A}_1$ and $\mathcal{A}_2$ on $A_1 \times A_2$, and let $\nu_1 \otimes \nu_2$ denote the product measure of $\nu_1$ and $\nu_2$ on $(A_1 \times A_2, \mathcal{A}_1 \otimes \mathcal{A}_2)$ (i.e., $\nu_1 \otimes \nu_2(E_1 \times E_2) = \nu_1(E_1)\nu_2(E_2)$ for any $E_1 \in \mathcal{A}_1$ and $E_2 \in \mathcal{A}_2$) whose existence is guaranteed by Carathéodory's extension theorem. Then, $(A_1 \times A_2, \mathcal{A}_1 \otimes \mathcal{A}_2, \nu_1 \otimes \nu_2)$ is the product measure space of $(A_1, \mathcal{A}_1, \nu_1)$ and $(A_2, \mathcal{A}_2, \nu_2)$. When $A_1 = A_2$ and $\mathcal{A}_1 = \mathcal{A}_2$, we will write $\mathcal{A}_1^2$ to represent $\mathcal{A}_1 \otimes \mathcal{A}_1$. When $A_1 = A_2$, $\mathcal{A}_1 = \mathcal{A}_2$, and $\nu_1 = \nu_2$, we will write $\nu_1^2$ to represent $\nu_1 \otimes \nu_1$.

Note that according to our assumptions, $\mathcal{X}$ is a Polish space equipped with the Borel $\sigma$-algebra $\mathcal{B}(\mathcal{X})$, $\phi_i : \mathcal{X} \times S \to [0, 1]$ is $(\mathcal{B}(\mathcal{X}) \otimes \mathcal{B}(S))/\mathcal{B}([0, 1])$-measurable for each $i \in [d]$, and $e : \mathcal{X} \times S \to [-1, 1]$ is $(\mathcal{B}(\mathcal{X}) \otimes \mathcal{B}(S))/\mathcal{B}([-1, 1])$-measurable.

In the proofs of the main results, we consider an arbitrary probability measure $\mathfrak{m}$ on $(S, \mathcal{B}(S))$. Since there is no ambiguity, for brevity, we omit "$d\mathfrak{m}$" in the notation for integrals. Note that some quantities defined below depend on the chosen probability measure $\mathfrak{m}$.

## B.1    Proofs of Theorem 1 and Proposition 2

In the proofs of Theorem 1 (Appendix B.1.1) and Proposition 2 (Appendix B.1.2), we use the following measure-theoretic treatment of probability spaces. (The notation we use can be found at the beginning of Appendix B.) The underlying probability space for the sample $\{(x^{(j)}, y^{(j)})\}_{j \in \mathbb{N}}$ is $([0,1]^\mathbb{N}, \mathcal{B}([0,1])^\mathbb{N}, \mathbb{P})$, where $[0,1]^\mathbb{N} = \{(\xi^{(1)}, \xi^{(2)}, \dots) : \xi^{(j)} \in [0,1]\}$, and,

$$\mathcal{B}([0,1])^\mathbb{N} := \sigma(\{B_1 \times \cdots \times B_n : B_1, \dots, B_n \in \mathcal{B}([0,1]), n \in \mathbb{N}\})$$

is the $\sigma$-algebra generated by all finite products of Borel sets on $[0, 1]$, and $\mathbb{P}|_{[0,1]^n} = \mathsf{Leb}^n = \otimes_{j=1}^n \mathsf{Leb}$ with $\mathsf{Leb}$ being the Lebesgue measure on $([0,1], \mathcal{B}([0,1]))$. The existence of the above probability space is guaranteed by Kolmogorov's extension theorem. Define the random vector $\Xi = (\Xi^{(j)})_{j \in \mathbb{N}}$ on $([0,1]^\mathbb{N}, \mathcal{B}([0,1])^\mathbb{N})$ to be the identity mapping, i.e., $\Xi : [0,1]^\mathbb{N} \to [0,1]^\mathbb{N}$, $(\xi^{(j)})_{j \in \mathbb{N}} \mapsto (\xi^{(j)})_{j \in \mathbb{N}}$. Then, $\mathbb{P}$ is also the probability measure on $([0,1]^\mathbb{N}, \mathcal{B}([0,1]^\mathbb{N}))$ induced by $\Xi$, and $\Xi$ follows the uniform distribution on $[0,1]^\mathbb{N}$. Suppose $\{(x^{(j)}, y^{(j)})\}_{j \in \mathbb{N}}$ is sampled according to Scheme I with $F$ defined in (1). Then, according to Bogachev (2007, Proposition 10.7.6), for each $j \in \mathbb{N}$, there exist some $(\mathcal{B}(\mathcal{X}) \otimes \mathcal{B}(S))^{j-1} \otimes \mathcal{B}([0,1])/\mathcal{B}(\mathcal{X})$-measurable function,

$$h_X^{(j)} : \ (\mathcal{X} \times S)^{j-1} \times [0,1] \to \mathcal{X},$$

and $(\mathcal{B}(\mathcal{X}) \otimes \mathcal{B}(S))^{j-1} \otimes \mathcal{B}(\mathcal{X}) \otimes \mathcal{B}([0,1])/\mathcal{B}(S)$-measurable function

$$h_Y^{(j)} : \ (\mathcal{X} \times S)^{j-1} \times \mathcal{X} \times [0,1] \to S$$

such that

$$x^{(j)} = h_X^{(j)}(x^{(1)}, y^{(1)}, \dots, x^{(j-1)}, y^{(j-1)}, \Xi^{(2j-1)}),$$
$$y^{(j)} = h_Y^{(j)}(x^{(1)}, y^{(1)}, \dots, x^{(j-1)}, y^{(j-1)}, x^{(j)}, \Xi^{(2j)}),$$

and,

$$\mathbb{E}\left[\mathbb{1}\left\{h_Y^{(j)}(x^{(1)}, y^{(1)}, \dots, x^{(j-1)}, y^{(j-1)}, x^{(j)}, \Xi^{(2j)}) \leqslant t\right\} \big| \mathcal{F}_{j-1}\right] = \theta_*^\top \Phi(x^{(j)}, t) \tag{32}$$

for any $t \in S$ and $j \in \mathbb{N}$, where $\mathcal{F}_j := \sigma\left(\{\Xi^{(k)} : k \in [2j+1]\}\right)$ is the sub $\sigma$-algebra of $\mathcal{B}([0,1])^{\mathbb{N}}$ generated by the random variables $\Xi^{(1)}, \ldots, \Xi^{(2j+1)}$. By definition, we have that $y^{(j)}$ is $\mathcal{F}_j/\mathcal{B}(S)$-measurable for each $j \in \mathbb{N}$ and $\{\mathcal{F}_j\}_{j=0}^{\infty}$ forms a filtration of $([0,1]^{\mathbb{N}}, \mathcal{B}([0,1])^{\mathbb{N}}, \mathbb{P})$. Therefore, $\left\{y^{(j)}\right\}_{j \in \mathbb{N}}$ is $\{\mathcal{F}_j\}_{j \in \mathbb{N}}$-adapted.

By the above construction, for each $j \in \mathbb{N}$, $x^{(j)} : [0,1]^{\mathbb{N}} \to \mathcal{X}$, $\xi \mapsto x^{(j)}(\xi)$ is a $\mathcal{F}_{j-1}/\mathcal{B}(\mathcal{X})$-measurable function. Thus, for each $j \in \mathbb{N}$, the function $\widetilde{h}_X : [0,1]^{\mathbb{N}} \times S \to \mathcal{X} \times S$, $(\xi, t) \mapsto (x^{(j)}(\xi), t)$ is $(\mathcal{F}_{j-1} \otimes \mathcal{B}(S))/(\mathcal{B}(\mathcal{X}) \otimes \mathcal{B}(S))$-measurable. Since $\phi_i : \mathcal{X} \times S \to [0,1]$, $(x,t) \mapsto \phi_i(x,t)$ is $\mathcal{B}(\mathcal{X}) \otimes \mathcal{B}(S)/\mathcal{B}([0,1])$-measurable, we know that $\widetilde{\phi}_i^{(j)} : [0,1]^{\mathbb{N}} \times S \to [0,1]$, $(\xi, t) \mapsto \phi_i(x^{(j)}(\xi), t)$ is $(\mathcal{F}_{j-1} \otimes \mathcal{B}(S))/\mathcal{B}([0,1])$-measurable. Therefore, the vector-valued function $\Phi_j : [0,1]^{\mathbb{N}} \times S \to [0,1]^d$, $(\xi, t) \mapsto [\phi_1(x^{(j)}(\xi), t), \ldots, \phi_d(x^{(j)}(\xi), t)] = [\widetilde{\phi}_1^{(j)}(\xi, t), \ldots, \widetilde{\phi}_d^{(j)}(\xi, t)]$ is $(\mathcal{F}_{j-1} \otimes \mathcal{B}(S))/\mathcal{B}([0,1]^d)$-measurable for each $j \in \mathbb{N}$.

### B.1.1 Proof of Theorem 1

*Proof.* Define $V_j := \int_S \mathrm{I}_{y^{(j)}} \Phi_j - \int_S \theta_*^{\top} \Phi_j \Phi_j$. Since we have proved above that for each $j \in \mathbb{N}$, $y^{(j)}$ is $\mathcal{F}_j$-measurable and the function $S \times S \ni (y,t) \mapsto \mathbb{1}\{y \leq t\} \in [0,1]$ is $\mathcal{B}(S)^2$-measurable, we have that $\mathrm{I}_{y^{(j)}} : [0,1]^{\mathbb{N}} \times S \to [0,1]$, $\mathrm{I}_{y^{(j)}}(\xi, t) = \mathbb{1}\{y^{(j)}(\xi) \leq t\}$ is $\mathcal{F}_j \otimes \mathcal{B}(S)$-measurable. Since we have also proved above that for each $j \in \mathbb{N}$, $\Phi_j$ is $\mathcal{F}_{j-1} \otimes \mathcal{B}(S)$-measurable, by Fubini's theorem and (32), we have that $\int_S \theta_*^{\top} \Phi_j \Phi_j$ is $\mathcal{F}_{j-1}$-measurable, $V_j$ is $\mathcal{F}_j$-measurable, and

$$
\begin{aligned}
\mathbb{E}[V_j | \mathcal{F}_{j-1}] &= \mathbb{E}\left[\int_S \mathrm{I}_{y^{(j)}} \Phi_j \Big| \mathcal{F}_{j-1}\right] - \int_S \theta_*^{\top} \Phi_j \Phi_j \\
&= \int_S \mathbb{E}\left[\mathrm{I}_{y^{(j)}} \Big| \mathcal{F}_{j-1}\right] \Phi_j - \int_S \theta_*^{\top} \Phi_j \Phi_j \\
&= \int_S \theta_*^{\top} \Phi_j \Phi_j - \int_S \theta_*^{\top} \Phi_j \Phi_j \\
&= 0.
\end{aligned}
\tag{33}
$$

For any $\alpha \in \mathbb{R}^d$, define $M_0(\alpha) = 1$. Then, $M_0(\alpha)$ is $\mathcal{F}_0$-measurable for any $\alpha \in \mathbb{R}^d$. For $n \in \mathbb{N}$, define $M_n(\alpha) := \exp\left\{\alpha^{\top} W_n - \frac{1}{2}\|\alpha\|_{U_n}^2\right\}$ with $W_n := \sum_{j=1}^n V_j$ and $U_n = \sum_{j=1}^n \int_S \Phi_j \Phi_j^{\top}$. Since $\Phi_j$ is $\mathcal{F}_{j-1} \otimes \mathcal{B}(S)$-measurable and $V_j$ is $\mathcal{F}_j$-measurable, by Fubini's theorem, $U_n$ is $\mathcal{F}_{n-1}$-measurable and $W_n$ is $\mathcal{F}_n$-measurable for each $n \in \mathbb{N}$. Thus, $M_n(\alpha)$ is also $\mathcal{F}_n$-measurable for any $\alpha \in \mathbb{R}^d$ and $n \in \mathbb{N}$. Moreover, note that the function $\mathbb{R}^d \times \mathbb{R}^d \times \mathbb{R}^{d \times d} \to (0, \infty)$, $(\alpha, W, U) \mapsto \exp\left\{\alpha^{\top} W - \frac{1}{2}\|\alpha\|_U^2\right\}$ is measurable. Hence, $M_n : [0,1]^{\mathbb{N}} \times \mathbb{R}^d \to (0, \infty)$, $(\xi, \alpha) \mapsto \exp\left\{\alpha^{\top} W_n(\xi) - \frac{1}{2}\|\alpha\|_{U_n(\xi)}^2\right\}$ is $\mathcal{F}_n \otimes \mathcal{B}(\mathbb{R}^d)$-measurable. Thus, for any $\alpha \in \mathbb{R}^d$, $\{M_n(\alpha)\}_{n \geq 0}$ is $\{\mathcal{F}_n\}_{n \geq 0}$-adapted. Besides, for any $\alpha \in \mathbb{R}^d$ and $n \in \mathbb{N}$, we have

$$
\begin{aligned}
\mathbb{E}[M_n(\alpha) | \mathcal{F}_{n-1}] &= M_{n-1}(\alpha) \mathbb{E}\left[\exp\left\{\alpha^{\top} V_n - \frac{1}{2}\alpha^{\top}\left(\int_S \Phi_n \Phi_n^{\top}\right)\alpha\right\} \Big| \mathcal{F}_{n-1}\right] \\
&= M_{n-1}(\alpha) \frac{\mathbb{E}\left[\exp\left\{\alpha^{\top} V_n\right\} | \mathcal{F}_{n-1}\right]}{\exp\left\{\frac{1}{2}\int_S (\alpha^{\top} \Phi_n)^2\right\}}.
\end{aligned}
\tag{34}
$$

Since $-\int_S |\alpha^{\top} \Phi_n| \leq \alpha^{\top} V_n \leq \int_S |\alpha^{\top} \Phi_n|$ almost surely (a.s.), we have

$$
\mathbb{E}\left[\exp\left\{\alpha^{\top} V_n\right\} | \mathcal{F}_{n-1}\right] \leq \exp\left\{\frac{4}{8}\left(\int_S |\alpha^{\top} \Phi_n|\right)^2\right\}
\tag{35}
$$

$$
\leq \exp\left\{\frac{1}{2}\int_S (\alpha^{\top} \Phi_n)^2\right\},
\tag{36}
$$

where (35) follows from Hoeffding's lemma (Hoeffding, 1963), and (36) follows from the Cauchy-Schwarz inequality and the fact that $\int_S 1 = \mathfrak{m}(S) = 1$. Then, by (34) and (36), we have

$$
\mathbb{E}[M_n(\alpha) | \mathcal{F}_{n-1}] \leq M_{n-1}(\alpha) \frac{\exp\left\{\frac{1}{2}\int_S (\alpha^{\top} \Phi_n)^2\right\}}{\exp\left\{\frac{1}{2}\int_S (\alpha^{\top} \Phi_n)^2\right\}} = M_{n-1}(\alpha).
\tag{37}
$$

Since $M_0(\alpha) = 1$ and $M_n(\alpha) \geqslant 0$, for any $\alpha \in \mathbb{R}^d$, $\{M_n(\alpha)\}_{n \geqslant 0}$ is a super-martingale.

Now for any $n \geqslant 0$, define $\overline{M}_n := \int_{\mathbb{R}^d} M_n(\alpha) h(\alpha) d\alpha$, with $d\alpha$ denoting $\mathsf{Leb}(d\alpha)$ where the Lebesgue measure is on $(\mathbb{R}^d, \mathcal{B}(\mathbb{R}^d))$ and

$$h(\alpha) = \left(\frac{\lambda}{2\pi}\right)^{\frac{d}{2}} \exp\left\{-\frac{\lambda}{2}\alpha^\top \alpha\right\} = \left(\frac{\lambda}{2\pi}\right)^{\frac{d}{2}} \exp\left\{-\frac{1}{2}\|\alpha\|_{\lambda I_d}^2\right\}. \tag{38}$$

Recall that $U_n(\lambda) = U_n + \lambda I_d$. Then, for $n \geqslant 1$, we have

$$\begin{aligned}
\overline{M}_n &= \left(\frac{\lambda}{2\pi}\right)^{\frac{d}{2}} \int_{\mathbb{R}^d} \exp\left\{\alpha^\top W_n - \frac{1}{2}\|\alpha\|_{U_n}^2 - \frac{1}{2}\|\alpha\|_{\lambda I_d}^2\right\} d\alpha \\
&= \left(\frac{\lambda}{2\pi}\right)^{\frac{d}{2}} \int_{\mathbb{R}^d} \exp\left\{\frac{1}{2}\|W_n\|_{U_n(\lambda)^{-1}}^2 - \frac{1}{2}\|\alpha - U_n(\lambda)^{-1}W_n\|_{U_n(\lambda)}^2\right\} d\alpha \tag{39} \\
&= \frac{\lambda^{\frac{d}{2}}}{\det(U_n(\lambda))^{\frac{1}{2}}} \exp\left(\frac{1}{2}\|W_n\|_{U_n(\lambda)^{-1}}^2\right) \cdot \\
&\qquad \frac{1}{(2\pi)^{\frac{d}{2}} \det(U_n(\lambda))^{-\frac{1}{2}}} \int_{\mathbb{R}^d} \exp\left\{-\frac{1}{2}\|\alpha - U_n(\lambda)^{-1}W_n\|_{U_n(\lambda)}^2\right\} d\alpha \\
&= \frac{\lambda^{\frac{d}{2}}}{\det(U_n(\lambda))^{\frac{1}{2}}} \exp\left(\frac{1}{2}\|W_n\|_{U_n(\lambda)^{-1}}^2\right), \tag{40}
\end{aligned}$$

where (39) follows from the calculation below:

$$\begin{aligned}
&\|W_n\|_{U_n(\lambda)^{-1}}^2 - \|\alpha - U_n(\lambda)^{-1}W_n\|_{U_n(\lambda)}^2 \\
&= \|W_n\|_{U_n(\lambda)^{-1}}^2 - \left(\alpha^\top - W_n^\top U_n(\lambda)^{-1}\right) U_n(\lambda) \left(\alpha - U_n(\lambda)^{-1}W_n\right) \\
&= \|W_n\|_{U_n(\lambda)^{-1}}^2 - \|\alpha\|_{U_n(\lambda)} - \|W_n\|_{U_n(\lambda)^{-1}}^2 + 2\alpha^\top W_n \\
&= 2\alpha^\top W_n - \|\alpha\|_{\lambda I_d} - \|\alpha\|_{U_n}. \tag{41}
\end{aligned}$$

For $n = 0$, $\overline{M}_0 = \int_{\mathbb{R}^d} M_0(\alpha) h(\alpha) d\alpha = \int_{\mathbb{R}^d} h(\alpha) d\alpha = 1$.

Moreover, since we have shown that $M_n$ is $\mathcal{F}_n \otimes \mathcal{B}(\mathbb{R}^d)$-measurable, by Fubini's theorem and (37), $\overline{M}_n$ is $\mathcal{F}_n$-measurable for any $n \geqslant 0$ and for any $n \in \mathbb{N}$,

$$\begin{aligned}
\mathbb{E}\left[\overline{M}_n | \mathcal{F}_{n-1}\right] &= \mathbb{E}\left[\int_{\mathbb{R}^d} M_n(\alpha) h(\alpha) d\alpha \Big| \mathcal{F}_{n-1}\right] \\
&= \int_{\mathbb{R}^d} \mathbb{E}\left[M_n(\alpha) | \mathcal{F}_{n-1}\right] h(\alpha) d\alpha \\
&\leqslant \int_{\mathbb{R}^d} M_{n-1}(\alpha) h(\alpha) d\alpha \\
&= \overline{M}_{n-1}. \tag{42}
\end{aligned}$$

Thus, $\{\overline{M}_n\}_{n \geqslant 0}$ is also a super-martingale. By Doob's maximal inequality for super-martingales,

$$\mathbb{P}\left[\sup_{n \in \mathbb{N}} \overline{M}_n \geqslant \delta\right] \leqslant \frac{\mathbb{E}[\overline{M}_0]}{\delta} = \frac{1}{\delta}$$

which, together with (40), implies that

$$\mathbb{P}\left[\exists n \in \mathbb{N} \text{ s.t. } \|W_n\|_{U_n(\lambda)^{-1}} \geqslant \sqrt{\log\frac{\det(U_n(\lambda))}{\lambda^d} + 2\log\frac{1}{\delta}}\right] \leqslant \delta. \tag{43}$$

Since

$$\theta_* = \left(\sum_{j=1}^n \int_S \Phi_j \Phi_j^\top + \lambda I_d\right)^{-1} \left(\sum_{j=1}^n \int_S \Phi_j \Phi_j^\top \theta_* + \lambda \theta_*\right), \tag{44}$$

by (3), we have

$$\widehat{\theta}_\lambda - \theta_* = U_n(\lambda)^{-1} \left(\sum_{j=1}^n V_j - \lambda \theta_*\right) = U_n(\lambda)^{-1} W_n - U_n(\lambda)^{-1}(\lambda \theta_*).$$

Thus, by the triangle inequality,

$$\begin{aligned}
\|\widehat{\theta}_\lambda - \theta_*\|_{U_n(\lambda)} &\leq \|U_n(\lambda)^{-1} W_n\|_{U_n(\lambda)} + \lambda \|U_n(\lambda)^{-1}\theta_*\|_{U_n(\lambda)} \\
&= \|W_n\|_{U_n(\lambda)^{-1}} + \|\lambda\theta_*\|_{U_n(\lambda)^{-1}} \\
&\leq \|W_n\|_{U_n(\lambda)^{-1}} + \sqrt{\lambda}\|\theta_*\|,
\end{aligned} \tag{45}$$

where the last inequality follows from the facts that $U_n(\lambda)^{-1} = \frac{1}{\lambda}\left(I - U_n(\lambda)^{-1}U_n\right)$ and $\|I - U_n(\lambda)^{-1}U_n\|_2 \leq 1$. By (43) and (45), with probability at least $1 - \delta$, for all $n \in \mathbb{N}$, we have

$$\|\widehat{\theta}_\lambda - \theta_*\|_{U_n(\lambda)} \leq \sqrt{\log \frac{\det(U_n(\lambda))}{\lambda^d} + 2\log \frac{1}{\delta}} + \sqrt{\lambda}\|\theta_*\|. \tag{46}$$

By the arithmetic mean-geometric mean (AM–GM) inequality, we have

$$\begin{aligned}
\log \det(U_n(\lambda)) &\leq d \log \left(\frac{\operatorname{trace}(U_n(\lambda))}{d}\right) \\
&= d \log \left(\frac{1}{d}\operatorname{trace}\left(\sum_{j=1}^n \int_S \Phi_j \Phi_j^\top + \lambda I_d\right)\right).
\end{aligned}$$

Since

$$\begin{aligned}
\operatorname{trace}\left(\sum_{j=1}^n \int_S \Phi_j \Phi_j^\top + \lambda I_d\right) &= d\lambda + \sum_{j=1}^n \int_S \operatorname{trace}\left(\Phi_j \Phi_j^\top\right) \\
&= d\lambda + \sum_{j=1}^n \int_S \|\Phi_j\|_2^2 \\
&\leq d\lambda + nd,
\end{aligned}$$

we have

$$\log \det(U_n(\lambda)) \leq d \log \left(\frac{1}{d}(d\lambda + nd)\right) = d \log (\lambda + n). \tag{47}$$

By (46) and (47), for any $\lambda > 0$, $\delta \in (0,1)$, with probability at least $1 - \delta$, for all $n \in \mathbb{N}$, we have

$$\|\widehat{\theta}_\lambda - \theta_*\|_{U_n(\lambda)} \leq \sqrt{d \log \left(1 + \frac{n}{\lambda}\right) + 2\log \frac{1}{\delta}} + \sqrt{\lambda}\|\theta_*\|. \tag{48}$$

Thus, Theorem 1 is proved for any probability measure $\mathfrak{m}$ on $(S, \mathcal{B}(S))$. $\qquad\square$

### B.1.2   Proof of Proposition 2

*Proof.* When $U_N$ is non-singular for some fixed $N \in \mathbb{N}$, since $\int_S \Phi_j \Phi_j^\top$ is positive semi-definite for any $j \in \mathbb{N}$, it immediately follows that $U_n$ are non-singular for any $n \geq N$. Then, $\widehat{\theta}$ is unique and is given by (3) with $\lambda = 0$ for any $n \geq N$, i.e.,

$$\widehat{\theta} = \left(\sum_{j=1}^n \int_S \Phi_j \Phi_j^\top\right)^{-1} \left(\sum_{j=1}^n \int_S \mathrm{I}_{y^{(j)}} \Phi_j\right)$$

for any $n \geqslant N$. Since

$$\theta_* = \left( \sum_{j=1}^{n} \int_S \Phi_j \Phi_j^\top \right)^{-1} \left( \sum_{j=1}^{n} \int_S \Phi_j \Phi_j^\top \theta_* \right), \tag{49}$$

we have

$$\widehat{\theta} - \theta_* = U_n^{-1} W_n. \tag{50}$$

By definition and the triangle inequality for integrals, we have

$$\|V_j\| \leqslant \int_S |\mathrm{I}_{y^{(j)}} - \theta_*^\top \Phi_j| \|\Phi_j\| \leqslant \int_S \sqrt{d} = \sqrt{d}, \tag{51}$$

which also implies that

$$\sum_{j=1}^{n} \mathbb{E}[\|V_j\|^2 | \mathcal{F}_{j-1}] \leqslant \sum_{j=1}^{n} d = nd. \tag{52}$$

Since $W_n = \sum_{j=1}^{n} V_j$, by (33), (51), (52), and Hsu et al. (2012a, Proposition 1.2), we have

$$\mathbb{P}[\|W_n\| \geqslant \sqrt{nd} + \sqrt{8nda} + (4/3)\sqrt{d}a] \leqslant e^{-a}$$

for any $a > 0$. Thus, for any $\delta \in (0,1)$ and $n \in \mathbb{N}$, with probability at least $1 - \delta$, we have

$$\|W_n\| \leqslant \sqrt{nd} + \sqrt{8nd \log \frac{1}{\delta}} + \frac{4}{3}\sqrt{d} \log \frac{1}{\delta}. \tag{53}$$

Since $U_n$ is positive definite, by (53), we have

$$\|W_n\|_{U_n^{-1}} = \sqrt{W_n^\top U_n^{-1} W_n} \leqslant \frac{\|W_n\|}{\sqrt{\mu_{\min}(U_n)}} \leqslant \frac{\sqrt{nd} + \sqrt{8nd \log \frac{1}{\delta}} + \frac{4}{3}\sqrt{d} \log \frac{1}{\delta}}{\sqrt{\mu_{\min}(U_n)}} \tag{54}$$

with probability at least $1 - \delta$. Hence, by (50), and (54), we have that for any $n \geqslant N$,

$$\|\widehat{\theta} - \theta_*\|_{U_n} = \|U_n^{-1} W_n\|_{U_n} = \|W_n\|_{U_n^{-1}} \leqslant \frac{\sqrt{nd} + \sqrt{8nd \log \frac{1}{\delta}} + \frac{4}{3}\sqrt{d} \log \frac{1}{\delta}}{\sqrt{\mu_{\min}(U_n)}}$$

with probability at least $1 - \delta$. In conclusion, Proposition 2 is proved for any probability measure $\mathfrak{m}$ on $(S, \mathcal{B}(S))$.

$$\square$$

## B.2 Proofs of Theorem 4 and Proposition 5

In this section, we follow the same construction of the probability space as in Appendix B.1. In particular, noting that Scheme II is a special case of Scheme I, we consider the underlying probability space for the sample $\{(x^{(j)}, y^{(j)})\}_{j \in \mathbb{N}}$ to be $([0,1]^\mathbb{N}, \mathcal{B}([0,1])^\mathbb{N}, \mathbb{P})$. Define the random vector $\Xi$ to be the identity mapping from $[0,1]^\mathbb{N}$ onto itself as in Appendix B.1. Then, $\Xi$ follows the uniform distribution on $[0,1]^\mathbb{N}$. Suppose $\{(x^{(j)}, y^{(j)})\}_{j \in \mathbb{N}}$ is sampled according to Scheme II with $F$ defined in (1). Then, according to Bogachev (2007, Proposition 10.7.6), for each $j \in \mathbb{N}$, there exist some $\mathcal{B}([0,1])/\mathcal{B}(\mathcal{X})$-measurable function $h_X^{(j)} : [0,1] \to \mathcal{X}$ and $\mathcal{B}(\mathcal{X}) \otimes \mathcal{B}([0,1])/\mathcal{B}(S)$-measurable function $h_Y^{(j)} : \mathcal{X} \times [0,1] \to S$ such that $x^{(j)} = h_X^{(j)}(\Xi^{(2j-1)})$, $y^{(j)} = h_Y^{(j)}(x^{(j)}, \Xi^{(2j)})$, and

$$\mathbb{E}\left[ \mathbb{1}\left\{ h_Y^{(j)}(x^{(j)}, \Xi^{(2j)}) \leqslant t \right\} \middle| \mathcal{F}_{j-1} \right] = \theta_*^\top \Phi(x^{(j)}, t)$$

for any $t \in S$ and $j \in \mathbb{N}$, where $\mathcal{F}_j := \sigma\left( \{\Xi^{(k)} : k \in [2j+1]\} \right)$ is the sub $\sigma$-algebra of $\mathcal{B}([0,1])^\mathbb{N}$ generated by the random variables $\Xi^{(1)}, \ldots, \Xi^{(2j+1)}$. With the same proof provided at the beginning of Appendix B.1, $\{y^{(j)}\}_{j \in \mathbb{N}}$ is $\{\mathcal{F}_j\}_{j \in \mathbb{N}}$-adapted and $\Phi_j$ is $(\mathcal{F}_{j-1} \otimes \mathcal{B}(S))/\mathcal{B}([0,1]^d)$-measurable for each $j \in \mathbb{N}$. Moreover, $\{x^{(j)}\}_{j \in \mathbb{N}}$ is independent, which implies that $\{\Phi_j(t)\}_{j \in \mathbb{N}}$ is independent for any $t \in S$ and $\{y^{(j)}\}_{j \in \mathbb{N}}$ is independent.

### B.2.1 Proof of Theorem 4

*Proof.* By definition and Fubini's theorem, we have $\Sigma^{(j)} = \mathbb{E}\left[\int_S \Phi_j \Phi_j^\top\right] = \int_S \mathbb{E}\left[\Phi_j \Phi_j^\top\right]$ for each $j \in [n]$, $\Sigma_n = \sum_{j=1}^n \Sigma^{(j)} = \mathbb{E}\left[U_n\right]$.

For the proof, we need to define $\Delta_n := \Sigma_n^{-\frac{1}{2}}\left(U_n - \Sigma_n\right)\Sigma_n^{-\frac{1}{2}}$, $\widetilde{\Sigma}_n^{(j)} := \Sigma_n^{-\frac{1}{2}}\Sigma^{(j)}\Sigma_n^{-\frac{1}{2}}$, and $\widetilde{\Phi}_j(t) := \Sigma_n^{-\frac{1}{2}}\Phi_j(t)$ for any $t \in \mathbb{R}$ and $j \in [n]$. For any $j \in \mathbb{N}$, we have

$$\|\Sigma^{(j)}\|_2 = \mu_{\max}\left(\Sigma^{(j)}\right) = \mu_{\max}\left(\mathbb{E}\left[\int_S \Phi_j \Phi_j^\top\right]\right) \leqslant \mathbb{E}\left[\int_S \|\Phi_j\|_2^2\right] \leqslant d. \tag{55}$$

By the assumption that $\mu_{\min}(\Sigma^{(j)}) \geqslant \sigma_{\min}$ for all $j \in \mathbb{N}$ and Weyl's inequality (Weyl, 1912), we have

$$\mu_{\min}\left(\Sigma_n\right) \geqslant n\sigma_{\min}. \tag{56}$$

By (55) and (56), for each $j \in [n]$, we have

$$\mu_{\max}\left(\widetilde{\Sigma}_n^{(j)}\right) \leqslant \frac{\mu_{\max}\left(\Sigma^{(j)}\right)}{\mu_{\min}\left(\Sigma_n\right)} \leqslant \frac{d}{n\sigma_{\min}}. \tag{57}$$

Consider the following random matrix for $j \in [n]$:

$$Z_j := \int_S \widetilde{\Phi}_j \widetilde{\Phi}_j^\top - \widetilde{\Sigma}_n^{(j)} = \Sigma_n^{-\frac{1}{2}}\left(\int_S \Phi_j \Phi_j^\top - \Sigma^{(j)}\right)\Sigma_n^{-\frac{1}{2}}.$$

We have that

$$\Delta_n = \sum_{j=1}^n Z_j, \tag{58}$$

and for any $j \in [n]$, we have,

$$\mathbb{E}[Z_j] = 0, \tag{59}$$

and, furthermore, we have,

$$\begin{aligned}
\|Z_j\|_2 &= \max\{\mu_{\max}(Z_j), -\mu_{\min}(Z_j)\} \\
&\leqslant \max\left\{\mu_{\max}\left(\int_S \widetilde{\Phi}_j \widetilde{\Phi}_j^\top\right), \mu_{\max}\left(\widetilde{\Sigma}_n^{(j)}\right)\right\} \\
&\leqslant \frac{d}{n\sigma_{\min}}
\end{aligned} \tag{60}$$

where (60) follows from (57) and

$$\mu_{\max}\left(\int_S \widetilde{\Phi}_j \widetilde{\Phi}_j^\top\right) \leqslant \int_S \|\widetilde{\Phi}_j\|^2 \leqslant \frac{1}{\mu_{\min}(\Sigma_n)}\int_S \|\Phi_j\|^2 \leqslant \frac{d}{n\sigma_{\min}}.$$

By (58), (59), (60), and Tropp (2012, Theorem 1.3), we have

$$\mathbb{P}\left[\mu_{\min}\left(\Delta_n\right) \leqslant -a\right] \leqslant d\exp\left(-\frac{n\sigma_{\min}^2 a^2}{8d^2}\right) \tag{61}$$

for any $a \geqslant 0$. Thus, with probability at least $1 - \delta$,

$$\mu_{\min}\left(\Delta_n\right) \geqslant -\frac{d}{\sigma_{\min}}\sqrt{\frac{8}{n}\log\left(\frac{d}{\delta}\right)}. \tag{62}$$

Since $\Delta_n = \Sigma_n^{-\frac{1}{2}} U_n \Sigma_n^{-\frac{1}{2}} - I_d$, we have $\mu_{\min}(\Sigma_n^{-\frac{1}{2}} U_n \Sigma_n^{-\frac{1}{2}}) = \mu_{\min}(\Delta_n) + 1$ which together with the fact that $U_n = \Sigma_n^{\frac{1}{2}} \Sigma_n^{-\frac{1}{2}} U_n \Sigma_n^{-\frac{1}{2}} \Sigma_n^{\frac{1}{2}}$ implies that

$$\mu_{\min}(U_n) \geqslant \mu_{\min}(\Sigma_n) \mu_{\min}(\Sigma_n^{-\frac{1}{2}} U_n \Sigma_n^{-\frac{1}{2}}) = \mu_{\min}(\Sigma_n)(\mu_{\min}(\Delta_n) + 1). \tag{63}$$

By (63), we have

$$\mu_{\min}(\Delta_n) \geqslant -\frac{1}{2} \implies \mu_{\min}(U_n) \geqslant \frac{1}{2}\mu_{\min}(\Sigma_n) \geqslant \frac{n}{2}\sigma_{\min} > 0. \tag{64}$$

Note that when $U_n$ is positive definite, we have

$$\Sigma_n^{\frac{1}{2}} U_n^{-1} \Sigma_n^{\frac{1}{2}} = \Sigma_n^{\frac{1}{2}} U_n^{-\frac{1}{2}} \left(\Sigma_n^{\frac{1}{2}} U_n^{-\frac{1}{2}}\right)^\top,$$

$$U_n^{-\frac{1}{2}} \Sigma_n U_n^{-\frac{1}{2}} = \left(\Sigma_n^{\frac{1}{2}} U_n^{-\frac{1}{2}}\right)^\top \Sigma_n^{\frac{1}{2}} U_n^{-\frac{1}{2}}.$$

Thus,

$$\begin{aligned}
\|U_n^{-\frac{1}{2}} \Sigma_n U_n^{-\frac{1}{2}}\|_2 &= \|\Sigma_n^{\frac{1}{2}} U_n^{-1} \Sigma_n^{\frac{1}{2}}\|_2 \\
&= \left\|\left(\Sigma_n^{-\frac{1}{2}} U_n \Sigma_n^{-\frac{1}{2}}\right)^{-1}\right\|_2 \\
&= \|(I_d + \Delta_n)^{-1}\|_2 \\
&= \frac{1}{\mu_{\min}(I_d + \Delta_n)} \\
&= \frac{1}{1 + \mu_{\min}(\Delta_n)}.
\end{aligned} \tag{65}$$

By (64) and (65), we have

$$\mu_{\min}(\Delta_n) \geqslant -\frac{1}{2} \implies \mu_{\min}(U_n) \geqslant \frac{n}{2}\sigma_{\min} \text{ and } \|U_n^{-\frac{1}{2}} \Sigma_n U_n^{-\frac{1}{2}}\|_2 \leqslant 2. \tag{66}$$

By (62), for any $\delta \in (0, 1)$, if $n \geqslant \frac{32 d^2}{\sigma_{\min}^2} \log(d/\delta)$, we have $\mu_{\min}(\Delta_n) \geqslant -\frac{1}{2}$ with probability at least $1 - \delta$. Then, by (66), we have

$$\|U_n^{-\frac{1}{2}} \Sigma_n U_n^{-\frac{1}{2}}\|_2 \leqslant 2 \text{ and } \mu_{\min}(U_n) \geqslant \frac{n}{2}\sigma_{\min}. \tag{67}$$

with probability at least $1 - \delta$.

Still define $W_n := \sum_{i=1}^n \left(\int_S \mathrm{I}_{y^{(j)}} \Phi_j - \int_S \theta_*^\top \Phi_j \Phi_j\right)$. By (50), we have $\hat{\theta} - \theta_* = U_n^{-1} W_n$ and $\|\hat{\theta} - \theta_*\|_{U_n} = \|W_n\|_{U_n^{-1}}$.

By (7), (67), and the union bound, for any $\delta_1 \in (0,1)$ and $\delta_2 \in (0, 1 - \delta_1)$, if $n \geq \frac{32d^2}{\sigma_{\min}^2} \log \frac{d}{\delta_1}$, we have

$$
\begin{aligned}
\|\widehat{\theta} - \theta_*\|_{\Sigma_n} &= \sqrt{W_n^\top U_n^{-\frac{1}{2}} U_n^{-\frac{1}{2}} \Sigma_n U_n^{-\frac{1}{2}} U_n^{-\frac{1}{2}} W_n} \\
&\leq \sqrt{\|U_n^{-\frac{1}{2}} \Sigma_n U_n^{-\frac{1}{2}}\|_2 \|W_n\|_{U_n^{-1}}^2} \\
&= \sqrt{\|U_n^{-\frac{1}{2}} \Sigma_n U_n^{-\frac{1}{2}}\|_2 \|\widehat{\theta} - \theta_*\|_{U_n}^2} \\
&\leq \sqrt{2} \|\widehat{\theta} - \theta_*\|_{U_n} \\
&\leq \frac{\sqrt{2nd} + 4\sqrt{nd \log \frac{1}{\delta_2}} + \frac{4}{3}\sqrt{2d} \log \frac{1}{\delta_2}}{\sqrt{\mu_{\min}(U_n)}} \\
&\leq \frac{2\sqrt{nd} + 4\sqrt{2nd \log \frac{1}{\delta_2}} + \frac{8}{3}\sqrt{d} \log \frac{1}{\delta_2}}{\sqrt{n\sigma_{\min}}} \\
&= \frac{2\sqrt{d} + 4\sqrt{2d \log \frac{1}{\delta_2}} + \frac{8}{3}\sqrt{d/n} \log \frac{1}{\delta_2}}{\sqrt{\sigma_{\min}}}
\end{aligned}
$$

with probability at least $1 - \delta_1 - \delta_2$.

By letting $\delta_1 = \delta_2 = \delta$, (8) is proved. In conclusion, Theorem 4 is proved for any probability measure $\mathfrak{m}$ on $(S, \mathcal{B}(S))$. $\qquad\square$

### B.2.2  Proof of Proposition 5

*Proof.* Since

$$
\Sigma_n^{-\frac{1}{2}} U_n(\lambda) \Sigma_n^{-\frac{1}{2}} = \Sigma_n^{-\frac{1}{2}} \left( \Sigma_n + \lambda I_d + U_n - \Sigma_n \right) \Sigma_n^{-\frac{1}{2}} = I_d + \lambda \Sigma_n^{-1} + \Delta_n
$$

and $\lambda \Sigma_n^{-1}$ is positive semi-definite for any $\lambda \geq 0$, we have

$$
\begin{aligned}
\|\Sigma_n^{\frac{1}{2}} U_n(\lambda)^{-1} \Sigma_n^{\frac{1}{2}}\|_2 &= \left\| \left( \Sigma_n^{-\frac{1}{2}} U_n(\lambda) \Sigma_n^{-\frac{1}{2}} \right)^{-1} \right\|_2 \\
&= \|(I_d + \lambda \Sigma_n^{-1} + \Delta_n)^{-1}\|_2 \\
&= \frac{1}{\mu_{\min}(I_d + \lambda \Sigma_n^{-1} + \Delta_n)} \\
&\leq \frac{1}{1 + \mu_{\min}(\Delta_n)}.
\end{aligned}
\tag{68}
$$

Since

$$
\begin{aligned}
\Sigma_n^{\frac{1}{2}} U_n(\lambda)^{-1} \Sigma_n^{\frac{1}{2}} &= \Sigma_n^{\frac{1}{2}} U_n(\lambda)^{-\frac{1}{2}} \left( \Sigma_n^{\frac{1}{2}} U_n(\lambda)^{-\frac{1}{2}} \right)^\top, \\
U_n(\lambda)^{-\frac{1}{2}} \Sigma_n U_n(\lambda)^{-\frac{1}{2}} &= \left( \Sigma_n^{\frac{1}{2}} U_n(\lambda)^{-\frac{1}{2}} \right)^\top \Sigma_n^{\frac{1}{2}} U_n(\lambda)^{-\frac{1}{2}},
\end{aligned}
$$

by (68), we have

$$
\|U_n(\lambda)^{-\frac{1}{2}} \Sigma_n U_n(\lambda)^{-\frac{1}{2}}\|_2 = \|\Sigma_n^{\frac{1}{2}} U_n(\lambda)^{-1} \Sigma_n^{\frac{1}{2}}\|_2 \leq \frac{1}{1 + \mu_{\min}(\Delta_n)}.
$$

Define $R_n = \sum_{i=1}^n V_j - \lambda \theta_*$ where $V_j := \int_S \mathbb{I}_{y^{(j)}} \Phi_j - \int_S \theta_*^\top \Phi_j \Phi_j$. Then, by (3) and (44), we have

$$
\widehat{\theta}_\lambda - \theta_* = U_n(\lambda)^{-1} R_n.
$$

Thus,

$$
\begin{aligned}
\|\widehat{\theta}_\lambda - \theta_*\|_{\Sigma_n}^2 &= R_n^\top U_n(\lambda)^{-\frac{1}{2}} U_n(\lambda)^{-\frac{1}{2}} \Sigma_n U_n(\lambda)^{-\frac{1}{2}} U_n(\lambda)^{-\frac{1}{2}} R_n \\
&\leqslant \|U_n(\lambda)^{-\frac{1}{2}} \Sigma_n U_n(\lambda)^{-\frac{1}{2}}\|_2 \|R_n\|_{U_n(\lambda)^{-1}}^2 \\
&= \|U_n(\lambda)^{-\frac{1}{2}} \Sigma_n U_n(\lambda)^{-\frac{1}{2}}\|_2 \|\widehat{\theta}_\lambda - \theta_*\|_{U_n(\lambda)}^2.
\end{aligned}
$$

By the above inequality, (62), and (6) in Theorem 1, for any $n \in \mathbb{N}$, $\delta_1 \in (0,1)$, $\delta_2 \in (0, 1 - \delta_1)$, we have

$$
\begin{aligned}
\|\widehat{\theta}_\lambda - \theta_*\|_{\Sigma_n} &\leqslant \frac{\|\widehat{\theta}_\lambda - \theta_*\|_{U_n(\lambda)}}{\sqrt{1 + \mu_{\min}(\Delta_n)}} \\
&\leqslant \frac{\sqrt{d \log\left(1 + \frac{n}{\lambda}\right) + 2 \log \frac{1}{\delta_2}} + \sqrt{\lambda}\|\theta_*\|}{\sqrt{1 - \frac{d}{\sigma_{\min}} \sqrt{\frac{8}{n} \log\left(\frac{d}{\delta}\right)}}}
\end{aligned}
$$

with probability at least $1 - \delta_1 - \delta_2$. Then, when $n \geqslant \frac{32 d^2}{\sigma_{\min}^2} \log(d/\delta_1)$, by the above inequality, we have

$$
\|\widehat{\theta}_\lambda - \theta\|_{\Sigma_n} \leqslant \sqrt{2 \left( d \log\left(1 + \frac{n}{\lambda}\right) + 2 \log \frac{1}{\delta_2} \right)} + \sqrt{2\lambda}\|\theta_*\| \tag{69}
$$

with probability at least $1 - \delta_1 - \delta_2$. Thus, (9) is obtained from (69) by setting $\delta_1 = \delta_2 = \delta \in (0, 1/2)$. Proposition 5 is proved for any probability measure $\mathfrak{m}$ on $(S, \mathcal{B}(S))$. □

## B.3 Proof of Theorem 7

*Proof.* Notice that by Fubini's theorem, we have $\mathbb{E}[U_n(\lambda)] = \Sigma_n(\lambda)$ and

$$
\mathbb{E}[u_n] = \mathbb{E}\left[ \sum_{j=1}^n \int_S \Phi_j \Phi_j^\top \theta_* d\mathfrak{m} \right] = \mathbb{E}\left[ \sum_{j=1}^n \int_S \Phi_j \Phi_j^\top d\mathfrak{m} \right] \theta_* = \Sigma_n \theta_*.
$$

Similar to the proof of Azizzadenesheli (2020, Lemma 4.3), by Pires & Szepesvári (2012, Theorem 3.4), we have that with probability at least $1 - \delta$,

$$
\begin{aligned}
\|\Sigma_n(\lambda)\breve{\theta}_\lambda - \Sigma_n \theta_*\| &\leqslant \|\Sigma_n(\lambda)\theta_* - \Sigma_n \theta_*\| + 2\Delta_n^U(\delta)\|\theta_*\| + 2\|u_n - \mathbb{E}[u_n]\| \\
&= (\lambda + 2\Delta_n^U(\delta))\|\theta_*\| + 2\|u_n - \mathbb{E}[u_n]\|.
\end{aligned}
$$

Since

$$
\|\Sigma_n(\lambda)\breve{\theta}_\lambda - \Sigma_n \theta_*\| = \|\Sigma_n(\breve{\theta}_\lambda - \theta_*) + \lambda \breve{\theta}_\lambda\|,
$$

we have

$$
\|\Sigma_n(\breve{\theta}_\lambda - \theta_*)\| \leqslant \lambda \|\breve{\theta}_\lambda\| + (\lambda + 2\Delta_n^U(\delta))\|\theta_*\| + 2\|u_n - \mathbb{E}[u_n]\|
$$

which also implies that

$$
\|\breve{\theta}_\lambda - \theta_*\| \leqslant \frac{1}{\mu_{\min}(\Sigma_n)} \left[ \lambda \|\breve{\theta}_\lambda\| + (\lambda + 2\Delta_n^U(\delta))\|\theta_*\| + 2\|u_n - \mathbb{E}[u_n]\| \right] \tag{70}
$$

Note that for any $j \in [n]$,

$$
\left\| \int_S \mathrm{I}_{y^{(j)}} \Phi_j - \int_S \mathbb{E}\left[ \theta_*^\top \Phi_j \Phi_j \right] \right\| \leqslant \sqrt{d}.
$$

According to Hsu et al. (2012a, Proposition 1.2), for any $\delta \in (0,1)$ and $n \in \mathbb{N}$, with probability at least $1 - \delta$, we have

$$\|u_n - \mathbb{E}[u_n]\| \leqslant \sqrt{nd} + \sqrt{8nd\log(1/\delta)} + \frac{4}{3}\sqrt{d}\log(1/\delta).$$

Then, by (70), for any $\delta \in (0,1)$, $\delta' \in (0, 1-\delta)$, and any $\lambda \geqslant 0$, with probability at least $1 - \delta + \delta'$, we have

$$\|\breve{\theta}_\lambda - \theta_*\| \leqslant \frac{1}{\mu_{\min}(\Sigma_n)}\Big[\lambda\|\breve{\theta}_\lambda\| + (\lambda + 2d\sqrt{8n\log(d/\delta_1)})\|\theta_*\|$$
$$+ 2\Big(\sqrt{nd} + \sqrt{8nd\log(1/\delta_2)} + \frac{4}{3}\sqrt{d}\log(1/\delta_2)\Big)\Big].$$

For $\lambda = 0$, we have

$$\|\breve{\theta} - \theta_*\| \leqslant \frac{1}{\mu_{\min}(\Sigma_n)}\left[2d\sqrt{8n\log(d/\delta_1)}\|\theta_*\| + 2\left(\sqrt{nd} + \sqrt{8nd\log(1/\delta_2)} + \frac{4}{3}\sqrt{d}\log(1/\delta_2)\right)\right].$$

Setting $\delta = \delta'$, we obtain (11). $\qquad\square$

## C  Proofs of minimax lower bounds

In this section, we prove Theorem 8 and Proposition 9.

### C.1  Proof of Theorem 8

*Proof.* First, we show that $\mathfrak{R}(\theta(\mathcal{P}^d_{x^{1:n}})) = \Omega(1)$ under the regime that $\mu_{\min}(U_n) = 0$. Suppose that $\phi_i(\cdot, \cdot) = \phi_1(\cdot, \cdot)$ for any $1 \leqslant i \leqslant d$. In this case, we have $\mu_{\min}(U_n) = 0$ and $\theta(P)$ can be arbitrary $\theta \in \Delta^{d-1}$ for any $P \in \mathcal{P}^d_{x^{1:n}}$. For any estimator $\breve{\theta} \in \Delta^{d-1}$, there exists $\theta' \in \Delta^{d-1}$ such that $\|\breve{\theta} - \theta'\| = \Omega(1)$ by the property of $\Delta^{d-1}$. Then, there always exists $P \in \mathcal{P}^d_{x^{1:n}}$ such that $\theta(P) = \theta'$ and hence, $\mathbb{E}_P\left[\|\breve{\theta}(y^{(1)}, \ldots, y^{(n)}) - \theta(P)\|\right] \leqslant \sup_{\theta^{(1)}, \theta^{(2)} \in \Delta^{d-1}} \|\theta^{(1)} - \theta^{(2)}\| = \Omega(1)$. Thus, we have, $\mathfrak{R}(\theta(\mathcal{P}^d_{x^{1:n}})) = \Omega(1)$ under the regime that $\mu_{\min}(U_n) = 0$.

Next, we show that $\mathfrak{R}(\theta(\mathcal{P}^d_{x^{1:n}})) = \Omega(\sqrt{\frac{d}{1 + \mu_{\min}(U_n)}})$ under the regime that $\mu_{\min}(U_n) > 0$ using Fano's method (Fano, 1961). In order to apply Fano's method, we first construct separated subset for $\Delta^{d-1}$.

Let $d_{\ell^2}$ denote the $\ell^2$ distance. For $\delta \in (0,1)$, let $P(\Delta^{d-1}, d_{\ell^2}, \delta)$ denote the $\delta$-packing number of the set $\Delta^{d-1}$. Then, we have the following lower bound on $P(\Delta^{d-1}, d_{\ell^2}, \delta)$.

**Lemma 21.** *For any $d \geqslant 2$, we have*

$$P(\Delta^{d-1}, d_{\ell^2}, \delta_0) > 2^d. \tag{71}$$

*where*

$$\delta_0 := \frac{\sqrt{e}}{2\sqrt{\pi d}}\left(\frac{\sqrt{d}}{3}\right)^{\frac{1}{d-1}}\left(\frac{1}{\sqrt{2}}\right)^{\frac{d}{d-1}} \geqslant \frac{\sqrt{2e}}{12\sqrt{\pi d}}. \tag{72}$$

The proof of Lemma 21 uses the volume method and is provided in Appendix H.

Lemma 21 implies that there exits a $\delta_0$-separated subset $\mathcal{V}_1$ of $\Delta^{d-1}$ of size $|\mathcal{V}_1| \geqslant 2^d$. Define $\mathcal{V}_a := \{l_a(\theta) : \theta \in \mathcal{V}_1\}$ where $l_a(\theta) = \left[a\theta_1, \ldots, a\theta_{d-1}, 1 - a\sum_{i=1}^{d-1}\theta_i\right]^\top$ for $0 \leqslant a < \frac{1}{\sup_{\theta \in \mathcal{V}_1}\sum_{i=1}^{d-1}\theta_i}$. Then, for any $\theta^{(1)}, \theta^{(2)} \in \mathcal{V}_1$ and any $j \in [n]$, we have

$$\|l_a(\theta^{(1)}) - l_a(\theta^{(2)})\| = \sqrt{a^2\sum_{i=1}^d\left(\theta_i^{(1)} - \theta_i^{(2)}\right)^2} = a\|\theta^{(1)} - \theta^{(2)}\|$$

Thus, we have

$$\|l_a(\theta^{(1)}) - l_a(\theta^{(2)})\| \leqslant a \sup_{x,y \in \Delta^{d-1}} \|x - y\| = \sqrt{2}a \tag{73}$$

and

$$\|l_a(\theta^{(1)}) - l_a(\theta^{(2)})\| \geqslant a\delta_0$$

which implies that $\mathcal{V}_a$ is a $(a\delta_0)$-separated subset of $\Delta^{d-1}$ of size $|\mathcal{V}_a| \geqslant 2^d$.

Let $D(Q_1\|Q_2)$ and $\chi^2(Q_1\|Q_2)$ denote the Kullback-Leibler (KL) divergence and $\chi^2$-divergence between two probability measures $Q_1$ and $Q_2$ on $\mathbb{R}$, respectively, where $Q_1$ is absolutely continuous w.r.t. $Q_2$. Their definitions are given below:

$$D(Q_1\|Q_2) := \int_{\mathbb{R}} \log\left(\frac{dQ_1}{dQ_2}\right) dQ_1 \text{ and } \chi^2(Q_1\|Q_2) := \int_{\mathbb{R}} \left(\frac{dQ_1}{dQ_2} - 1\right)^2 dQ_2,$$

where $\frac{dQ_1}{dQ_2}$ denotes the Radon-Nikodym derivative of $Q_1$ w.r.t. $Q_2$.

**Lemma 22.** *For any $d \geqslant 2$, there exists some nonempty subset $\mathcal{B}_d^B \subseteq \mathcal{B}_d$ such that for any $n \geqslant d$, we have*

$$D\left(\otimes_{j=1}^n P^\Phi_{Y|x^{(j)},\theta^{(1)}} \,\Big\|\, \otimes_{j=1}^n P^\Phi_{Y|x^{(j)},\theta^{(2)}}\right) \leqslant \frac{8a^2}{d}\left(1 + 2\mu_{\min}\left(U_n\right)\right) \tag{74}$$

*and $\mu_{\min}(U_n) > 0$ for any $\theta^{(1)}, \theta^{(2)} \in \mathcal{V}_a$ and any $\Phi \in \mathcal{B}_d^B$.*

*Proof of Lemma 22.* For any $\theta^{(1)}, \theta^{(2)} \in \mathcal{V}_a$, and $\Phi \in \mathcal{B}_d$, we have

$$D\left(P^\Phi_{Y|x^{(j)},\theta^{(1)}}\|P^\Phi_{Y|x^{(j)},\theta^{(2)}}\right) \leqslant \chi^2\left(P^\Phi_{Y|x^{(j)},\theta^{(1)}}\|P^\Phi_{Y|x^{(j)},\theta^{(2)}}\right) \tag{75}$$

where (75) follows from the bound on KL divergence w.r.t. $\chi^2$-divergence (Su, 1995) (also see Makur (2019, Lemma 2.3) or Makur & Zheng (2020, Lemma 3) and the references therein). By the tensorization of KL divergence, we have

$$D\left(\otimes_{j=1}^n P^\Phi_{Y|x^{(j)},\theta^{(1)}} \,\Big\|\, \otimes_{j=1}^n P^\Phi_{Y|x^{(j)},\theta^{(2)}}\right) = \sum_{j=1}^n D\left(P^\Phi_{Y|x^{(j)},\theta^{(1)}}\|P^\Phi_{Y|x^{(j)},\theta^{(2)}}\right). \tag{76}$$

Now, we consider a special case where $\Phi$ consists of CDFs of Bernoulli distributions. Under this Bernoulli setting, we set $S = [0,1]$ and $\mathfrak{m} = \mathsf{Leb}$. Specifically, for any $\boldsymbol{p} = (p_{ji})_{j \in [n], i \in [d]} \in [0,1]^{n \times d}$, define $\Phi^{\boldsymbol{p}}_{ji}(t) := \mathbb{P}[Z_i \leqslant t]$ with $Z_j \sim \mathsf{Bernoulli}(p_{ji})$ for any $i \in [d]$ and $j \in [n]$. Then, for any $\theta \in \Delta^{d-1}$ and $j \in [n]$, we have that $\sum_{i=1}^d \theta_i \Phi^{\boldsymbol{p}}_{ji}(t) = \mathbb{P}[Z^{(j)}_\theta \leqslant t]$ with $Z^{(j)}_\theta \sim \mathsf{Bernoulli}(p_j^\top \theta)$ where $p_j = [p_{j1}, \ldots, p_{jd}]^\top$. Let $P^B_\rho$ be the probability measure induced by the Bernoulli distribution with parameter $\rho \in [0,1]$. Define $q_{ji} := 1 - p_{ji}$ and $q_j := [q_{j1}, \ldots, q_{jd}]^\top$ for any $j \in [n]$ and $i \in [d]$. By definition, the $\chi^2$-divergence between two different Bernoulli distributions with parameters $p_j^\top \theta^{(1)}$ and $p_j^\top \theta^{(2)}$ is

$$\begin{aligned}
\chi^2\left(P^{\Phi^{\boldsymbol{p}}}_{Y|x^{(j)};\theta^{(1)}} \,\Big\|\, P^{\Phi^{\boldsymbol{p}}}_{Y|x^{(j)};\theta^{(2)}}\right) &= \chi^2\left(P^B_{p_j^\top \theta^{(1)}} \,\Big\|\, P^B_{p_j^\top \theta^{(2)}}\right) \\
&= \frac{\left(q_j^\top\left(\theta^{(1)} - \theta^{(2)}\right)\right)^2}{p_j^\top \theta^{(2)}} + \frac{\left(q_j^\top\left(\theta^{(1)} - \theta^{(2)}\right)\right)^2}{q_j^\top \theta^{(2)}} \\
&= \frac{\left(q_j^\top\left(\theta^{(1)} - \theta^{(2)}\right)\right)^2}{\left(q_j^\top \theta^{(2)}\right) p_j^\top \theta^{(2)}} \\
&\leqslant \frac{2a^2 \sum_{i=1}^d q_{ji}^2}{\left(q_j^\top \theta^{(2)}\right) p_j^\top \theta^{(2)}}
\end{aligned} \tag{77}$$

where (77) is by Cauchy-Schwarz inequality and (73). Since $S = [0, 1]$, $\mathfrak{m} = \mathsf{Leb}$, and $\Phi_j(t) = q_j$ for any $t \in [0, 1)$ and $j \in [n]$, we have

$$U_n = \sum_{j=1}^{n} \int_S \Phi_j \Phi_j^\top d\mathfrak{m} = \sum_{j=1}^{n} q_j q_j^\top.$$

Assume $d \geqslant 2$. Suppose that for any $j \in [d]$ and $i \in [d]$, $p_j$ satisfies that

$$1 - \frac{1}{d^3} \leqslant p_{ji} \leqslant 1 - \frac{1}{2d^3} \text{ and } \mu_{\min}\left(\sum_{j=1}^{d} q_j q_j^\top\right) > 0. \tag{78}$$

Since $\mu_{\min}\left(\sum_{j=1}^{d} q_j q_j^\top\right) > 0$ if $\{q_j\}_{j \in [d]}$ is linearly independent, such vectors $p_j$'s exist. For example, we can set $p_{jj} = 1 - \frac{1}{d^3}$ for any $j \in [d]$ and $p_{ji} = 1 - \frac{1}{2d^3}$ for any $i, j \in [d]$ with $i \neq j$. Then, it is clear that $q_j$'s are linearly and thus, $\mu_{\min}\left(\sum_{j=1}^{d} q_j q_j^\top\right) > 0$. Therefore, $\mu_{\min}(U_n) > 0$ for any $d \geqslant n$.

Now, for any $j \geqslant d + 1$ and $i \in [d]$, suppose that $p_j$ satisfies

$$1 - \frac{\mu_{\min}(R_{j-1})}{d^2} \leqslant p_{ji} \leqslant 1 - \frac{\mu_{\min}(R_{j-1})}{2d^2}, \tag{79}$$

where $R_j := q_j q_j^\top + \frac{1}{n} \sum_{k=1}^{j-1} q_k q_k^\top$ for any $j \geqslant d$. Then, according to the condition that $\mu_{\min}(U_d) > 0$, we have

$$0 < \mu_{\min}\left(\frac{1}{n} U_d\right) \leqslant \mu_{\min}(R_j) \leqslant \frac{1}{d} \mathrm{trace}(R_j) = \frac{1}{d}\left(q_j^\top q_j + \frac{1}{n} \sum_{k=1}^{j-1} q_k^\top q_k\right) \leqslant 2$$

for any $j \geqslant d$, which implies that $0 < \frac{\mu_{\min}(R_{j-1})}{d^2} \leqslant \frac{2}{d^2} \leqslant \frac{1}{d}$. Thus, for any $j \geqslant d$ and $i \in [d]$, the above $p_{ji}$'s are indeed defined in $[0, 1]$ and $q_{ji}$ satisfies $\frac{\mu_{\min}(R_{j-1})}{2d^2} \leqslant q_{ji} \leqslant \frac{\mu_{\min}(R_{j-1})}{d^2}$.

For notational convenience, define $R_j := \frac{1}{d} I_d$ for any $0 \leqslant j \leqslant n - 1$. Then, we have

$$\frac{\sum_{i=1}^{d} q_{ji}^2}{q_j^\top \theta^{(2)}} \leqslant \frac{2}{d} \mu_{\min}(R_{j-1})$$

and

$$p_j^\top \theta^{(2)} \geqslant 1 - \frac{\mu_{\min}(R_{j-1})}{d^2} \geqslant 1 - \frac{2}{d^2} \geqslant \frac{1}{2}.$$

It follows that

$$\frac{2a^2 \sum_{i=1}^{d} q_{ji}^2}{\left(q_j^\top \theta^{(2)}\right) p_j^\top \theta^{(2)}} \leqslant \frac{8a^2 \mu_{\min}(R_{j-1})}{d} \tag{80}$$

which, together with (75) and (77), implies that

$$D\left(P_{Y|x^{(j)}, \theta^{(1)}}^{\Phi^P} \big\| P_{Y|x^{(j)}, \theta^{(2)}}^{\Phi^P}\right) \leqslant \frac{8a^2 \mu_{\min}(R_{j-1})}{d}. \tag{81}$$

Then, by (76), we have

$$D\left(\otimes_{j=1}^{n} P_{Y|x^{(j)}, \theta^{(1)}}^{\Phi^P} \big\| \otimes_{j=1}^{n} P_{Y|x^{(j)}, \theta^{(2)}}^{\Phi^P}\right) \leqslant \frac{8a^2}{d} \sum_{j=1}^{n} \mu_{\min}(R_{j-1}))$$

$$\leqslant \frac{8a^2}{d}\left(1 + \sum_{j=d}^{n} \mu_{\min}(R_j)\right)$$

$$\leqslant \frac{8a^2}{d}\left(1 + \mu_{\min}\left(\sum_{j=d}^{n} R_j\right)\right)$$

where the second inequality follows from the fact that $\mu_{\min}(R_j) = \frac{1}{d}$ for any $0 \leqslant j \leqslant d-1$ and $\mu_{\min}(R_n) \geqslant 0$. The last inequality follows from Weyl's inequality (Weyl, 1912). Note that

$$\sum_{j=d}^{n} R_j = \sum_{j=d}^{n} q_j q_j^\top + \sum_{j=1}^{d-1} \frac{n-d+1}{n} q_j q_j^\top + \sum_{j=d}^{n-1} \frac{n-j}{n} q_j q_j^\top \preceq 2 \sum_{j=1}^{n} q_j q_j^\top = 2U_n$$

where we say $A \preceq B$ for two square matrices $A$ and $B$ of the same size if $\mu_{\min}(B-A) \geqslant 0$. Therefore, by Weyl's inequality (Weyl, 1912) again, we have $\mu_{\min}\left(\sum_{j=d}^{n} R_j\right) \leqslant 2\mu_{\min}(U_n)$ and

$$D\left(\otimes_{j=1}^{n} P_{Y|x^{(j)},\theta^{(1)}}^{\Phi^{\boldsymbol{P}}} \,\middle\|\, \otimes_{j=1}^{n} P_{Y|x^{(j)},\theta^{(2)}}^{\Phi^{\boldsymbol{P}}}\right) \leqslant \frac{8a^2}{d}\left(1 + 2\mu_{\min}(U_n)\right).$$

for any $\theta^{(1)}, \theta^{(2)} \in \mathcal{V}_a$.

In conclusion, we have proved that $\mu_{\min}(U_n) > 0$ and (74) holds for any $\theta^{(1)}, \theta^{(2)} \in \mathcal{V}_a$ and any $\Phi \in \mathcal{B}_d^B$ with

$$\mathcal{B}_d^B := \{\Phi^{\boldsymbol{P}} : p_{ji}\text{'s satisfy (78) for any } i, j \in [d] \text{ and (79) for any } j \geqslant d+1 \text{ and } i \in [d]\}.$$

As is shown in the discussions below (78) and (79), $\mathcal{B}_d^B \neq \varnothing$. $\qquad\square$

Now, define $\mathcal{P}_{x^{1:n}}^{B,d} := \left\{\otimes_{j=1}^{n} P_{Y|x^{(j)};\theta}^{\Phi} : \theta \in \Delta^{d-1}, \Phi \in \mathfrak{B}_d^B\right\} \subseteq \mathcal{P}_{x^{1:n}}^{B,d}$ with $\mathfrak{B}_d^B$ specified in Lemma 22. Then, by Lemma 22, (72), (81), and Fano's method (Fano, 1961), we have

$$\begin{aligned}
\mathfrak{R}(\theta(\mathcal{P}_{x^{1:n}}^d)) \geqslant{} & \mathfrak{R}(\theta(\mathcal{P}_{x^{1:n}}^{B,d})) && (82)\\
={} & \inf_{\hat{\theta}} \sup_{P \in \mathcal{P}_{x^{1:n}}^{B,d}} \mathbb{E}_P[\|\hat{\theta}(y^{(1)}, \ldots, y^{(n)}) - \theta(P)\|] \\
\geqslant{} & a\delta_0 \left(1 - \frac{\sup_{\theta^{(1)},\theta^{(2)} \in \mathcal{V}_a} D\left(\otimes_{j=1}^{n} F_{\theta^{(1)}}(x^{(j)}, \cdot) \,\middle\|\, \otimes_{j=1}^{n} F_{\theta^{(2)}}(x^{(j)}, \cdot)\right) + \log 2}{\log|\mathcal{V}_a|}\right) \\
\geqslant{} & a\delta_0 \left(1 - \frac{\frac{8a^2}{d}(1 + 2\mu_{\min}(U_n)) + \log 2}{d \log 2}\right) \\
\geqslant{} & \frac{a\sqrt{2e}}{12\sqrt{\pi d}}\left(1 - \frac{8a^2(1 + 2\mu_{\min}(U_n)) + d\log 2}{d^2 \log(2)}\right) && (83)
\end{aligned}$$

where (82) follows from the fact that $\mathcal{P}_{x^{1:n}}^{B,d} \subseteq \mathcal{P}_{x^{1:n}}$.

Choosing $a = \Theta\left(\frac{d}{\sqrt{1+(1+\mu_{\min}(U_n))}}\right)$, by (83), we have $\mathfrak{R}(\theta(\mathcal{P}_{x^{1:n}}^d)) = \Omega\left(\sqrt{\frac{d}{1+\mu_{\min}(U_n)}}\right)$ under the regime that $\mu_{\min}(U_n) > 0$.

Given the above results, we can conclude that

$$\mathfrak{R}(\theta(\mathcal{P}_{x^{1:n}})) = \Omega\left(\min\left\{1, \sqrt{\frac{d}{1 + \mu_{\min}(U_n)}}\right\}\right).$$

$\qquad\square$

## C.2 Proof of Corollary 9

*Proof.* Assume that $X^{(1)}, \ldots, X^{(n)}$ are independent random variables in $\mathcal{X}$. For any fixed sequence $x^{1:n} = (x^{(1)}, \ldots, x^{(n)}) \in \mathcal{X}^n$, denote by $\mathcal{P}_{X,Y;x^{1:n}}^d \subseteq \mathcal{P}_n^d$ the family of the joint distributions of

$(Y^{(1)}, X^{(2)}, \ldots, Y^{(n)}, X^{(n)})$ whose marginal distribution on $(X^{(1)}, \ldots, X^{(n)})$ is $\mathbb{1}_{x^{1:n}}$, i.e., the delta mass on $x^{1:n}$. Then, we have $\Sigma_n = U_n$ almost surely (a.s.) and

$$
\begin{aligned}
\mathfrak{R}(\theta(\mathcal{P})) &= \inf_{\hat{\theta}} \sup_{P \in \mathcal{P}_n^d} \mathbb{E}_P[\|\hat{\theta}(y^{(1)}, \ldots, y^{(n)}) - \theta(P)\|] \\
&= \inf_{\hat{\theta}} \sup_{P \in \mathcal{P}_n^d} \mathbb{E}_P \left[ \mathbb{E}_P \left[ \|\hat{\theta}(y^{(1)}, \ldots, y^{(n)}) - \theta(P)\| \big| X^{(1)}, \ldots, X^{(n)} \right] \right] \\
&\geqslant \inf_{\hat{\theta}} \sup_{P \in \mathcal{P}_{X,Y;x^{1:n}}^d} \mathbb{E}_P \left[ \|\hat{\theta}(y^{(1)}, \ldots, y^{(n)}) - \theta(P)\| \big| x^{(1)}, \ldots, x^{(n)} \right] \\
&= \Omega \left( \min \left\{ 1, \sqrt{\frac{d}{1 + \mu_{\min}(\Sigma_n)}} \right\} \right)
\end{aligned}
\tag{84}
$$

Thus, $\mathfrak{R}(\theta(\mathcal{P}_n^d)) = \Omega \left( \min \left\{ 1, \sqrt{\frac{d}{1 + \mu_{\min}(\Sigma_n)}} \right\} \right)$. $\qquad \square$

## D  Proofs of upper bounds for the infinite dimensional model

In this section, we prove Theorem 19.

*Proof of Theorem 19.* For $\theta_* \in \mathcal{H}_{\boldsymbol{\sigma}, \boldsymbol{e}}$, define the function for any $m \in \mathbb{N}$

$$
\widetilde{\theta}_{*,m} := U_n \theta_* + \sum_{i=1}^m \frac{\langle e_i, \theta_* \rangle}{\sigma_i^2} e_i.
$$

Then, we have $\widetilde{\theta}_{*,m} \in \mathcal{L}^2(\Omega, \mathfrak{n})$. For any $i \in [m]$, we have

$$
\begin{aligned}
\langle e_i, \widetilde{\theta}_{*,m} \rangle &= \langle e_i, U_n \theta_* \rangle + \frac{\langle e_i, \theta_* \rangle}{\sigma_i^2} \\
&= \langle U_n e_i, \theta_* \rangle + \frac{\langle e_i, \theta_* \rangle}{\sigma_i^2} \\
&= \left( \lambda_i + \frac{1}{\sigma_i^2} \right) \langle e_i, \theta_* \rangle.
\end{aligned}
$$

For any $i \geqslant m + 1$, we have

$$
\langle e_i, \widetilde{\theta}_{*,m} \rangle = \langle e_i, U_n \theta_* \rangle = \lambda_i \langle e_i, \theta_* \rangle.
$$

Thus, we have

$$
U_{n,\boldsymbol{\sigma}}^{-1} \widetilde{\theta}_{*,m} = \sum_{i=1}^\infty \frac{\sigma_i^2 \langle e_i, \widetilde{\theta}_{*,m} \rangle}{1 + \lambda_i \sigma_i^2} e_i = \sum_{i=1}^m \langle e_i, \theta_* \rangle e_i + \sum_{i=m+1}^\infty \frac{\lambda_i \sigma_i^2 \langle e_i, \theta_* \rangle}{1 + \lambda_i \sigma_i^2} e_i
$$

and

$$
\|\theta_* - U_{n,\boldsymbol{\sigma}}^{-1} \widetilde{\theta}_{*,m}\|^2 = \sum_{i=m+1}^\infty \frac{|\langle e_i, \theta_* \rangle|^2}{(1 + \lambda_i \sigma_i^2)^2}
$$

Since $\lim_{i \to \infty} \lambda_i = 0 = \lim_{i \to} \sigma_i$ and $\sum_{i=1}^\infty |\langle e_i, \theta_* \rangle|^2 < \infty$, we have

$$
\lim_{m \to \infty} \sum_{i=m+1}^\infty \frac{|\langle e_i, \theta_* \rangle|^2}{(1 + \lambda_i \sigma_i^2)^2} = 0.
$$

Thus, defining $\theta_{*,m} := U_{n,\sigma}^{-1}\widetilde{\theta}_{*,m}$, we have $\theta_{*,m} \to \theta_*$ as $m \to \infty$ in $\mathcal{L}^2(\Omega, \mathfrak{n})$. Moreover, by the definition of $U_n$, we have

$$\theta_{*,m}(\omega) = U_{n,\sigma}^{-1}\widetilde{\theta}_{*,m} = U_{n,\sigma}^{-1}\left(\sum_{j=1}^n \int_S \Psi_j(\theta_*, t)\Phi_j(\omega, t)\mathfrak{m}(dt) + \sum_{i=1}^m \frac{\langle e_i, \theta_*\rangle}{\sigma_i^2}e_i(\omega)\right)$$

for $\mathfrak{n}$-a.e. $\omega \in \Omega$.

We follow the same probability space constructed in Appendix B.1. Define

$$V_j(\omega) = \int_S \mathrm{I}_{y^{(j)}}(t)\Phi_j(\omega, t)\mathfrak{m}(dt) - \int_S \Psi_j(\theta_*, t)\Phi_j(\omega, t)\mathfrak{m}(dt)$$

for any $j \in [n]$ and $\mathfrak{n}$-a.e. $\omega \in \Omega$. Since $\Phi$ is $(\mathcal{B}(\mathcal{X}) \otimes \mathcal{F}_\Omega \otimes \mathcal{B}(\mathbb{R}))/\mathcal{B}([0,1])$-measurable, according to the similar arguments as in Appendix B.1.1, we have that $V_j$ is $\mathcal{F}_\Omega \otimes \mathcal{F}_j$-measurable for any $j \in [n]$. For $\mathfrak{n}$-a.e. $\omega \in \Omega$, we have $|\int_S \mathrm{I}_{y^{(j)}}(t)\Phi_j(\omega, t)\mathfrak{m}(dt)| \leq \int_S \mathfrak{m}(dt) = 1$ and $|\int_S \Psi_j(\theta_*, t)\Phi_j(\omega, t)\mathfrak{m}(dt)| \leq \int_S \mathfrak{m}(dt) = 1$. Thus, We have $-1 \leq V_j(\omega) \leq 1$ for $\mathfrak{n}$-a.e. $\omega \in \Omega$ and $V_j \in \mathcal{L}^2(\Omega, \mathfrak{n})$ because $\mathfrak{n}(\Omega) < \infty$. By Fubini's theorem, for any $j \in [n]$ and $\mathfrak{n}$-a.e. $\omega \in \Omega$, we have

$$\begin{aligned}
\mathbb{E}[V_j(\omega)|\mathcal{F}_{j-1}] &= \int_S \mathbb{E}[\mathrm{I}_{y^{(j)}}(t)|\mathcal{F}_{j-1}]\Phi_j(\omega, t)\mathfrak{m}(dt) - \int_S \Psi_j(\theta_*, t)\Phi_j(\omega, t)\mathfrak{m}(dt) \\
&= \int_S \Psi_j(\theta_*, t)\Phi_j(\omega, t)\mathfrak{m}(dt) - \int_S \Psi_j(\theta_*, t)\Phi_j(\omega, t)\mathfrak{m}(dt) \\
&= 0.
\end{aligned}$$

For any $\alpha \in \mathcal{L}^2(\Omega, \mathfrak{n})$, define $M_0(\alpha) := 1$. For any $n \in \mathbb{N}$ and $\alpha \in \mathcal{L}^2(\Omega, \mathfrak{n})$, define $W_n := \sum_{j=1}^n V_j$ and $M_n(\alpha) := \exp\{\langle\alpha, W_n\rangle - \frac{1}{2}\|\alpha\|_{U_n}^2\}$ with $\|\alpha\|_{U_n}^2 = \langle\alpha, U_n\alpha\rangle$. We have that

**Lemma 23.** *For any $\alpha \in \mathcal{L}^2(\Omega, \mathfrak{n})$, $M_n(\alpha)_{n\geq 0}$ is a non-negative super-martingale.*

The proof of Lemma 23 is similar to that in Appendix B.1.1 and is provided in Appendix J. By Lemma 23, we have

$$\mathbb{E}[M_n(\alpha)] \leq \mathbb{E}[M_0(\alpha)] = 1.$$

Since $U_{n,\sigma}^{-1}$ is a bounded linear operator on $\mathcal{L}^2(\Omega, \mathfrak{n})$, we have $U_{n,\sigma}^{-1}W_n \in \mathcal{L}^2(\Omega, \mathfrak{n})$. There exists $(w_i)_{i\in\mathbb{N}} \in \mathbb{R}^\mathbb{N}$ such that $U_{n,\sigma}^{-1}W_n = \sum_{i=1}^\infty w_i e_i$.

Let $\{\zeta_i\}_{i\geq 1}$ be a sequence of independent normal distributed random variables such that $\zeta_i \sim N(0,1)$ for any $i \in \mathbb{N}$ and $\mathcal{F}^\zeta := \sigma(\zeta_1, \zeta_2, \dots)$ is independent of $\mathcal{F}_\infty := \sigma(\cup_{j=1}^\infty \mathcal{F}_j)$.

Define $\beta_i := \sigma_i\zeta_i$ for any $i \in \mathbb{N}$. By the monotone convergence theorem, we have

$$\mathbb{E}\left[\sum_{i=1}^\infty \beta_i^2\right] = \mathbb{E}\left[\sum_{i=1}^\infty \sigma_i^2\zeta_i^2\right] = \sum_{i=1}^\infty \sigma_i^2\mathbb{E}\left[\zeta_i^2\right] = \sum_{i=1}^\infty \sigma_i^2 < \infty.$$

Thus, we have $\sum_{i=1}^\infty \beta_i^2 < \infty$ a.s., which implies that $\{\sum_{i=1}^m \beta_i e_i\}_{m\geq 1}$ and $\{\sum_{i=1}^m \sigma_i\zeta_i e_i\}_{m\geq 1}$ converges in $\mathcal{L}^2(\Omega, \mathfrak{n})$ a.s.. In particular, we have $\beta := \sum_{i=1}^\infty \sigma_i\beta_i e_i \in \mathcal{L}^2(\Omega, \mathfrak{n})$ with $\|\beta\|^2 < \infty$ a.s..

Define $\overline{M}_n := \mathbb{E}[M_n(\beta)|\mathcal{F}_\infty]$. Then, $\overline{M}_n \geq 0$. Since $M_n(\alpha)$ is $\mathcal{F}_n$-measurable for any fixed $\alpha \in \mathcal{L}^2(\Omega, \mathfrak{n})$ and $\mathcal{F}^\beta$ and $\mathcal{F}_\infty$ are independent, we have that $\overline{M}_n$ is $\mathcal{F}_n$-measurable. Then, we have

$$\begin{aligned}
\mathbb{E}[\overline{M}_n|\mathcal{F}_{n-1}] &= \mathbb{E}[M_n(\beta)|\mathcal{F}_{n-1}] \\
&= \mathbb{E}[\mathbb{E}[M_n(\beta)|\mathcal{F}^\zeta, \mathcal{F}_{n-1}]|\mathcal{F}_{n-1}] \\
&\leq \mathbb{E}[\mathbb{E}[M_{n-1}(\beta)|\mathcal{F}^\zeta, \mathcal{F}_{n-1}]|\mathcal{F}_{n-1}] \\
&= \mathbb{E}[\mathbb{E}[M_{n-1}(\beta)|\mathcal{F}^\zeta, \mathcal{F}_\infty]|\mathcal{F}_{n-1}] \\
&= \mathbb{E}[\mathbb{E}[M_{n-1}(\beta)|\mathcal{F}_\infty]|\mathcal{F}_{n-1}] \\
&= \overline{M}_{n-1}
\end{aligned}$$

and

$$\mathbb{E}[|\overline{M}_n|] = \mathbb{E}[\overline{M}_n] = \mathbb{E}[M_n(\beta)] = \mathbb{E}[\mathbb{E}[M_n(\beta)|\mathcal{F}^\beta]] \leqslant 1.$$

Thus, $\{\overline{M}_n\}_{n \geqslant 0}$ is a non-negative super-martingale.

Since $\|U_n\| \leqslant n\mathfrak{n}(\Omega)$ and $U_n$ is positive, we have $0 \leqslant \lambda_i = \langle e_i, U_n e_i \rangle \leqslant n\mathfrak{n}(\Omega)$ for any $i \in \mathbb{N}$. Define $w_i' := \langle e_i, W_n \rangle$ for any $i \in \mathbb{N}$. Define $H_m := \sum_{i=1}^m \beta_i w_i' - \frac{1}{2} \sum_{i=1}^m \lambda_i \beta_i^2$ for any $m \in \mathbb{N}$ and $H_\infty := \langle \beta, W_n \rangle - \frac{1}{2}\|\beta\|_{U_n}^2 = \sum_{i=1}^\infty \beta_i w_i' - \frac{1}{2} \sum_{i=1}^\infty \lambda_i \beta_i^2$. Then, we have $M_n(\beta) = \exp(H_\infty)$. Moreover, we prove the following convergence result.

**Lemma 24.**

$$\mathbb{E}[\exp(H_m)|\mathcal{F}_\infty] \to \mathbb{E}[M_n(\beta)|\mathcal{F}_\infty] = \overline{M}_n$$

*as $m \to \infty$ a.s..*

The proof of Lemma 24 uses the conditional dominated convergence theorem and is provided in Appendix J. Specifically, we first show that $\lim_{m \to \infty} |H_m - H_\infty| = 0$ a.s.. Then, we verify that the dominating function of $\exp(H_m)$,

$$\exp\left(n \sum_{i=1}^\infty |\sigma_i \zeta_i| + \frac{1}{2} \sum_{i=1}^\infty \lambda_i \sigma_i^2 \zeta_i^2\right),$$

is integrable. Since $\zeta_1, \zeta_2, \ldots$ are independent and $N(0,1)$-distributed random variables, we can use the monotone convergence theorem to calculate $\mathbb{E}[\exp\left(n \sum_{i=1}^\infty |\sigma_i \zeta_i| + \frac{1}{2} \sum_{i=1}^\infty \lambda_i \sigma_i^2 \zeta_i^2\right)]$. Then, it suffices to verify the convergence of the resulting series; e.g., the conditions that $|\sigma_i| < \frac{1}{\sqrt{\lambda_i}}$, $\forall i \in \mathbb{N}$ and $\sum_{i=1}^\infty |\sigma_i| < \infty$ are needed to show that

$$\prod_{i=1}^\infty \left(2\Phi_{N(0,1)}\left(n|\sigma_i|/\sqrt{1 - \lambda_i \sigma_i^2}\right)\right)$$

exists as a non-negative real number, where $\Phi_{N(0,1)}$ denotes the CDF of the $N(0,1)$ distribution.

For any $m \in \mathbb{N}$, define $\beta_m = \sum_{i=1}^m \beta_i e_i$ and $W_{n,m} = \sum_{i=1}^m w_i' e_i$. Then, we have $\beta_m, W_{n,m} \in \mathcal{L}_{\boldsymbol{\sigma}}^2(\Omega, \mathfrak{n})$ and

$$\|W_{n,m}\|_{U_{n,\boldsymbol{\sigma}}^{-1}}^2 - \|\beta_m - U_{n,\boldsymbol{\sigma}}^{-1} W_{n,m}\|_{U_{n,\boldsymbol{\sigma}}}^2 = 2\langle \beta_m, W_{n,m} \rangle - \|\beta_m\|_{U_{n,\boldsymbol{\sigma}}}^2$$

For any $m \in \mathbb{N}$, we have

$$\begin{aligned}
\mathbb{E}[\exp(H_m)|\mathcal{F}_\infty] &= \mathbb{E}\left[\exp\left\{\langle \beta_m, W_{n,m} \rangle - \frac{1}{2}\|\beta_m\|_{U_n}^2\right\} \Big| \mathcal{F}_\infty\right] \\
&= \mathbb{E}\left[\exp\left\{\langle \beta_m, W_{n,m} \rangle - \frac{1}{2}\|\beta_m\|_{U_{n,\boldsymbol{\sigma}}}^2 + \frac{1}{2}\sum_{i=1}^m \zeta_i^2\right\} \Big| \mathcal{F}_\infty\right] \\
&= \exp\left(\frac{1}{2}\|W_{n,m}\|_{U_{n,\boldsymbol{\sigma}}^{-1}}^2\right) \mathbb{E}\left[\exp\left\{\frac{1}{2}\sum_{j=1}^m \zeta_i^2 - \frac{1}{2}\|\beta_m - U_{n,\boldsymbol{\sigma}}^{-1} W_{n,m}\|_{U_{n,\boldsymbol{\sigma}}}^2\right\} \Big| \mathcal{F}_\infty\right] \\
&= \exp\left(\frac{1}{2}\|W_{n,m}\|_{U_{n,\boldsymbol{\sigma}}^{-1}}^2\right) \int_{\mathbb{R}^m} \exp\left\{-\frac{1}{2}\|\beta_m - U_{n,\boldsymbol{\sigma}}^{-1} W_{n,m}\|_{U_{n,\boldsymbol{\sigma}}}^2\right\} \frac{1}{\sqrt{2\pi}}d\zeta_1 \cdots \frac{1}{\sqrt{2\pi}}d\zeta_m \\
&= \exp\left(\frac{1}{2}\|W_{n,m}\|_{U_{n,\boldsymbol{\sigma}}^{-1}}^2\right) \int_{\mathbb{R}^m} \exp\left\{-\frac{1}{2}\sum_{i=1}^m \left(\lambda_i + \frac{1}{\sigma_i^2}\right)\beta_i^2\right\} \frac{1}{\sqrt{2\pi}}d\zeta_1 \cdots \frac{1}{\sqrt{2\pi}}d\zeta_m \\
&= \exp\left(\frac{1}{2}\|W_{n,m}\|_{U_{n,\boldsymbol{\sigma}}^{-1}}^2\right) \int_{\mathbb{R}^m} \exp\left\{-\frac{1}{2}\sum_{i=1}^m \left(1 + \lambda_i \sigma_i^2\right)\zeta_i^2\right\} \frac{1}{\sqrt{2\pi}}d\zeta_1 \cdots \frac{1}{\sqrt{2\pi}}d\zeta_m \\
&= \frac{1}{\sqrt{\prod_{i=1}^m (1 + \lambda_i \sigma_i^2)}} \exp\left(\frac{1}{2}\|W_{n,m}\|_{U_{n,\boldsymbol{\sigma}}^{-1}}^2\right)
\end{aligned}$$

Since $\lambda_i \sigma_i^2 \geqslant 0$ for any $i \in \mathbb{N}$ and $\sum_{i=1}^{\infty} \lambda_i \sigma_i^2 < \infty$, we have

$$1 \leqslant \prod_{i=1}^{\infty} (1 + \lambda_i \sigma_i^2) < \infty.$$

Since

$$\|W_{n,m}\|_{U_{n,\sigma}^{-1}}^2 = \sum_{i=1}^{m} \left(\lambda_i + \frac{1}{\sigma_i^2}\right)^{-1} (w_i')^2 = \sum_{i=1}^{m} \frac{\sigma_i^2 (w_i')^2}{1 + \lambda_i \sigma_i^2}$$

and

$$\sum_{i=1}^{\infty} \frac{\sigma_i^2 (w_i')^2}{1 + \lambda_i \sigma_i^2} \leqslant \sup_{k \in \mathbb{N}} \sigma_k^2 \sum_{i=1}^{\infty} (w_i')^2 = \sup_{k \in \mathbb{N}} \sigma_k^2 \|W_n\|^2 < \infty,$$

we have

$$\lim_{m \to \infty} \|W_{n,m}\|_{U_{n,\sigma}^{-1}}^2 = \sum_{i=1}^{\infty} \frac{\sigma_i^2 (w_i')^2}{1 + \lambda_i \sigma_i^2} = \|W_n\|_{U_{n,\sigma}^{-1}}^2 < \infty.$$

In conclusion, we have

$$\overline{M}_n = \lim_{m \to \infty} \mathbb{E}[\exp(H_m)|\mathcal{F}_\infty] = \frac{1}{\sqrt{\prod_{i=1}^{\infty}(1 + \lambda_i \sigma_i^2)}} \exp\left(\frac{1}{2}\|W_n\|_{U_{n,\sigma}^{-1}}^2\right)$$

Since $\{\overline{M}_n\}_{n \geqslant 0}$ is a super-martingale. By Doob's maximal inequality for super-martingales,

$$\mathbb{P}\left[\sup_{n \in \mathbb{N}} \overline{M}_n \geqslant \delta\right] \leqslant \frac{\mathbb{E}[\overline{M}_0]}{\delta} = \frac{1}{\delta}$$

which, implies that,

$$\mathbb{P}\left[\exists n \in \mathbb{N} \text{ s.t. } \|W_n\|_{U_{n,\sigma}^{-1}} \geqslant \sqrt{\log\left(\prod_{i=1}^{\infty}(1 + \lambda_i \sigma_i^2)\right) + 2\log\frac{1}{\delta}}\right] \leqslant \delta. \tag{85}$$

Define the finite rank operator $\varsigma_m : \mathcal{H}_{\boldsymbol{\sigma},\boldsymbol{e}} \to \mathcal{L}^2(\Omega, \mathfrak{n})$, $\theta \mapsto \sum_{i=1}^{m} \frac{\langle e_i, \theta \rangle}{\sigma_i^2} e_i$. Since

$$\widehat{\theta}_{\boldsymbol{\sigma}} - \theta_{*,m} = U_{n,\boldsymbol{\sigma}}^{-1}(W_n - \varsigma_m \theta_*) = U_{n,\boldsymbol{\sigma}}^{-1} W_n - U_{n,\boldsymbol{\sigma}}^{-1} \varsigma_m \theta_*,$$

by the triangle inequality, we have

$$\begin{aligned}
\|\widehat{\theta}_{\boldsymbol{\sigma}} - \theta_{*,m}\|_{U_{n,\boldsymbol{\sigma}}} &\leqslant \|U_{n,\boldsymbol{\sigma}}^{-1} W_n\|_{U_{n,\boldsymbol{\sigma}}} + \|U_{n,\boldsymbol{\sigma}}^{-1} \varsigma_m \theta_*\|_{U_{n,\boldsymbol{\sigma}}} \\
&= \|W_n\|_{U_{n,\boldsymbol{\sigma}}^{-1}} + \|\varsigma_m \theta_*\|_{U_{n,\boldsymbol{\sigma}}^{-1}} \\
&= \|W_n\|_{U_{n,\boldsymbol{\sigma}}^{-1}} + \sqrt{\sum_{i=1}^{m} \frac{|\theta_{*,i}|^2}{(1 + \lambda_i \sigma_i^2)\sigma_i^2}} \\
&\leqslant \|W_n\|_{U_{n,\boldsymbol{\sigma}}^{-1}} + \sqrt{\sum_{i=1}^{m} \frac{|\theta_{*,i}|^2}{\sigma_i^2}}
\end{aligned}$$

Besides, we have

$$\|\theta_{*,m} - \theta_*\|_{U_{n,\boldsymbol{\sigma}}}^2 = \sum_{i=m+1}^{\infty} \frac{\left(\lambda_i + \frac{1}{\sigma_i^2}\right)|\langle e_i, \theta_* \rangle|^2}{(1 + \lambda_i \sigma_i^2)^2} = \sum_{i=m+1}^{\infty} \frac{|\langle e_i, \theta_* \rangle|^2}{\sigma_i^2(1 + \lambda_i \sigma_i^2)}$$

Since $\lim_{i\to\infty}\lambda_i = 0 = \lim_{i\to}\sigma_i$ and $\sum_{i=1}^{\infty}\frac{|\langle e_i,\theta_*\rangle|^2}{\sigma_i^2} < \infty$, we have

$$\lim_{m\to\infty}\|\theta_{*,m} - \theta_*\|_{U_{n,\sigma}}^2 = \lim_{m\to\infty}\sum_{i=m+1}^{\infty}\frac{|\langle e_i,\theta_*\rangle|^2}{\sigma_i^2(1+\lambda_i\sigma_i^2)^2} = 0.$$

Thus,

$$\|\widehat{\theta}_{\boldsymbol{\sigma}} - \theta_*\|_{U_{n,\sigma}} \leqslant \limsup_{m\to\infty}\|\widehat{\theta}_{\boldsymbol{\sigma}} - \theta_{*,m}\|_{U_{n,\sigma}} \leqslant \|W_n\|_{U_{n,\sigma}^{-1}} + \sqrt{\sum_{i=1}^{\infty}\frac{|\theta_{*,i}|^2}{\sigma_i^2}} = \|W_n\|_{U_{n,\sigma}^{-1}} + \|\theta_*\|_{\boldsymbol{\sigma},\boldsymbol{e}}.$$

With probability at least $1-\delta$, for all $n\in\mathbb{N}$, we have

$$\|\widehat{\theta}_{\boldsymbol{\sigma}} - \theta_*\|_{U_{n,\sigma}} \leqslant \sqrt{\left(\sum_{i=1}^{\infty}\log\left(1+\lambda_i\sigma_i^2\right)\right) + 2\log\frac{1}{\delta}} + \|\theta_*\|_{\boldsymbol{\sigma},\boldsymbol{e}}.$$

Since $0\leqslant\lambda_i = \langle e_i, U_n e_i\rangle \leqslant n\mathfrak{n}(\Omega)$ for any $i\in\mathbb{N}$, the above inequality implies that

$$\|\widehat{\theta}_{\boldsymbol{\sigma}} - \theta_*\|_{U_{n,\sigma}} \leqslant \sqrt{\left(\sum_{i=1}^{\infty}\log\left(1+n\mathfrak{n}(\Omega)\sigma_i^2\right)\right) + 2\log\frac{1}{\delta}} + \|\theta_*\|_{\boldsymbol{\sigma},\boldsymbol{e}}.$$

$\square$

## E    Proof of Lemma 6

*Proof.* For any $n\in\mathbb{N}$, define $\Delta_n^U := \|U_n - \Sigma_n\|$ and $Z_j := \int_S \Phi_j\Phi_j^\top - \Sigma_n$ for $j\in[n]$. We have $\mathbb{E}[Z_j] = 0$ and

$$\left\|\int_S \Phi_j\Phi_j^\top\right\|_2 = \mu_{\max}\left(\int_S \Phi_j\Phi_j^\top\right) \leqslant \int_S \|\Phi_j\|_2^2 \leqslant d,$$

$$\|\Sigma^{(j)}\|_2 = \mu_{\max}\left(\Sigma^{(j)}\right) = \mu_{\max}\left(\mathbb{E}\left[\int_S \Phi_j\Phi_j^\top\right]\right) \leqslant \mathbb{E}\left[\int_S \|\Phi_j\|_2^2\right] \leqslant d.$$

Thus, for each $j\in[n]$, we have

$$\|Z_j\| \leqslant \max\left\{\left\|\int_S \Phi_j\Phi_j^\top\right\|_2, \ \|\Sigma^{(j)}\|_2\right\} \leqslant d.$$

By (Tropp, 2012, Theorem 1.3), for any $a\geqslant 0$, we have

$$\mathbb{P}[\Delta_n^U \geqslant a] \leqslant d\exp\left(-\frac{a^2}{8nd^2}\right).$$

In other words, for any $\delta\in(0,1)$, with probability at least $1-\delta$, we have

$$\Delta_n^U \leqslant d\sqrt{8n\log(d/\delta)}.$$

Thus, we can set $\Delta_n^U(\delta) \geqslant d\sqrt{8n\log(d/\delta)}.$   $\square$

## F    Proofs of theoretical results in Section 6.2

In this section, we provide the proofs of the stated theoretical results in Section 6.2.

*Proof of Lemma 12.* By Fubini's theorem, for any $\theta \in \mathcal{L}^2(\Omega, \mathfrak{n})$, we have

$$(U_n\theta)(\omega) = \int_\Omega \theta(\omega') \sum_{j=1}^n \int_S \Phi_j(\omega, t)\Phi_j(\omega', t)\mathfrak{m}(dt)\mathfrak{n}(d\omega').$$

Define the function

$$u_n : \Omega^2 \to \mathbb{R}, \ (\omega, \omega') \mapsto \sum_{j=1}^n \int_S \Phi_j(\omega, t)\Phi_j(\omega', t)\mathfrak{m}(dt).$$

Then, we have $(U_n\theta)(\omega) = \int_\Omega u_n(\omega, \omega')\theta(\omega')\mathfrak{n}(d\omega')$. Since $\Phi_j(\omega, t) \in [0, 1]$ for any $\omega \in \Omega$, $t \in S$ and $\mathfrak{m}$ is a probability measure on $S$, we have $u_n(\omega, \omega') \in [0, n]$ for any $\omega, \omega' \in \Omega$ and

$$\int_\Omega \int_\Omega |u_n(\omega, \omega')|^2 \mathfrak{n}(d\omega)\mathfrak{n}(d\omega') \leqslant n^2\mathfrak{n}(\Omega)^2.$$

Thus, $u_n \in \mathcal{L}^2(\Omega^2, \mathfrak{n}^2)$ and $U_n$ is Hilbert-Schmidt integral operator for any $n \in \mathbb{N}$. Thus, it is also a compact operator.

Because $u_n(\omega, \omega') = \sum_{j=1}^n \int_S \Phi_j(\omega, t)\Phi_j(\omega', t)\mathfrak{m}(dt) = u_n(\omega', \omega) \in \mathbb{R}$, $U_n$ is self-adjoint. For any $\theta \in \mathcal{L}^2(\Omega, \mathfrak{n})$, we have

$$\begin{aligned}
\|U_n\theta\|^2 &= \int_\Omega \left| \int_\Omega \theta(\omega')u_n(\omega, \omega')\mathfrak{n}(d\omega') \right|^2 \mathfrak{n}(d\omega) \\
&\leqslant \int_\Omega \int_\Omega |\theta(\omega')|^2\mathfrak{n}(d\omega') \int_\Omega |u_n(\omega, \omega')|^2\mathfrak{n}(d\omega')\mathfrak{n}(d\omega) \\
&\leqslant n^2\mathfrak{n}(\Omega)^2\|\theta\|^2
\end{aligned}$$

and

$$\begin{aligned}
\langle U_n\theta, \theta \rangle &= \int_\Omega \int_\Omega u_n(\omega, \omega')\theta(\omega')\theta(\omega)\mathfrak{n}(\omega')\mathfrak{n}(\omega) \\
&= \sum_{j=1}^n \int_S \langle \theta(\cdot), \Phi_j(\cdot, t) \rangle^2 \mathfrak{m}(dt) \\
&\geqslant 0.
\end{aligned}$$

Thus, $U_n$ is a positive operator with $\|U_n\| \leqslant n\mathfrak{n}(\Omega)$. Note that $\langle U_n\theta, \theta \rangle = 0$ iff $\langle \theta(\cdot), \Phi_j(\cdot, t) \rangle = 0$ for $\mathfrak{m}$-a.e. $t \in S$ for all $j \in [n]$. Since $U_n$ is compact, if $\dim(\mathcal{L}^2(\Omega, \mathfrak{n})) = \infty$, $U_n$ is not invertible. $\square$

*Proof of Corollary 14.* By Lemma 12, since $\|U_n\| \leqslant n\mathfrak{n}(\Omega)$ and $U_n$ is positive, we have $0 \leqslant \lambda_i = \langle e_i, U_ne_i \rangle \leqslant n\mathfrak{n}(\Omega)$ for any $i \in \mathbb{N}$. Since $U_n$ is a compact operator and $\{e_i\}_{i \in \mathbb{N}}$ is an orthonormal basis consisting of eigenfunctions of $U_n$, by the Riesz-Schauder theorem (see e.g., Reed & Simon, 1972), we have that $\lambda_i \to 0$. $\square$

*Proof of Lemma 15.* For any $f, g \in \mathcal{L}_{\boldsymbol{\sigma}}^2(\Omega, \mathfrak{n})$ and $\alpha \in \mathbb{R}$, we have

$$\begin{aligned}
\sum_{i=1}^\infty \frac{|\alpha\langle e_i, f \rangle + \langle e_i, g \rangle|^2}{\sigma_i^4} &\leqslant \sum_{i=1}^\infty \frac{|\alpha|^2|\langle e_i, f \rangle|^2 + |\langle e_i, g \rangle|^2 + 2|\alpha||\langle e_i, f \rangle||\langle e_i, g \rangle|}{\sigma_i^4} \\
&\leqslant |\alpha|^2 \sum_{i=1}^\infty \frac{|\langle e_i, f \rangle|^2}{\sigma_i^4} + \sum_{i=1}^\infty \frac{|\langle e_i, g \rangle|^2}{\sigma_i^4} + 2|\alpha|\sqrt{\sum_{i=1}^\infty \frac{|\langle e_i, f \rangle|^2}{\sigma_i^4} \sum_{i=1}^\infty \frac{|\langle e_i, g \rangle|^2}{\sigma_i^4}} \\
&< \infty
\end{aligned}$$

Then, we have $\alpha f + g \in \mathcal{L}_{\boldsymbol{\sigma}}^2(\Omega, \mathfrak{n})$ and $\mathcal{L}_{\boldsymbol{\sigma}}^2(\Omega, \mathfrak{n})$ is a linear subspace of $\mathcal{L}^2(\Omega, \mathfrak{n})$. $\square$

*Proof of Lemma 16.* For any $\alpha \in \mathbb{R}$ and $f, g \in \mathcal{L}^2_{\boldsymbol{\sigma}}(\Omega, \mathfrak{n})$, we have $\alpha f + g \in \mathcal{L}^2_{\boldsymbol{\sigma}}(\Omega, \mathfrak{n})$,

$$
\begin{aligned}
U_{n,\boldsymbol{\sigma}}(\alpha f + g) &= \lim_{m \to \infty} \sum_{i=1}^{m} \left( \lambda_i + \frac{1}{\sigma_i^2} \right) \langle e_i, \alpha f + g \rangle e_i \\
&= \lim_{m \to \infty} \left[ \alpha \sum_{i=1}^{m} \left( \lambda_i + \frac{1}{\sigma_i^2} \right) \langle e_i, f \rangle e_i + \sum_{i=1}^{m} \left( \lambda_i + \frac{1}{\sigma_i^2} \right) \langle e_i, g \rangle e_i \right] \\
&= \alpha U_{n,\boldsymbol{\sigma}} f + U_{n,\boldsymbol{\sigma}} g,
\end{aligned}
$$

and

$$
\begin{aligned}
\| U_{n,\boldsymbol{\sigma}} f \|^2 &= \sum_{i=1}^{\infty} \left( \lambda_i + \frac{1}{\sigma_i^2} \right)^2 |\langle e_i, f \rangle|^2 \\
&\geqslant \inf_{k \in \mathbb{N}} \left( \lambda_k + \frac{1}{\sigma_k^2} \right)^2 \sum_{i=1}^{\infty} |\langle e_i, f \rangle|^2 \\
&\geqslant \frac{1}{\sup_{k \in \mathbb{N}} \sigma_k^4} \| f \|^2
\end{aligned}
$$

Thus, $U_{n,\boldsymbol{\sigma}}$ is a linear operator. Since $0 \leqslant \sup_{i \in \mathbb{N}} \sigma_i^2 < \infty$, we can conclude that $\| U_{n,\boldsymbol{\sigma}} f \| = 0$ iff $f = 0$. Therefore, $U_{n,\boldsymbol{\sigma}}$ is injective.

For any $\theta = \sum_{i=1}^{\infty} \langle e_i, \theta \rangle e_i \in \mathcal{L}^2(\Omega, \mathfrak{n})$, we have

$$
\sum_{i=1}^{\infty} \frac{\sigma_i^4 |\langle e_i, \theta \rangle|^2}{(1 + \lambda_i \sigma_i^2)^2} \leqslant \sup_{k \in \mathbb{N}} \sigma_k^4 \sum_{i=1}^{\infty} |\langle e_i, \theta \rangle|^2 < \infty.
$$

and

$$
\sum_{i=1}^{\infty} \frac{|\langle e_i, \theta \rangle|^2}{(1 + \lambda_i \sigma_i^2)^2} \leqslant \sum_{i=1}^{\infty} |\langle e_i, \theta \rangle|^2 < \infty.
$$

Thus, we know that $\check{\theta} := \sum_{i=1}^{\infty} \frac{\sigma_i^2 \langle e_i, \theta \rangle}{1 + \lambda_i \sigma_i^2} e_i \in \mathcal{L}^2_{\boldsymbol{\sigma}}(\Omega, \mathfrak{n})$. Since

$$
U_{n,\boldsymbol{\sigma}} \check{\theta} = \sum_{i=1}^{\infty} \left( \lambda_i + \frac{1}{\sigma_i^2} \right) \frac{\sigma_i^2 \langle e_i, \theta \rangle}{1 + \lambda_i \sigma_i^2} e_i = \sum_{i=1}^{\infty} \langle e_i, \theta \rangle e_i = \theta,
$$

we can conclude that

$$
U_{n,\boldsymbol{\sigma}}^{-1} \theta = \check{\theta} = \sum_{i=1}^{\infty} \frac{\sigma_i^2 \langle e_i, \theta \rangle}{1 + \lambda_i \sigma_i^2} e_i
$$

and $U_{n,\boldsymbol{\sigma}}$ is a bijective linear operator from $\mathcal{L}^2_{\boldsymbol{\sigma}}(\Omega, \mathfrak{n})$ onto $\mathcal{L}^2(\Omega, \mathfrak{n})$. Then, $U_{n,\boldsymbol{\sigma}}^{-1}$ exists as a bijective linear operator from $\mathcal{L}^2(\Omega, \mathfrak{n})$ onto $\mathcal{L}^2_{\boldsymbol{\sigma}}(\Omega, \mathfrak{n})$. Since for any $f \in \mathcal{L}^2_{\boldsymbol{\sigma}}(\Omega, \mathfrak{n})$, we have proved that $\| f \| \leqslant \sup_{k \in \mathbb{N}} \sigma_k^2 \| U_{n,\boldsymbol{\sigma}} f \|$, we have $\| U_{n,\boldsymbol{\sigma}}^{-1} \| \leqslant \sup_{i \in \mathbb{N}} \sigma_i^2$ and $U_{n,\boldsymbol{\sigma}}^{-1}$ is a bounded linear operator on $\mathcal{L}^2(\Omega, \mathfrak{n})$. $\qquad \square$

*Proof of Lemma 17.* Since $\lim_{i \to \infty} \sigma_i = 0$, there exists some $N \in \mathbb{N}$ such that $\sigma_i \leqslant 1$ for any $i \geqslant N$. Then, for any $\theta \in \mathcal{H}_{\boldsymbol{\sigma}, \boldsymbol{e}}$, we have

$$
\sum_{i=1}^{\infty} \frac{|\langle e_i, \theta \rangle|^2}{\sigma_i^4} \leqslant \sum_{i=1}^{N} \frac{|\langle e_i, \theta \rangle|^2}{\sigma_i^4} + \sum_{i=N+1}^{\infty} \frac{|\langle e_i, \theta \rangle|^2}{\sigma_i^2} \leqslant \sum_{i=1}^{N} \frac{|\langle e_i, \theta \rangle|^2}{\sigma_i^4} + \sum_{i=1}^{\infty} \frac{|\langle e_i, \theta \rangle|^2}{\sigma_i^2} < \infty.
$$

Thus, we have $\theta \in \mathcal{L}^2_{\boldsymbol{\sigma}}(\Omega, \mathfrak{n})$. $\qquad \square$

# G   Proof of Proposition 18

*Proof.* First, we show that $\sum_{j=1}^{n} \int_S \mathrm{I}_{y^{(j)}}(t) \Phi_j(\cdot, t) \mathfrak{m}(dt) \in \mathcal{L}^2(\Omega, \mathfrak{n})$. Indeed, we have

$$
\|\sum_{j=1}^{n} \int_S \mathrm{I}_{y^{(j)}}(t) \Phi_j(\cdot, t) \mathfrak{m}(dt)\|^2 \leqslant \sum_{j=1}^{n} \| \int_S \mathrm{I}_{y^{(j)}}(t) \Phi_j(\cdot, t) \mathfrak{m}(dt)\|^2
$$
$$
\leqslant \sum_{j=1}^{n} \| \int_S |\mathrm{I}_{y^{(j)}}(t) \Phi_j(\cdot, t)| \mathfrak{m}(dt)\|^2
$$
$$
\leqslant n \mathfrak{n}(\Omega)
$$
$$
\leqslant \infty.
$$

Define $\theta_0 := U_{n,\boldsymbol{\sigma}}^{-1} \left( \sum_{j=1}^{n} \int_S \mathrm{I}_{y^{(j)}}(t) \Phi_j(\cdot, t) \mathfrak{m}(dt) \right)$. We show that $\theta_0 \in \mathcal{L}_{\boldsymbol{\sigma}}^2(\Omega, \mathfrak{n})$. Then, by Lemma 17, we have $\theta_0 \in \mathcal{H}_{\boldsymbol{\sigma}, \boldsymbol{e}}$. Since $\lambda_i \geqslant 0$ for any $i \in \mathbb{N}$, we have

$$
\sum_{i=1}^{\infty} \frac{1}{\sigma_i^4} |\langle e_i, \theta \rangle|^2 \leqslant \sum_{i=1}^{\infty} \left( \lambda_i + \frac{1}{\sigma_i^2} \right)^2 |\langle e_i, \theta_0 \rangle|^2
$$
$$
\leqslant \|U_{n,\boldsymbol{\sigma}} \theta_0\|^2
$$
$$
= \|\sum_{j=1}^{n} \int_S \mathrm{I}_{y^{(j)}}(t) \Phi_j(\cdot, t) \mathfrak{m}(dt)\|^2 < \infty.
$$

Thus, $\theta_0 \in \mathcal{L}_{\boldsymbol{\sigma}}^2(\Omega, \mathfrak{n})$. Then, for any $j \in [n]$, we have

$$
|\Psi_j(\theta_0, t)| = |\langle \theta_0(\cdot), \Phi_j(\cdot, t) \rangle|
$$
$$
\leqslant \int_{\Omega} |\theta_0(\omega) \Phi_j(\omega, t)| \mathfrak{n}(d\omega)
$$
$$
\leqslant \|\theta_0\| \|\Phi_j(\cdot, t)\|
$$
$$
\leqslant \sqrt{\mathfrak{n}(\Omega)} \|\theta_0\|
$$
$$
< \infty,
$$

which implies that

$$
L(\theta_0; \boldsymbol{\sigma}) = \sum_{j=1}^{n} \|\mathrm{I}_{y^{(j)}}(\cdot) - \Psi_j(\theta_0, t)\|_{\mathcal{L}^2(S, \mathfrak{m})} + \|\theta_0\|_{\boldsymbol{\sigma}, \boldsymbol{e}} < \infty
$$

since $\mathfrak{m}(S) = 1$ and $|\mathrm{I}_{y^{(j)}}(t)| \leqslant 1$ for any $j \in [n]$ and $t \in S$.

For any $\theta \in \mathcal{H}_{\boldsymbol{\sigma}, \boldsymbol{e}}$, we have

$$
L(\theta_0 + \theta; \boldsymbol{\sigma}) = L(\theta_0) + \sum_{j=1}^{n} \int_S |\Psi_j(\theta, t)|^2 \mathfrak{m}(dt) + \sum_{i=1}^{\infty} \frac{|\langle e_i, \theta \rangle|^2}{\sigma_i^2}
$$
$$
+ \sum_{j=1}^{n} \int_S \Psi_j(\theta, t)(\Psi_j(\theta_0, t) - \mathrm{I}_{y^{(j)}}(t)) \mathfrak{m}(dt) + \sum_{i=1}^{\infty} \frac{\langle e_i, \theta \rangle \langle e_i, \theta_0 \rangle}{\sigma_i^2}.
$$

Notice that by Fubini's theorem, we have

$$
\begin{aligned}
\sum_{j=1}^{n} \int_{S} \Psi_j(\theta, t)\Psi_j(\theta_0, t)\mathfrak{m}(dt) &= \sum_{j=1}^{n} \int_{S} \int_{\Omega} \int_{\Omega} \theta(\omega)\Phi_j(\omega, t)\theta_0(\omega')\Phi_j(\omega', t)\mathfrak{n}(d\omega')\mathfrak{n}(d\omega)\mathfrak{m}(dt) \\
&= \int_{\Omega} \theta(\omega) \int_{\Omega} \theta_0(\omega') \left( \sum_{j=1}^{n} \int_{S} \Phi_j(\omega, t)\Phi_j(\omega', t)\mathfrak{m}(dt) \right) \mathfrak{n}(d\omega')\mathfrak{n}(d\omega) \\
&= \int_{\Omega} \theta(\omega) \int_{\Omega} \theta_0(\omega') u_n(\omega, \omega')\mathfrak{n}(d\omega')\mathfrak{n}(d\omega) \\
&= \langle \theta, U_n\theta_0 \rangle,
\end{aligned}
$$

$$
\begin{aligned}
\sum_{j=1}^{n} \int_{S} \Psi_j(\theta, t)\mathrm{I}_{y^{(j)}}(t)\mathfrak{m}(dt) &= \sum_{j=1}^{n} \int_{S} \int_{\Omega} \theta(\omega)\Phi_j(\omega, t)\mathrm{I}_{y^{(j)}}(t)\mathfrak{n}(d\omega)\mathfrak{m}(dt) \\
&= \int_{\Omega} \theta(\omega) \left( \sum_{j=1}^{n} \int_{S} \mathrm{I}_{y^{(j)}}(t)\Phi_j(\omega, t)\mathfrak{m}(dt) \right) \mathfrak{n}(d\omega) \\
&= \langle \theta(\cdot), \sum_{j=1}^{n} \int_{S} \mathrm{I}_{y^{(j)}}(t)\Phi_j(\cdot, t)\mathfrak{m}(dt) \rangle,
\end{aligned}
$$

and

$$
\sum_{i=1}^{\infty} \frac{\langle e_i, \theta \rangle \langle e_i, \theta_0 \rangle}{\sigma_i^2} = \sum_{i=1}^{\infty} \langle \theta, \frac{\langle e_i, \theta_0 \rangle}{\sigma_i^2} e_i \rangle
$$

Since we have proved that

$$
\sum_{i=1}^{\infty} \frac{|\langle e_i, \theta_0 \rangle|^2}{\sigma_i^4} < \infty,
$$

we can conclude that $\sum_{i=1}^{\infty} \langle \theta, \frac{\langle e_i, \theta_0 \rangle}{\sigma_i^2} e_i \rangle = \langle \theta, \sum_{i=1}^{\infty} \frac{\langle e_i, \theta_0 \rangle}{\sigma_i^2} e_i \rangle$. Thus,

$$
\begin{aligned}
&\sum_{j=1}^{n} \int_{S} \Psi_j(\theta, t)(\Psi_j(\theta_0, t) - \mathrm{I}_{y^{(j)}}(t))\mathfrak{m}(dt) + \sum_{i=1}^{\infty} \frac{\langle e_i, \theta \rangle \langle e_i, \theta_0 \rangle}{\sigma_i^2} \\
&= \langle \theta(\cdot), (U_n\theta_0)(\cdot) + \sum_{i=1}^{\infty} \frac{\langle e_i, \theta_0 \rangle}{\sigma_i^2} e_i(\cdot) - \sum_{j=1}^{n} \int_{S} \mathrm{I}_{y^{(j)}}(t)\Phi_j(\cdot, t)\mathfrak{m}(dt) \rangle \\
&= \langle \theta(\cdot), (U_{n,\boldsymbol{\sigma}}\theta_0)(\cdot) - \sum_{j=1}^{n} \int_{S} \mathrm{I}_{y^{(j)}}(t)\Phi_j(\cdot, t)\mathfrak{m}(dt) \rangle \\
&= \langle \theta(\cdot), \sum_{j=1}^{n} \int_{S} \mathrm{I}_{y^{(j)}}(t)\Phi_j(\cdot, t)\mathfrak{m}(dt) - \sum_{j=1}^{n} \int_{S} \mathrm{I}_{y^{(j)}}(t)\Phi_j(\cdot, t)\mathfrak{m}(dt) \rangle \\
&= 0.
\end{aligned}
$$

Then, we have

$$
L(\theta_0 + \theta; \boldsymbol{\sigma}) = L(\theta_0) + \sum_{j=1}^{n} \int_{S} |\Psi_j(\theta, t)|^2 \mathfrak{m}(dt) + \sum_{i=1}^{\infty} \frac{|\langle e_i, \theta \rangle|^2}{\sigma_i^2} \geqslant L(\theta_0; \boldsymbol{\sigma}).
$$

Since $\frac{1}{\sigma_i^2} > 0$ for all $i \in \mathbb{N}$, we have that $\sum_{i=1}^{\infty} \frac{|\langle e_i, \theta \rangle|^2}{\sigma_i^2} > 0$ for any $\theta \in \mathcal{H}_{\boldsymbol{\sigma},e}$ with $\theta \neq 0$. Since $L(\theta_0; \boldsymbol{\sigma}) < \infty$, we can conclude that $L(\theta; \boldsymbol{\sigma}) > L(\theta_0; \boldsymbol{\sigma})$ for any $\theta \in \mathcal{H}_{\boldsymbol{\sigma},e} \backslash \{\theta_0\}$ and $\widehat{\theta}_{\boldsymbol{\sigma}} = \theta_0$. $\qquad \square$

## H   Proof of Lemma 21

*Proof.* By Vershynin (2018, Proposition 4.2.12), we have

$$P(\Delta^{d-1}, d_{\ell^2}, \delta) \geqslant \frac{\text{Vol}(\Delta^{d-1})}{\text{Vol}(B_\delta^{d-1}(0))} \tag{86}$$

where $B_\delta^{d-1}(0) := \{x \in \mathbb{R}^{d-1} : \|x\|_2 \leqslant \delta\}$ and for any $E \subseteq \mathbb{R}^{d-1}$, $\text{Vol}(E)$ is the volume of $E$ under the Lebesgue measure in $\mathbb{R}^{d-1}$. According to Stein (1966); DLMF, We have

$$\text{Vol}(\Delta^{d-1}) = \frac{\sqrt{d}}{(d-1)!}, \tag{87}$$

$$\text{Vol}(B_\delta^{d-1}(0)) = \frac{(\sqrt{\pi}\delta)^{d-1}}{\Gamma(\frac{d+1}{2})}. \tag{88}$$

Thus,

$$P(\Delta^{d-1}, d_{\ell^2}, \delta) \geqslant \frac{\Gamma(\frac{d+1}{2})\sqrt{d}}{(d-1)!(\sqrt{\pi}\delta)^{d-1}}$$

$$= \frac{\Gamma(\frac{d+1}{2})\sqrt{d}}{\Gamma(d)(\sqrt{\pi}\delta)^{d-1}}. \tag{89}$$

When $d \geqslant 3$, we have $\frac{d+1}{2} \geqslant 2$ and $d \geqslant 2$. According to Batir (2008, Theorem 1.5), we have $2((x-1/2)/e)^{x-1/2} < \Gamma(x) < 3((x-1/2)/e)^{x-1/2}$ for any $x \geqslant 2$. Thus, for $d \geqslant 3$, we have

$$\frac{\Gamma(\frac{d+1}{2})}{\Gamma(d)} > \frac{2}{3}\frac{\left(\frac{d}{2e}\right)^{d/2}}{\left(\frac{d-1/2}{e}\right)^{d-1/2}}.$$

We verify that the above inequality also holds when $d = 2$. Therefore, for any $d \geqslant 2$, we have

$$\frac{\Gamma(\frac{d+1}{2})}{\Gamma(d)} > \frac{2}{3}\frac{\left(\frac{d}{2e}\right)^{d/2}}{\left(\frac{d-1/2}{e}\right)^{d-1/2}}$$

which implies that

$$P(\Delta^{d-1}, d_{\ell^2}, \delta) > \frac{2\sqrt{d}}{3(\sqrt{\pi}\delta)^{d-1}}\frac{\left(\frac{d}{2e}\right)^{d/2}}{\left(\frac{d-1/2}{e}\right)^{d-1/2}}$$

$$\geqslant \frac{2\sqrt{d}}{3(\sqrt{\pi}\delta)^{d-1}}\frac{\left(\frac{d}{2e}\right)^{d/2}}{\left(\frac{d}{e}\right)^{d-1/2}}$$

$$= \frac{2\sqrt{d}}{3(\sqrt{\pi}\delta)^{d-1}}\frac{1}{2^{d/2}}e^{\frac{d-1}{2}}d^{-\frac{d-1}{2}}. \tag{90}$$

Let $\delta = \delta_0 = \frac{\sqrt{e}}{2\sqrt{\pi d}}\left(\frac{\sqrt{d}}{3}\right)^{\frac{1}{d-1}}\left(\frac{1}{\sqrt{2}}\right)^{\frac{d}{d-1}}$. Then, by (90), we have

$$P(\Delta^{d-1}, d_{\ell^2}, \delta_0) > 2^d$$

which is exactly (71). For $d \geqslant 2$, we have $\delta_0 \geqslant \frac{\sqrt{e}}{4\sqrt{\pi d}}\left(\frac{\sqrt{d}}{3}\right)^{\frac{1}{d-1}}$. Consider the function $f(x) = \frac{1}{x-1}\log\left(\frac{\sqrt{x}}{3}\right)$ with $x \geqslant 2$. We have that

$$f'(x) = \frac{1 - \frac{1}{x} - \log x + 2\log 3}{2(x-1)^2}.$$

Since the function $g : x \mapsto -\frac{1}{x} - \log x$ is a decreasing function when $x \geqslant 2$ and $f'(2) > 0$, $f'(e^5) < 0$, we have that $f$ first increases and then decreases when $x$ increases from 2 to infinity. Since $\lim_{x \to \infty} f(x) = 0$, $f(2) = \log(\sqrt{2}/3)$, we have that $f(x) \geqslant f(2) = \log(\sqrt{2}/3)$. Therefore, for any $d \geqslant 2$, we have $\left( \frac{\sqrt{d}}{3} \right)^{\frac{1}{d-1}} \geqslant \sqrt{2}/3$ and

$$\delta_0 \geqslant \frac{\sqrt{2e}}{12\sqrt{\pi d}}$$

which gives (72). $\qquad\square$

# I Proofs of upper bounds for the mismatched model

In this Section, we prove Theorem 10 in Appendix I.1 and Corollary 11 in Appendix I.2.

## I.1 Proof of Theorem 10

*Proof.* In the setting of Theorem 10, the sample $\{(x^{(j)}, y^{(j)})\}_{j \in \mathbb{N}}$ is generated according to Scheme I, and similar to setting of Appendix B.1, we consider the underlying probability space for the sample to be $([0,1]^{\mathbb{N}}, \mathcal{B}([0,1])^{\mathbb{N}}, \mathbb{P})$ which is already defined at the beginning of Appendix B.1. Define the random vector $\Xi$ to be the identity mapping from $[0,1]^{\mathbb{N}}$ onto itself as in Appendix B.1. Then, $\Xi$ follows the uniform distribution on $[0,1]^{\mathbb{N}}$. Suppose $\{(x^{(j)}, y^{(j)})\}_{j \in \mathbb{N}}$ is sampled according to Scheme I with $F$ defined in (15). Then, according to Bogachev (2007, Proposition 10.7.6), for each $j \in \mathbb{N}$, there exist some $(\mathcal{B}(\mathcal{X}) \otimes \mathcal{B}(S))^{j-1} \otimes \mathcal{B}([0,1])/\mathcal{B}(\mathcal{X})$-measurable function $h_X^{(j)} : (\mathcal{X} \times S)^{j-1} \times [0,1] \to \mathcal{X}$ and $(\mathcal{B}(\mathcal{X}) \otimes \mathcal{B}(S))^{j-1} \otimes \mathcal{B}(\mathcal{X}) \otimes \mathcal{B}([0,1])/\mathcal{B}(S)$-measurable function $h_Y^{(j)} : (\mathcal{X} \times S)^{j-1} \times \mathcal{X} \times [0,1] \to S$ such that $x^{(j)} = h_X^{(j)}(x^{(1)}, y^{(1)}, \ldots, x^{(j-1)}, y^{(j-1)}, \Xi^{(2j-1)})$, $y^{(j)} = h_Y^{(j)}(x^{(1)}, y^{(1)}, \ldots, x^{(j-1)}, y^{(j-1)}, x^{(j)}, \Xi^{(2j)})$, and

$$\mathbb{E}\left[ \mathbb{1}\left\{ h_Y^{(j)}(x^{(1)}, y^{(1)}, \ldots, x^{(j-1)}, y^{(j-1)}, x^{(j)}, \Xi^{(2j)}) \leqslant t \right\} \Big| \mathcal{F}_{j-1} \right] = \theta_*^\top \Phi(x^{(j)}, t) + e(x^{(j)}, t) \qquad (91)$$

for any $t \in S$ and $j \in \mathbb{N}$, where $\mathcal{F}_j := \sigma\left( \{ \Xi^{(k)} : k \in [2j+1] \} \right)$. With the same proof provided at the beginning of Appendix B.1, $\{y^{(j)}\}_{j \in \mathbb{N}}$ is $\{\mathcal{F}_j\}_{j \in \mathbb{N}}$-adapted, $x^{(j)}$ is $\mathcal{F}_{j-1}/\mathcal{B}(\mathcal{X})$-measurable, and $\Phi_j$ is $(\mathcal{F}_{j-1} \otimes \mathcal{B}(S))/\mathcal{B}([0,1]^d)$-measurable for each $j \in \mathbb{N}$. Since $e : \mathcal{X} \times S \to [-1,1]$, $(x,t) \mapsto e(x,t)$ is $\mathcal{B}(\mathcal{X}) \otimes \mathcal{B}(S)$-measurable and $x^{(j)}$ is $\mathcal{F}_{j-1}/\mathcal{B}(\mathcal{X})$-measurable, we have that $e_j : [0,1]^{\mathbb{N}} \times S \to [-1,1]$, $(\xi,t) \mapsto e(x^{(j)}(\xi),t)$ is $\mathcal{F}_{j-1} \otimes \mathcal{B}(S)$-measurable.

Define $V_j := \int_S \mathrm{I}_{y^{(j)}} \Phi_j - \int_S (\theta_*^\top \Phi_j + e_j) \Phi_j$. Since $\mathrm{I}_{y^{(j)}}$ is $\mathcal{F}_j \otimes \mathcal{B}(S)$-measurable and $e_j$ and $\Phi_j$ are $\mathcal{F}_{j-1} \otimes \mathcal{B}(S)$-measurable, by Fubini's theorem and (91), We have

$$\begin{aligned}
\mathbb{E}[V_j | \mathcal{F}_{j-1}] &= \mathbb{E}\left[ \int_S \mathrm{I}_{y^{(j)}} \Phi_j \Big| \mathcal{F}_{j-1} \right] - \int_S (\theta_*^\top \Phi_j + e_j) \Phi_j \\
&= \int_S \mathbb{E}\left[ \mathrm{I}_{y^{(j)}} | \mathcal{F}_{j-1} \right] \Phi_j - \int_S (\theta_*^\top \Phi_j + e_j) \Phi_j \\
&= \int_S (\theta_*^\top \Phi_j + e_j) \Phi_j - \int_S (\theta_*^\top \Phi_j + e_j) \Phi_j \\
&= 0.
\end{aligned}$$

For any $\alpha \in \mathbb{R}^d$, if $n = 0$, define $M_n(\alpha) = 1$. If $n \geqslant 1$, define $M_n(\alpha) := \exp\left\{ \alpha^\top W_n - \frac{1}{2} \|\alpha\|_{U_n}^2 \right\}$ for $W_n := \sum_{j=1}^n V_j$ and $U_n = \sum_{j=1}^n \int_S \Phi_j \Phi_j^\top$. Then, with the similar proof as in Appendix B.1.1, we can show that $W_n$ is $\mathcal{F}_n$-measurable, $U_n$ is $\mathcal{F}_{n-1}$-measurable, and $M_n$ is $\mathcal{F}_n \otimes \mathcal{B}(\mathbb{R}^d)$-measurable for any $n \in \mathbb{N}$. Thus, for any $\alpha \in \mathbb{R}^d$, $\{M_n(\alpha)\}_{n \geqslant 0}$ is $\{\mathcal{F}_n\}_{n \geqslant 0}$-adapted. Moreover, for any $\alpha \in \mathbb{R}^d$ and $n \in \mathbb{N}$, we have

$$\begin{aligned}
\mathbb{E}[M_n(\alpha) | \mathcal{F}_{n-1}] &= M_{n-1}(\alpha) \mathbb{E}\left[ \exp\left\{ \alpha^\top V_n - \frac{1}{2} \alpha^\top \left( \int_S \Phi_n \Phi_n^\top \right) \alpha \right\} \Big| \mathcal{F}_{n-1} \right] \\
&= M_{n-1}(\alpha) \frac{\mathbb{E}\left[ \exp\left\{ \alpha^\top V_n \right\} | \mathcal{F}_{n-1} \right]}{\exp\left\{ \frac{1}{2} \int_S (\alpha^\top \Phi_n)^2 \right\}}.
\end{aligned} \qquad (92)$$

Since $-\int_S |\alpha^\top \Phi_n| \leqslant \alpha^\top V_n \leqslant \int_S |\alpha^\top \Phi_n|$ a.s., we have

$$\mathbb{E}\left[\exp\left\{\alpha^\top V_n\right\} | \mathcal{F}_{n-1}\right] \leqslant \exp\left\{\frac{4}{8}\left(\int_S |\alpha^\top \Phi_n|\right)^2\right\}$$

$$\leqslant \exp\left\{\frac{1}{2}\int_S \left(\alpha^\top \Phi_n\right)^2\right\} \tag{93}$$

$$\leqslant \exp\left\{\frac{1}{2}\int_S \left(\alpha^\top \Phi_n\right)^2\right\} \tag{94}$$

where (93) is by Cauchy-Schwarz inequality and $\int_S 1 = \mathfrak{m}(S) = 1$. Then, by (92) and (94), we have

$$\mathbb{E}[M_n(\alpha)|\mathcal{F}_{n-1}] \leqslant M_{n-1}(\alpha)\frac{\exp\left\{\frac{1}{2}\int_S \left(\alpha^\top \Phi_n\right)^2\right\}}{\exp\left\{\frac{1}{2}\int_S \left(\alpha^\top \Phi_n\right)^2\right\}} = M_{n-1}(\alpha).$$

Thus, for any $\alpha \in \mathbb{R}^d$, $\{M_n(\alpha)\}_{n\geqslant 0}$ is a super-martingale.

Now define $\overline{M}_n := \int_{\mathbb{R}^d} M_n(\alpha)h(\alpha)d\alpha$ for

$$h(\alpha) = \left(\frac{\lambda}{2\pi}\right)^{\frac{d}{2}}\exp\left\{-\frac{\lambda}{2}\alpha^\top\alpha\right\} = \left(\frac{\lambda}{2\pi}\right)^{\frac{d}{2}}\exp\left\{-\frac{1}{2}\|\alpha\|_{\lambda I_d}^2\right\}.$$

Then, with the same calculation as (40) in Appendix B.1.1, we have $\overline{M}_n = \frac{\lambda^{d/2}}{\det(U_n(\lambda))^{1/2}}\exp\left(\frac{1}{2}\|W_n\|_{U_n(\lambda)^{-1}}^2\right)$. By Fubini's theorem, $M_n$ is $\mathcal{F}_n \otimes \mathcal{B}(\mathbb{R}^d)$-measurable implies that $\overline{M}_n$ is $\mathcal{F}_n$-measurable for any $n \geqslant 0$. With the same analysis as (42), $\{\overline{M}_n\}_{n\geqslant 0}$ is a super-martingale. By Doob's maximal inequality for super-martingales, we have that

$$\mathbb{P}\left[\sup_{n\in\mathbb{N}}\overline{M}_n \geqslant \delta\right] \leqslant \frac{\mathbb{E}[\overline{M}_0]}{\delta} = \frac{1}{\delta}$$

which implies that for any $N \in \mathbb{N}$,

$$\mathbb{P}\left[\exists n \in [N] \text{ s.t. } \|W_n\|_{U_n(\lambda)^{-1}} \geqslant \sqrt{\log\frac{\det(U_n(\lambda))}{\lambda^d} + 2\log\frac{1}{\delta}}\right] \leqslant \delta.$$

According to (47), we have

$$\|W_n\|_{U_n(\lambda)^{-1}} \leqslant \sqrt{d\log\left(1 + \frac{n}{\lambda}\right) + 2\log\frac{1}{\delta}} \tag{95}$$

for all $n \in \mathbb{N}$ with probability at least $1 - \delta$.

By (3), (44), and the definition of $V_j$, we have

$$\hat{\theta}_\lambda - \theta_* = U_n(\lambda)^{-1}\left(\sum_{j=1}^n V_j + E_n - \lambda\theta_*\right) = U_n(\lambda)^{-1}W_n + U_n(\lambda)^{-1}(E_n - \lambda\theta_*). \tag{96}$$

where $E_n = \sum_{j=1}^n \int_S e_j \Phi_j$ by definition. Thus,

$$\begin{aligned}\|\hat{\theta}_\lambda - \theta_*\|_{U_n(\lambda)} &\leqslant \|U_n(\lambda)^{-1}W_n\|_{U_n(\lambda)} + \|U_n(\lambda)^{-1}(E_n - \lambda\theta_*)\|_{U_n(\lambda)} \\ &\leqslant \|W_n\|_{U_n(\lambda)^{-1}} + \|\lambda\theta_*\|_{U_n(\lambda)^{-1}} + \|E_n\|_{U_n(\lambda)^{-1}} \\ &\leqslant \|W_n\|_{U_n(\lambda)^{-1}} + \sqrt{\lambda}\|\theta_*\| + \|E_n\|_{U_n(\lambda)^{-1}}\end{aligned} \tag{97}$$

where (97) is because of $U_n(\lambda)^{-1} = \frac{1}{\lambda}\left(I - U_n(\lambda)^{-1}U_n\right)$ and $\|I - U_n(\lambda)^{-1}U_n\|_2 \leqslant 1$.

By (95) and (97), for any $\delta \in (0, 1)$, with probability at least $1 - \delta$, we have

$$\|\hat{\theta}_\lambda - \theta_*\|_{U_n(\lambda)} \leqslant \sqrt{d\log\left(1 + \frac{n}{\lambda}\right) + 2\log\frac{1}{\delta}} + \sqrt{\lambda}\|\theta_*\| + \|E_n\|_{U_n(\lambda)^{-1}}. \tag{98}$$

for all $n \in \mathbb{N}$. Since $U_n(\lambda) - \lambda I_d$ is positive semi-definite, (98) immediately implies that

$$\|\widehat{\theta}_\lambda - \theta\|_{U_n(\lambda)} \leqslant \sqrt{d \log\left(1 + \frac{n}{\lambda}\right) + 2 \log \frac{1}{\delta}} + \sqrt{\lambda}\|\theta_*\| + \frac{1}{\sqrt{\lambda}}\|E_n\| \tag{99}$$

which is exactly (16).

$\square$

## I.2  Proof of Corollary 11

*Proof.* In the setting of Corollary 11, the sample $\{(x^{(j)}, y^{(j)})\}_{j \in \mathbb{N}}$ is generated according to Scheme II. In the following proof, we consider the underlying probability space for the sample $\{(x^{(j)}, y^{(j)})\}_{j \in \mathbb{N}}$ to be $([0,1]^{\mathbb{N}}, \mathcal{B}([0,1])^{\mathbb{N}}, \mathbb{P})$ which has already been defined at the beginning of Appendix B.1. Define the random vector $\Xi$ to be the identity mapping from $[0,1]^{\mathbb{N}}$ onto itself as in Appendix B.1. Then, $\Xi$ follows the uniform distribution on $[0,1]^{\mathbb{N}}$. Suppose $\{(x^{(j)}, y^{(j)})\}_{j \in \mathbb{N}}$ is sampled according to Scheme II with $F$ defined in (15). Then, according to Bogachev (2007, Proposition 10.7.6), for each $j \in \mathbb{N}$, there exist some $\mathcal{B}([0,1])/\mathcal{B}(\mathcal{X})$-measurable function $h_X^{(j)} : [0,1] \to \mathcal{X}$ and $\mathcal{B}(\mathcal{X}) \otimes \mathcal{B}([0,1])/\mathcal{B}(S)$-measurable function $h_Y^{(j)} : \mathcal{X} \times [0,1] \to S$ such that $x^{(j)} = h_X^{(j)}(\Xi^{(2j-1)})$, $y^{(j)} = h_Y^{(j)}(x^{(j)}, \Xi^{(2j)})$, and

$$\mathbb{E}\left[\mathbb{1}\left\{h_Y^{(j)}(x^{(j)}, \Xi^{(2j)}) \leqslant t\right\} \big| \mathcal{F}_{j-1}\right] = \theta_*^\top \Phi(x^{(j)}, t) + e(x^{(j)}, t) \tag{100}$$

for any $t \in S$ and $j \in \mathbb{N}$, where $\mathcal{F}_j := \sigma\left(\{\Xi^{(k)} : k \in [2j+1]\}\right)$. With the same proof provided at the beginning of Appendix B.1, $\{y^{(j)}\}_{j \in \mathbb{N}}$ is $\{\mathcal{F}_j\}_{j \in \mathbb{N}}$-adapted and $\Phi_j$ is $(\mathcal{F}_{j-1} \otimes \mathcal{B}(S))/\mathcal{B}([0,1]^d)$-measurable for each $j \in \mathbb{N}$. Moreover, $\{x^{(j)}\}_{j \in \mathbb{N}}$ is independent, which implies that $\{\Phi_j(t)\}_{j \in \mathbb{N}}$ is independent for any $t \in S$, $\{e_j(t)\}_{j \in \mathbb{N}}$ is independent for any $t \in S$, and $\{y^{(j)}\}_{j \in \mathbb{N}}$ is independent.

Let $b_j(t) := \mathbb{E}[e_j(t)\Phi_j(t)]$ for $t \in S$ and $j \in [n]$. Then, by Fubini's theorem, $b_j$ is measurable with $b_{ji}(t) \in [-1,1]$ for $t \in S$, $j \in \mathbb{N}$ and $i \in [d]$. By definition and Fubini's theorem, we have $B_n = \sum_{j=1}^n \int_S b_j$.

Define $V_j := \int_S \mathrm{I}_{y^{(j)}} \Phi_j - \int_S \theta_*^\top \Phi_j \Phi_j - \int_S b_j$. By Fubini's theorem and (100), we have

$$\begin{aligned}
\mathbb{E}[V_j] &= \mathbb{E}\left[\int_S (\mathrm{I}_{y^{(j)}} - \theta_*^\top \Phi_j)\Phi_j\right] - \int_S b_j \\
&= \int_S \mathbb{E}\left[(\mathrm{I}_{y^{(j)}} - \theta_*^\top \Phi_j)\Phi_j\right] - \int_S b_j \\
&= \int_S \mathbb{E}\left[\mathbb{E}[\mathrm{I}_{y^{(j)}} - \theta_*^\top \Phi_j | \mathcal{F}_{j-1}]\Phi_j\right] - \int_S b_j \\
&= \int_S \mathbb{E}\left[e_j \Phi_j\right] - \int_S b_j \\
&= \int_S b_j - \int_S b_j \\
&= 0.
\end{aligned}$$

For any $\alpha \in \mathbb{R}^d$, if $n = 0$, define $M_n(\alpha) = 1$. If $n \geqslant 1$, define $M_n(\alpha) := \exp\left\{\alpha^\top W_n - \frac{1}{2}\|\alpha\|_{U_n}^2\right\}$ for $W_n := \sum_{j=1}^n V_j$ and $U_n = \sum_{j=1}^n \int_S \Phi_j \Phi_j^\top$. Similar to Appendix I.1, we can show that $M_n$ is $\mathcal{F}_n \otimes \mathcal{B}(\mathbb{R}^d)$-measurable for any $n \geqslant 0$. Moreover, for any $n \in \mathbb{N}$,

$$\begin{aligned}
\mathbb{E}[M_n(\alpha)|\mathcal{F}_{n-1}] &= M_{n-1}(\alpha)\mathbb{E}\left[\exp\left\{\alpha^\top V_n - \frac{1}{2}\alpha^\top\left(\int_S \Phi_n \Phi_n^\top\right)\alpha\right\} \bigg| \mathcal{F}_{n-1}\right] \\
&= M_{n-1}(\alpha)\frac{\mathbb{E}\left[\exp\left\{\alpha^\top V_n\right\}|\mathcal{F}_{n-1}\right]}{\exp\left\{\frac{1}{2}\int_S (\alpha^\top \Phi_n)^2\right\}}
\end{aligned} \tag{101}$$

with $-\int_S |\alpha^\top \Phi_n| - \int_S \alpha^\top b_n \leqslant \alpha^\top V_n \leqslant \int_S |\alpha^\top \Phi_n| - \int_S \alpha^\top b_n$ a.s.. Thus,

$$
\begin{aligned}
\mathbb{E}\left[\exp\left\{\alpha^\top V_n\right\} | \mathcal{F}_{n-1}\right] &\leqslant \exp\left\{\frac{4}{8}\left(\int_S |\alpha^\top \Phi_n|\right)^2\right\} \\
&\leqslant \exp\left\{\frac{1}{2}\int_S \left(\alpha^\top \Phi_n\right)^2\right\} \\
&\leqslant \exp\left\{\frac{1}{2}\int_S \left(\alpha^\top \Phi_n\right)^2\right\}
\end{aligned}
\tag{102}
$$

Then, by (101) and (102), we have

$$
\mathbb{E}[M_n(\alpha)|\mathcal{F}_{n-1}] \leqslant M_{n-1}(\alpha)\frac{\exp\left\{\frac{1}{2}\int_S \left(\alpha^\top \Phi_n\right)^2\right\}}{\exp\left\{\frac{1}{2}\int_S \left(\alpha^\top \Phi_n\right)^2\right\}} = M_{n-1}(\alpha).
$$

Thus, for any $\alpha \in \mathbb{R}^d$, $\{M_n(\alpha)\}_{n\geqslant 0}$ is a super-martingale. With the same approach as in Appendix I.1, for any $\lambda \in (0, \infty)$, we can show that

$$
\|\widehat{\theta}_\lambda - \theta_*\|_{U_n(\lambda)} \leqslant \sqrt{d\log\left(1 + \frac{n}{\lambda}\right) + 2\log\frac{1}{\delta}} + \sqrt{\lambda}\|\theta_*\| + \frac{1}{\sqrt{\lambda}}\|B_n\|
\tag{103}
$$

for all $n \in \mathbb{N}$ with probability at least $1 - \delta$. Then, using the same analysis as in Appendix B.2.2, we can show that for any $\delta_1 \in (0,1)$, $\delta_2 \in (0, 1-\delta_1)$, and $n \geqslant \frac{32d^2}{\sigma_{\min}^2}\log(d/\delta_1)$, we have

$$
\|\widehat{\theta}_\lambda - \theta_*\|_{\Sigma_n} \leqslant \sqrt{2\left(d\log\left(1 + \frac{1}{\lambda}\right) + 2\log\frac{1}{\delta_2}\right)} + \sqrt{2\lambda}\|\theta_*\| + \sqrt{\frac{2}{\lambda}}\|B_n\|
$$

with probability at least $1 - \delta_1 - \delta_2$. Then, (17) is obtained by setting $\delta_1 = \delta_2 = \delta \in (0, 1/2)$. $\qquad\square$

## J    Proofs of the lemmas in Appendix D

In this section, we provide the proofs of the technical lemmas in Appendix D.

*Proof of Lemma 23.* For any $n \in \mathbb{N}$, since $V_j$ is $\mathcal{F}_\Omega \otimes \mathcal{F}_j$-measurable for any $j \in [n]$ and $W_n = \sum_{j=1}^n V_j$, we have that $W_n$ is also $\mathcal{F}_\Omega \otimes \mathcal{F}_j$-measurable. According to the similar arguments as in Appendix B.1.1, we know that $U_n$ is $\mathcal{F}_\Omega \otimes \mathcal{F}_{n-1}$-measurable. Thus, by Fubini's theorem, for any $\alpha \in \mathcal{L}^2(\Omega, \mathfrak{n})$, $M_n(\alpha) = \exp\left\{\langle\alpha, W_n\rangle - \frac{1}{2}\|\alpha\|_{U_n}^2\right\}$ is $\mathcal{F}_n$-measurable, which implies that $\{M_n(\alpha)\}_{n\geqslant 0}$ is $\{\mathcal{F}_n\}_{n\geqslant 0}$-adapted. For any $n \in \mathbb{N}$, we have that

$$
\begin{aligned}
\mathbb{E}[M_n(\alpha)|\mathcal{F}_{n-1}] &= M_{n-1}(\alpha)\mathbb{E}\left[\exp\left\{\langle\alpha, V_n\rangle - \frac{1}{2}\int_S \Psi_n(\alpha, t)^2 \mathfrak{m}(dt)\right\} \Big| \mathcal{F}_{n-1}\right] \\
&= M_{n-1}(\alpha)\frac{\mathbb{E}\left[\exp\left\{\langle\alpha, V_n\rangle\right\} | \mathcal{F}_{n-1}\right]}{\exp\left\{\frac{1}{2}\int_S \Psi_n(\alpha, t)^2 \mathfrak{m}(dt)\right\}}.
\end{aligned}
$$

Since $-\int_S |\Psi_n(\alpha, t)|\mathfrak{m}(dt) \leqslant \langle\alpha, V_n\rangle \leqslant \int_S |\Psi_n(\alpha, t)|\mathfrak{m}(dt)$, according to Hoeffding's lemma (Hoeffding, 1963) and Cauchy-Schwarz inequality, we have

$$
\begin{aligned}
\mathbb{E}\left[\exp\left\{\langle\alpha, V_n\rangle\right\} | \mathcal{F}_{n-1}\right] &\leqslant \exp\left\{\frac{4}{8}\left(\int_S |\Psi_n(\alpha, t)|\mathfrak{m}(dt)\right)^2\right\} \\
&\leqslant \exp\left\{\frac{1}{2}\int_S \Psi_n(\alpha, t)^2 \mathfrak{m}(dt)\right\} \\
&\leqslant \exp\left\{\frac{1}{2}\int_S \Psi_n(\alpha, t)^2 \mathfrak{m}(dt)\right\}.
\end{aligned}
$$

Then, we have

$$\mathbb{E}[M_n(\alpha)|\mathcal{F}_{n-1}] \leqslant M_{n-1}(\alpha)\frac{\exp\left\{\frac{1}{2}\int_S \Psi_n(\alpha,t)^2\mathfrak{m}(dt)\right\}}{\exp\left\{\frac{1}{2}\int_S \Psi_n(\alpha,t)^2\mathfrak{m}(dt)\right\}} = M_{n-1}(\alpha).$$

Since $M_0(\alpha) = 1$ and $M_n(\alpha) \geqslant 0$, for any $\alpha \in \mathcal{L}^2(\Omega,\mathfrak{n})$, $\{M_n(\alpha)\}_{n\geqslant 0}$ is a non-negative super-martingale. $\square$

*Proof of Lemma 24.* For any $m \in \mathbb{N}$, we have

$$|H_\infty - H_m| = \left|\sum_{i=m+1}^\infty \left(\beta_i w_i' - \frac{1}{2}\lambda_i\beta_i^2\right)\right|$$

$$\leqslant \sqrt{\sum_{i=m+1}^\infty \beta_i^2}\sqrt{\sum_{i=m+1}^\infty (w_i')^2} + n\mathfrak{n}(\Omega)\sum_{i=m+1}^\infty \beta_i^2.$$

Since $\sum_{i=1}^\infty (w_i')^2 = \|W_n\|^2 < \infty$ and $\sum_{i=1}^\infty \beta_i^2 < \infty$ a.s., we have that $\lim_{m\to\infty}|H_m - H_\infty| = 0$ a.s.. Thus,

$$\lim_{m\to\infty}|\exp(H_m) - M_n(\beta)| = \lim_{m\to\infty}|\exp(H_m) - \exp(H_\infty)| = 0.$$

Since $|W_n| \leqslant n$ a.s., we have

$$|\exp(H_m)| \leqslant \exp\left(|\langle\beta, W_n\rangle| + \frac{1}{2}\|\beta\|_{U_n}^2\right)$$

$$\leqslant \exp\left(n\sum_{i=1}^\infty |\sigma_i\zeta_i| + \frac{1}{2}\sum_{i=1}^\infty \lambda_i\sigma_i^2\zeta_i^2\right)$$

for all $m \in \mathbb{N}$ a.s..

Moreover, for any $m \in \mathbb{N}$, by the independence of $\{\zeta_i\}_{i\in\mathbb{N}}$, if $|\sigma_i| < \frac{1}{\sqrt{\lambda_i}}$ for all $i \in \mathbb{N}$, then we have

$$\mathbb{E}\left[\exp\left(n\sum_{i=1}^m |\sigma_i\zeta_i| + \frac{1}{2}\sum_{i=1}^m \lambda_i\sigma_i^2\zeta_i^2\right)\right]$$

$$= \prod_{i=1}^m \mathbb{E}\left[\exp\left(|\sigma_i\zeta_i| + \frac{1}{2}\lambda_i\sigma_i^2\zeta_i^2\right)\right]$$

$$= \prod_{i=1}^m \exp\left(n|\sigma_i\zeta_i| + \frac{1}{2}(\lambda_i\sigma_i^2 - 1)\zeta_i^2\right)\frac{d\zeta_i}{\sqrt{2\pi}}$$

$$= \prod_{i=1}^m \frac{2}{\sqrt{1 - \lambda_i\sigma_i^2}}\exp\left(\frac{n^2\sigma_i^2}{2(1 - \lambda_i\sigma_i^2)}\right)\Phi_{N(0,1)}\left(\frac{n|\sigma_i|}{\sqrt{1 - \lambda_i\sigma_i^2}}\right)$$

where $\Phi_{N(0,1)}$ denotes the CDF of the $N(0,1)$ distribution. By the monotone convergence theorem, we have

$$\mathbb{E}\left[\exp\left(n\sum_{i=1}^\infty |\sigma_i\zeta_i| + \frac{1}{2}\sum_{i=1}^\infty \lambda_i\sigma_i^2\zeta_i^2\right)\right]$$

$$= \prod_{i=1}^\infty \frac{2}{\sqrt{1 - \lambda_i\sigma_i^2}}\exp\left(\frac{n^2\sigma_i^2}{2(1 - \lambda_i\sigma_i^2)}\right)\Phi_{N(0,1)}\left(\frac{n|\sigma_i|}{\sqrt{1 - \lambda_i\sigma_i^2}}\right)$$

$$\leqslant \frac{1}{\sqrt{\prod_{i=1}^\infty(1 - \lambda_i\sigma_i^2)}}\exp\left(\frac{n^2}{2}\sum_{i=1}^\infty \frac{\sigma_i^2}{1 - \lambda_i\sigma_i^2}\right)\prod_{i=1}^\infty\left[1 + 2\Phi_{N(0,1)}\left(\frac{n|\sigma_i|}{\sqrt{1 - \lambda_i\sigma_i^2}}\right) - 1\right]$$

Since $\lim_{i\to\infty}\lambda_i\sigma_i^2 = 0$ and $\sum_{i=1}^{\infty}\sigma_i^2 < \infty$, we have

$$\sum_{i=1}^{\infty}\frac{\sigma_i^2}{1-\lambda_i\sigma_i^2} < \infty \text{ and } \sum_{i=1}^{\infty}\lambda_i\sigma_i^2 < \infty,$$

which also implies that

$$\prod_{i=1}^{\infty}(1-\lambda_i\sigma_i^2) < \infty.$$

For any sequence $\{a_i\}_{i\in\mathbb{N}}$ such that $a_i > 0$ and $\sum_{i=1}^{\infty}a_i < \infty$, we have $\lim_{i\to\infty}a_i = 0$ and

$$\lim_{i\to\infty}\frac{2\Phi_{N(0,1)}(a_i)-1}{a_i} = \lim_{i\to\infty}\frac{1}{\sqrt{\pi}}\frac{\exp\left(-\frac{a_i^2}{2}\right)(a_i + o(a_i))}{a_i} = \frac{1}{\sqrt{\pi}}.$$

Since $\sum_{i=1}^{\infty}|a_i| < \infty$, we can conclude that

$$\sum_{i=1}^{\infty}(2\Phi_{N(0,1)}(a_i)-1) < \infty.$$

Therefore, if we assume that $\sum_{i=1}^{\infty}|\sigma_i| < \infty$, we have $\sum_{i=1}^{\infty}\frac{n|\sigma_i|}{\sqrt{1-\lambda_i\sigma_i^2}} < \infty$ and

$$\prod_{i=1}^{\infty}\left[1 + 2\Phi_{N(0,1)}\left(\frac{n|\sigma_i|}{\sqrt{1-\lambda_i\sigma_i^2}}\right) - 1\right] < \infty.$$

In conclusion, we have

$$\mathbb{E}\left[\exp\left(n\sum_{i=1}^{\infty}|\sigma_i\zeta_i| + \frac{1}{2}\sum_{i=1}^{\infty}\lambda_i\sigma_i^2\zeta_i^2\right)\right] < \infty.$$

Then, by the conditional dominated convergence theorem, we have

$$\mathbb{E}[\exp(H_m)|\mathcal{F}_\infty] \to \mathbb{E}[M_n(\beta)|\mathcal{F}_\infty] = \overline{M}_n$$

as $m \to \infty$ a.s.. $\qquad\square$

