# OpenReview forum: "Functional Linear Regression of Cumulative Distribution Functions"
_TMLR — Accepted by TMLR_

### Review · Reviewer_9aWg · 2023-11-23

**Summary Of Contributions:**

This paper studies contextual CDF estimation under a linear model:

1. The proposed estimators for the coefficients have \tilde{O}(\sqrt{d/n}) upper bounds, both in the adversarial and random settings.

2. The authors also provided a lower-bound analysis, showing their upper error bounds are reasonable, i.e., the bounds are tight up to logarithmic factors.

3. Acknowledging the limitations of the linear model assumptions, they analyzed the effect of model misspecification.

4. Numerical experiments demonstrate the applicability of the proposed estimators.

**Audience:**

Yes

**Broader Impact Concerns:**

None.

**Claims And Evidence:**

Yes

**Requested Changes:**

1. Can the authors specify why is it not possible to estimate a CDF using quantile regression? Considering a continuous response, one can estimate a range of quantiles, e.g., 0.01,…,0.99, to form the estimated CDF. This approach has been taken by various works.

2. It’s hard to understand Figure 1, please improve it.

3. I have reservations about the objective function in Eq. (2). First, it seems unnatural to minimize the L2 distance between the indicator function and the estimated CDF. Can the authors explain this choice, why not use a link function? Second, \hat{\theta} is improper (as the authors noted). It is not clear to me why the authors focus on minimizing this objective in the first place.

3. In the experiments, I suggest comparing the KS distance and L2 error to CDF estimation to baseline “distribution regression” methods, even if these are only supported by pointwise guarantees.

**Strengths And Weaknesses:**

*Strengths:*

See contributions above. While I'm not an expert in this specific area, I found the analysis thorough (including minimax analysis, studying the effect of modeling errors, burn-in time, and more).

*Weaknesses:*

See the requested changes below. My major concern is with the objective function in Eq. (2).

---

> ### Author Response · Authors · 2023-12-15
> **Response to the first and the second points**
>
> Thank you very much for your valuable time and review. We have revised our paper where the added contents are marked in red. Our responses and the detailed explanations of the revisions are provided below (the page/equation/theorem numbers used below refer to those in the current version of the paper).
>
> ## Response to the first point in requested changes
>
> We have discussed the drawbacks of quantile regression in related works of our paper. In the following, we explain more on why not estimating multiple quantiles to construct CDFs.
>
> 1. The estimation in quantile regression is pointwise. We can use quantile regression to estimate multiple quantiles, but theoretically, it requires a simultaneous analysis of multiple estimates, which typically requires a union bound on the failure probability that increases linearly with the number of estimates.
>
> 2. The estimated quantiles may not be monotonically increasing with respect to the probability values. It requires extra effort to construct a valid CDF from a finite series of quantile estimates. Moreover, any such construction will incur a non-convergent KS distance between the estimated CDF and the true CDF for some distribution, as a general CDF can exhibit jumps or flat regions at any position.
>
> 3. The quality of the estimated CDF from multiple estimated quantiles relies heavily on the selection of grid points of probability values, which is instance-dependent and may require knowledge of the distribution the learner seeks to estimate. Thus, establishing a universal rule for choosing grid points that yield reasonable CDF estimates via quantile regression across diverse distributions proves challenging. In practice, the introduction of grid points introduces numerous hyperparameters to tune, adding artificial complexity to the methodology.
>
> 4. Additionally, the performance of quantile regression can be poor for discontinuous CDFs, emphasizing a limitation in its performance under such distributions.
>
> **We have incorporated the above points (marked in red) to page 2 of the revised paper.**
>
> ## Response to the second point in requested changes
>
> Thank you for pointing out this. In our revised paper, we have reorganized the layout of Figure 1 to make it more readable. Now, each column corresponds to a sample, where the first row displays the contextual CDFs and the second row displays the one-sample empirical CDFs. We have also adjusted the explanation in the caption of Figure 1 accordingly (marked in red).

---

> ### Author Response · Authors · 2023-12-15
> **Response to the third and the fourth points**
>
> ## Response to the third point in requested changes
>
> For the first reservation, we believe it is natural to minimize the $\mathcal{L}^2$-distance between the indicator function and the estimated CDF for the following reasons.
>
> 1. Researchers have considered the $\mathcal{L}^2$-distance between an indicator function and a CDF in the definition of Continuous Ranked Probability Score (CRPS) (Hersbach, 2000) to assess the performance of the CDF in approximating data distribution (see page 20 in our paper).
>
> 2. As the first work in functional linear regression of CDFs, our paper minimizes the $\mathcal{L}^2$-distance in Eq. (2) analogous to the minimization of the $\ell^2$-distance between the response and linear transformations of the feature in canonical linear regression. Viewing the indicator function as the response and the basis contextual CDFs as the feature in linear regression, it is natural to start with the least squares method, precisely corresponding to minimizing the $\mathcal{L}^2$-distance in our functional setting.
>
> 3. Using the $\mathcal{L}^2$-distance in the objective function in Eq. (2) enables the derivation of a closed-form analytic solution for our estimator $\widehat{\theta}_{\lambda}$ in Eq. (3), facilitating theoretical analyses of the estimation error.
>
> 4. The employment of link functions constrains the estimated CDFs to some specific families, limiting the application of the estimates for general CDFs. Under our linear model Eq. (1), as we have argued above, it is more natural to extend the least squares method in linear regression by minimizing the $\mathcal{L}^2$-error in Eq. (2) than to use a link function that further imposes artificial structure on our model. Furthermore, as we have mentioned at the beginning of page 3, CDFs based on link functions can be integrated into the family of feature contextual CDFs in our method.
>
> For the second reservation, we have established in page 6 that the projection of the improper estimator to the probability simplex has smaller estimation error than the improper estimator. Therefore, an upper bound on the estimation error of the improper estimator directly yields an upper bound on the error of the projected estimator. As we have mentioned earlier, the improper estimator has a closed-form analytic solution in Eq. (3) which benefits the analysis of the estimation error. Thus, we focus on bounding the estimation error of this improper estimator, consequently bounding the estimation error of the projected proper estimator.
>
> **We have incorporated the above points (marked in red) to page 5 and 6 of the revised paper.**
>
>
> ## Response to the fourth point in requested changes
>
> The term “distribution regression” lacks a standard definition as it refers to distinct learning problems across different literature, thereby precluding the existence of a baseline method for “distribution regression”. For example, in our related works, the “distribution regression” problem studied in Chernozhukov et al. (2013) and Koenker et al. (2013) is different from that studied in Póczos et al. (2013) and Szabó et al. (2016). As we have mentioned in related works, our model is different from the notion of distribution regression in Póczos et al. (2013) and Szabó et al. (2016).
>
> Chernozhukov et al. (2013) and Koenker et al. (2013) maximize the likelihood of the indicators $\mathbb{1}\lbrace y\ge Y_j\rbrace$ for samples $Y_1,\dots,Y_n$ and a fixed $y$, rather than maximizing the likelihood of the samples $Y_1,\dots,Y_n$ themselves, which is not a natural way of applying MLE. In related works, we have also discussed the drawbacks of using a single $y$ in their methods. Thus, we believe that their methods are not proper baselines for our model. In our numerical experiments, we have already conducted comparisons with the standard MLE for the samples $Y_1,\dots,Y_n$ as the baseline.
>
> Moreover, given the predominantly theoretical nature of our paper, we believe that we have provided sufficient experimental results that demonstrate the scaling of the estimation error as well as the practical performance of our method.

---

> > ### Comment · Reviewer_9aWg · 2024-01-19
> > **Follow-up**
> >
> > Thank you very much for your detailed response.
> >
> > Your points are well taken! I recommend accepting the paper.

---

### Review · Reviewer_wsMf · 2023-11-25

**Summary Of Contributions:**

This manuscript considered an estimation of the CDF for a mixture model with known basis. The authors proved several theoretical results for the finite basis w/(o) realizability, as well as the result for infinite basis. Empirical results are also provided.

**Audience:**

Yes

**Broader Impact Concerns:**

N/A.

**Claims And Evidence:**

Yes

**Requested Changes:**

* Please discuss the technical novelty, compared with Yasin’s work as well as recent work on linear mixture mdp.
* Please discuss the potential application of the work, beyond the estimation of the mixture model.

**Strengths And Weaknesses:**

###  Strengths

* A complete story.

### Weaknesses

* I don’t feel the theoretical results quite novel. In fact, beyond Yasin’s pioneering work, the recent work on linear mixture mdp (e.g. [1]) also have studied this problem under the realizability assumption. I don’t see lots of new technical insights. The unrealizable case can be one, but the infinite basis case is just an extension to the RKHs (although authors do not formally mention it).

* I don’t quite get the intuition on such problem, which makes me confusing about the potential application of the work. Estimating the mixture model may be the only case (just as [1]). I also don’t agree with the argument on MLE in Section 3.1. With the basis CDF and the parameter set, it is also feasible to define the likelihood and perform MLE, although the guarantee is on the TV distance, which is substantially weaker than the KS distance due to not utilizing the fact that the basis is a CDF.

[1] Zhou, Dongruo, Quanquan Gu, and Csaba Szepesvari. "Nearly minimax optimal reinforcement learning for linear mixture markov decision processes." Conference on Learning Theory. PMLR, 2021.

---

> ### Author Response · Authors · 2023-12-15
> **Response to Weakness 1**
>
> Thank you very much for your valuable time and review. **We have revised our paper where the revisions are marked in red**. Our responses and the detailed explanations of the revisions are provided below (the page/equation/theorem numbers used below refer to those in the current version of the paper).
>
> ## Response to Weakness 1
>
> First, we quote the following sentences from the criteria of TMLR [Link](https://jmlr.org/tmlr/acceptance-criteria.html):
>
> >Crucially, it should not be used as a reason to reject work that isn't considered “significant” or “impactful” because it isn't achieving a new state-of-the-art on some benchmark. Nor should it form the basis for rejecting work on a method considered not “novel enough”, as novelty of the studied method is not a necessary criteria for acceptance. We explicitly avoid these terms (“significant”, “impactful”, “novel”), and focus instead on the notion of “interest”. If the authors make it clear that there is something to be learned by some researchers in their area from their work, then the criterion of interest is considered satisfied.
>
> As we have stated in the introduction, CDF estimation is a fundamental problem in machine learning with broad applications in domains such as risk assessment, contextual bandits, and treatment effects, where it is especially crucial to simultaneously learn CDFs of multiple random variables. In this paper, we are the first to propose functional linear regression of CDFs and provide a complete analysis for our method in various settings, which makes a noteworthy contribution to the aforementioned problem of CDF estimation. Thus, we believe that our work satisfies the criteria of TMLR quoted above.
>
> As we have mentioned in page 8 and 9, distinguishing our work from existing literature [1] and [2] which study ridge regression for finite dimensional feature vectors and scalar responses, we focus on the functional setting where not only the features and responses are both functions, but the parameter is also a function in our infinite-dimensional parameter model (Eq. (18)). Consequently, the theoretical results of [1] and [2] are not directly applicable to our estimator, necessitating substantial efforts to extend the techniques used in their proof to our setting.
>
> Moreover, we draw attention to a limitation in the upper bounds of estimation errors established by [1] and [2]. These bounds are susceptible to the magnitudes of the responses, with [1] growing linearly with the upper bound and standard deviation of the noise term, and [2] growing linearly with the sub-Gaussian parameter of the noise term. In contrast, our proposed model and estimator offer an advantage: our upper bound for estimation error remains independent of response scales. This independence constitutes a notable benefit, distinguishing our linear model from those explored in previous works.
>
> In addition to the upper bound of the estimation error of the regularized estimator, we also prove the upper bound of the estimation error of unregularized estimators (with and without burn-in time), the lower bound of the estimation error, the upper bound of the estimation error in the unrealizable setting, and the upper bound of the estimation error for infinite dimensional parameter. Those results are not provided in [1] and [2] and illustrate substantial technical novelties in our analyses.
>
> **We have added a summary of the discussion above on the comparison to [1] and [2] to page 8 and 9 of the revised paper, where the added contents are marked in red.**
>
> It is worth mentioning that our theoretical result in the infinite dimensional parameter model (Eq. (18)) is _not_ “just an extension to the RKHS”. In our proof, we do not use the techniques for RKHS. Our parameter is a function living in a separable Hilbert space $\mathcal H_{\boldsymbol{\sigma}, \boldsymbol{e}}$ (see the equation below Eq. (20)), which is _not_ an RKHS. Besides, it is important to highlight that $\Phi$ is _not_ a kernel. The domain of $\theta_{*}$, $\Omega$ (a space equipped with a finite measure) is different from the context domain $\mathcal{X}$ (a Polish space) and the domain of the CDF $\mathbb{R}$. In our proof sketch of Theorem 19 (page 15 and 16), we have provided insight into the technical challenges we successfully addressed.
>
> [1] Dongruo Zhou, Quanquan Gu, and Csaba Szepesvari. "Nearly minimax optimal reinforcement learning for linear mixture markov decision processes." Conference on Learning Theory. PMLR, 2021.
>
> [2] Yasin Abbasi-Yadkori, Dávid Pál, and Csaba Szepesvári. Improved algorithms for linear stochastic bandits. Advances in neural information processing systems, 24, 2011.

---

> ### Author Response · Authors · 2023-12-15
> **Response to Weakness 2**
>
> ## Response to Weakness 2
>
> As we have explained in the beginning of the introduction, one intuition of the problem of functional regression of CDFs is rooted in risk assessment [1,2,3,4]. In risk assessment, researchers focus on assessing various risk functionals which map CDFs of target random variables to risk values. For example, in contextual bandits and MDPs [5,6,7], researchers may seek to evaluate not only the value function corresponding to the expected rewards but also other risk functionals of the CDFs of the rewards. For the family of Lipschitz risk functionals, an accurate estimation of the CDF leads to accurate estimations of risk values across all such functionals [1]. Thus, accurate CDF estimation stands as a foundational element for effective risk assessment. Moreover, given that risk assessment often involves the estimation of CDFs for multiple (potentially infinite) random variables, analogous to canonical linear regression, we introduce linear models Eq. (1) and Eq. (18) and propose different estimators for the functional linear regression problem in different settings, which marks an initial step towards developing general CDF estimation methods that fulfill the needs of the aforementioned learning problems.
>
> Furthermore, as we have also discussed in the related works, our estimation method accommodates the integration of existing CDF estimation methods into the family of basis contextual CDFs in our model, thereby enhancing the overall quality of the contextual CDF estimates.
>
> We believe that the likelihood of a random variable cannot be defined directly from its CDF.  As discussed in Section 3.1, the PDFs/PMFs of the basis distributions may not exist and the distributions of the contexts are usually unknown in Scheme I and II, which makes it unattainable to define the likelihood in the general adversarial or random setting. Even though it is possible to define the likelihood for the indicators $\mathbb{1}\lbrace y\ge Y_j \rbrace$ for samples $Y_1,\dots,Y_n$ and fixed cutoff $y$, which is the “distribution regression” method [8] we have discussed in related works (page 3), the likelihood is _not_ defined for the random variables $Y_1,\dots,Y_n$ themselves and it is not a natural way of applying MLE. As we have pointed out in related works, the MLE used here only accesses the indicators denoting whether the samples $Y_1,\dots, Y_n$ surpass a fixed cutoff $y$, which under-utilizes the wealth of information inherent in the samples. In practice, the performance of the MLE used here may heavily rely on the choice of the cutoff $y$, especially when the information on the support of the CDFs is unknown. Our estimator (Eq. (2) or Eq. (21)) which evaluates the whole indicator function does not have this shortcoming and can fully exploit the sample information.
>
> Regarding the reviewer's comment on "the guarantee is on the TV distance", we seek further clarification on this aspect to ensure a more thorough understanding. We appreciate additional insights the reviewer can provide in this regard.
>
> [1] Huang, Audrey, et al. "Off-policy risk assessment in contextual bandits." Advances in Neural Information Processing Systems 34 (2021): 23714-23726.
>
> [2] Sani, Amir, Alessandro Lazaric, and Rémi Munos. "Risk-aversion in multi-armed bandits." Advances in neural information processing systems 25 (2012).
>
> [3] Shapiro, Alexander, Darinka Dentcheva, and Andrzej Ruszczynski. Lectures on stochastic programming: modeling and theory. Society for Industrial and Applied Mathematics, 2021.
>
> [4] Denneberg, Dieter. "Premium calculation: Why standard deviation should be replaced by absolute deviation1." ASTIN Bulletin: The Journal of the IAA 20.2 (1990): 181-190.
>
> [5] Huang, Audrey, et al. "Off-Policy Risk Assessment for Markov Decision Processes." International Conference on Artificial Intelligence and Statistics. PMLR, 2022.
>
> [6] Huang, Audrey, et al. "On the convergence and optimality of policy gradient for Markov coherent risk." arXiv preprint arXiv:2103.02827 (2021).
>
> [7] Tamar, Aviv, et al. "Policy gradient for coherent risk measures." Advances in neural information processing systems 28 (2015).
>
> [8] Chernozhukov, Victor, Iván Fernández‐Val, and Blaise Melly. "Inference on counterfactual distributions." Econometrica 81.6 (2013): 2205-2268.

---

### Review · Reviewer_4rEh · 2024-01-04

**Summary Of Contributions:**

This paper presents algorithms and lower bounds for the problem of regression of contextual CDFs. Each data point is assumed to be sampled from a mixture of context dependent and known CDF basis functions. A functional ridge-regression based algorithm is proposed to estimate the mixture weights, leading to a CDF estimation algorithm. The KS error of the resulting estimate is assumed to scale as $\widetilde{O} (\sqrt{d/n})$ both when the contexts are adversarial as well as in the random design setting. These results follow from self-normalized error bounds established for the algorithm. The bounds are shown to be information theoretically optimal, and the results are extended to misspecified setting where the true CDF is not an exact mixture of the basis CDF functions where a self-normalized error bound is again established. These results are finally extended to infinite dimensional models where the parameter space is an infinite dimensional Hilbert space, which generalizes the previous results in the paper. Finally, numerical results demonstrate that the theoretical analysis tracks empirical behavior well - the estimators are also shown to perform competitively on real-world data under suitably chosen bases.

**Audience:**

Yes

**Claims And Evidence:**

Yes

**Requested Changes:**

My suggestions are listed in the previous section.

**Strengths And Weaknesses:**

The presentation in the paper is clean, and to the extent I checked seems technically sound. The results are novel and interesting, and it's pleasant to see an empirical analysis of the proposed algorithm. As the authors point out, results of a similar flavor are present for adaptive linear regression (Abbasi-Yadkori et al. 2011, for instance). However, there is a significant difference between this setting and the case of CDF estimation. While the observation for the linear regression are (noisy) estimates of the function at the queried context, the right way to view the response in the distribution estimation setting are empirical CDFs, which are infinite dimensional objects. Thus, it is a-priori unclear how to use the analysis of the previous ridge-regression based algorithms for online regression. Overall, I think the results are well motivated and interesting.

I would have appreciated 2 kinds of discussion in the paper:

1. Are there any downstream applications of this kind of a result for stochastic bandits? Stochastic bandit analyses which are based on online regression combine two components - (i) an online regression algorithm, and (ii) an exploration algorithm to choose actions. The results in this paper with adversarial contexts seem to provide an analysis for the first component. A related stochastic bandit paper that comes to mind is the setting of "infinite armed bandits" (see for instance, https://arxiv.org/pdf/2211.01743.pdf).

2. What can be said in the adaptive setting where the basis CDFs are not known (but are assumed to be learnable)? Here, the algorithm can query any $i \in [d]$ and get a sample from the corresponding CDF $\phi_i (x,\cdot)$. The learner's objective is therefore to balance learning the individual CDFs, alongside with learning the mixture weights; the problem is non-trivial even with a single context. Similar results have recently been established for classification when the contexts are randomly sampled: see https://arxiv.org/abs/2307.12135 and references therein. While I believe a full solution to this problem may detract from the existing results in the paper, I think it is worth including a comment about this very natural setting.

Finally, a question and a minor comment:

1. I am a bit confused by the results about eliminating the burn-in cost. The burn-in cost that is eliminated by the new algorithm in section 3.3 is $\frac{32 d^2}{\sigma_{\min }^2} \log \left(\frac{d}{\delta}\right)$. However, the error scaling of the new algorithm is $\frac{1}{\mu_{\min }\left(\Sigma_n\right)} 2 d \sqrt{8 n \log (d / \delta)} || \theta_* || + \cdots$ which requires $n \gtrsim \frac{d^2}{\mu_{\min }\left(\Sigma_n\right)^2} \log \left(\frac{d}{\delta}\right)$ for a non-trivial estimation error guarantee. What is this section trying to achieve?

2. There is a typo on the line below eq. (11).

Overall, I am supportive of accepting this paper to the TMLR.

---

> ### Author Response · Authors · 2024-01-19
> **Response to the suggested discussions**
>
> Thank you very much for your valuable time and review. **We have revised our paper where the revisions are marked in orange.** Our responses and the detailed explanations of the revisions are provided below.
>
> ## Response to the suggested discussions
>
> We express our gratitude to the reviewer for pointing out the two connections between our work and other research topics. **We have added the first discussion to the related work section between page 3 and 4 of the revised paper and added the second discussion to the conclusion section on page 22 of the revised paper.**
>
> 1. For the first point, our work does have downstream applications in stochastic bandits. As highlighted by the reviewer, our method can serve as the online regression/estimation algorithm in stochastic bandits with our theoretical results ready to be integrated in the analysis. In the infinite-armed bandit problem [1] mentioned by the reviewer, assuming that the underlying distribution of the arms satisfies our linear model in eq. (1), our method, in conjunction with an exploration algorithm for arm selection, can be employed to estimate the CDF of the underlying distribution, which actually enables the estimation of any distribution functional, thereby extending the class of indicator-based functionals considered in [1].
> Moreover, as stated in the introduction, CDF estimation is sufficient for risk assessment, which also has application in stochastic bandits. Specifically, in [2], which we cite in the introduction, estimating the CDF of the reward under a target policy in stochastic bandits is adequate for assessing various risk functionals associated with the target policy. Consequently, with a linear assumption on the reward distribution, our method can be applied to the risk assessment of policies in stochastic bandits. Then, combined with an exploration algorithm to choose policies, our method can be used to minimize different risks in stochastic bandit problems, which also extends the conventional scope of minimizing expected regret in stochastic bandits.
>
> 2. For the second point, we believe that it is a promising future direction to consider the adaptive setting where one must also learn the basis CDFs. Drawing inspiration from [3], we believe it is possible to formalize this problem within the PAC framework as follows. Suppose that there are $d+1$ oracles $O^{1},\dots,O^{d+1}$ where $O^i$ yields a context $x$ and samples an instance from $\phi_i(x,\cdot)$ for each $i\in[d]$, while $O^{d+1}$ provides a context $x$ and samples an instance from $\theta^\top_*\Phi(x,\cdot)$. In round $t\in\mathbb{N}$, the learner chooses $i^{(t)}\in[d+1]$ to query oracle $O^{i}$ and receives $(x^{(t)},y^{(t)})$. After $n$ rounds, the learner produces an estimate $\widehat\theta$ of $\theta_*$. The objective is to study the optimal sample complexity of estimating $\theta_*$ with the estimation error bounded by $\epsilon>0$ and a probability at least $1-\delta$ for $\delta\in(0,1)$. Please be mindful that this is a primitive thought and rigorous consideration is required to verify these ideas.
> Leveraging our existing results on estimating $\theta_*$ based on $\phi_i$’s, we can decompose this problem into two subproblems: (i) the estimation of contextual CDF $\phi_i$ based on its samples and (ii) the planning of the sequence $(i^{(1)},\dots,i^{(n)})$. As we have discussed in related works, there are numerous existing results on the estimation of a single contextual CDF based on certain distributional assumptions. Thus, the primary challenge lies in addressing (ii). A natural idea is to initiate from the offline algorithm, akin to [1]. The learner queries $O^i$ $n_i$ times for each $i\in[d+1]$, uses $n_i$ samples to estimate $\phi_i$ for each $i\in[d]$, and finally, employs $n_{d+1}$ to estimate $\theta_*$ based on the estimated $\phi_i$’s. Subsequently, we optimize the upper bound of the estimation error wrt the choice of $n_i$’s under the constraint that their sum is $n$. Though this approach is straightforward, we raise the conjecture of whether the offline algorithm is optimal, as it appears that the learning of $\theta_*$ may not contribute much to the learning of each individual basis CDF. Moving forward, we can explore online algorithms which adaptively determine the next oracle to query, for which the design of the online algorithms in [1] and [3] will definitely provide valuable insights.
>
> [1] Wang, Yifei, et al. "Beyond the Best: Estimating Distribution Functionals in Infinite-Armed Bandits." arXiv preprint arXiv:2211.01743 (2022).
>
> [2] Audrey Huang, Leqi Liu, Zachary Lipton, and Kamyar Azizzadenesheli. Off-policy risk assessment in contextual bandits. Advances in Neural Information Processing Systems, 34, 2021.
>
> [3] Awasthi, Pranjal, Nika Haghtalab, and Eric Zhao. "Open problem: The sample complexity of multi-distribution learning for VC classes." The Thirty Sixth Annual Conference on Learning Theory. PMLR, 2023.

---

> ### Author Response · Authors · 2024-01-19
> **Response to the question and the minor comment**
>
> ## Response to the question
>
> We thank the reviewer for this valuable observation. We acknowledge that for proper estimators, the upper bound in Theorem 7 is not useful for $n\lesssim \frac{d^2}{\sigma_{\min}^2}\log(\frac{d}{\delta})$ which is the scaling of the burn-in time in Theorem 4 and Proposition 5. **We have added this point (marked in orange) to page 10 of the revised paper.**
>
> Nevertheless, the result in Theorem 7 still furnishes an upper bound for the estimation error of the improper estimator $\check\theta_{\lambda}$ in eq. (10) for any $n\in\mathbb{N}$ even when $n<\frac{32d^2}{\sigma_{\min}^2}\log(\frac{d}{\delta})$. In contrast, for the improper estimator $\widehat\theta_{\lambda}$ in eq. (2), we do not have an upper bound of its estimation error in Theorem 4 and Proposition 5 when $n<\frac{32d^2}{\sigma_{\min}^2}\log(\frac{d}{\delta})$. It is noteworthy that the estimation error of an improper estimator can be arbitrarily large. For instance, the estimation error of $\widehat{\theta}_{\lambda}$ can become arbitrarily large as $\lambda$ approaches zero.
>
>
> ## Response to the minor comment
>
> We thank the reviewer for pointing out this typo. **We have corrected the typo on the line below eq. (11).**

---

### Decision · Action_Editor_Sgtu · 2024-02-27

**Recommendation:** Accept with minor revision

**Comment:**

All reviewers agree that the results are sound and of interest to the community. A main concern was that the results are mostly an direct adaptation of the self-normalized concentration techniques developed in the stochastic linear bandits literature, e.g. Abbasi-Yadkori et al. (2011ab). However, given the soundness and completeness of the result, I think the paper could be a good addition to TMLR.

I encourage the authors to further emphasize the connections between the techniques used in this paper and the existing self-normalized concentration techniques. In particular, the added discussions on Page 8-9 about why infinite-dimensional CDFs introduce additional technical challenges could be expanded (the only difference in Eq(3) from standard linear regression seems to be integrating the features/covariances over $dm$, but why that brings additional challenge). A separate point worth further discussion is the relationship between the linear mixture MDP models (as mentioned by Reviewer wsMf), as there the state transition is also a linear mixture of known basis CDFs.

**Audience:**

The results could be of interest to the statistical learning theory and bandits community.

**Claims And Evidence:**

This paper studies the problem of estimating cumulative distribution functions (CDFs) from a known set of basis CDFs. The paper develops fast-rate results for ridge regression based estimators using self-normalized concentration techniques (both Hoeffding type and Bernstein type). The results and proofs appear to be sound.